# $\mathcal{E}motion\mathcal{H}allucer$: Evaluating Emotion Hallucinations in Multimodal Large Language Models

**Bohao Xing**
Lappeenranta-Lahti University of Technology LUT
Finland
bohao.xing@lut.fi

**Xin Liu** *
Lappeenranta-Lahti University of Technology LUT
Finland
linuxsino@gmail.com

**Guoying Zhao**
ELLIS Institute Finland
University of Oulu, Finland
guoying.zhao@oulu.fi

**Chengyu Liu**
Southeast University
China
chengyu@seu.edu.cn

**Xiaolan Fu**
Shanghai Jiao Tong University
China
fuxiaolan@sjtu.edu.cn

**Heikki Kälviäinen**
Lappeenranta-Lahti University of Technology LUT
Finland
Brno University of Technology
Czech Republic
heikki.kalviainen@lut.fi

## Abstract

Emotion understanding is a critical yet challenging task. Recent advances in Multimodal Large Language Models (MLLMs) have significantly enhanced their capabilities in this area. However, MLLMs often suffer from "hallucinations", generating irrelevant or nonsensical content. To the best of our knowledge, and despite the importance of this issue, there has been no dedicated effort to evaluate emotion-related hallucinations in MLLMs. In this work, we introduce **Emotion-Hallucer**, the first benchmark for detecting and analyzing emotion hallucinations in MLLMs. Unlike humans, whose emotion understanding stems from the interplay of biology and social learning, MLLMs rely solely on data-driven learning and lack innate emotional instincts. Fortunately, emotion psychology provides a solid foundation of knowledge about human emotions. Building on this knowledge, we assess emotion hallucinations from two perspectives: emotion psychology knowledge and real world multimodal perception. To support robust evaluation, we utilize an adversarial binary question–answer (QA) framework, which employs carefully crafted basic and hallucinated pairs to assess the emotion hallucination tendencies of MLLMs. By evaluating 41 LLMs and MLLMs on EmotionHallucer, we find that: (1) most current models exhibit substantial issues with emotion hallucinations; (2) closed-source models outperform open-source models in detecting emotion hallucinations, and reasoning capability provides additional advantages; and (3) existing models perform better in emotion psychology knowledge than in multimodal emotion perception. As a byproduct, these findings inspire us to propose the **PEP-MEK** framework, which yields an average improvement of 9.90% in emotion hallucination detection across selected models.

*Inevitably, emotions are inseparable from the idea of good and evil.*

Antonio Damasio, *The Feeling of What Happens.*

---

*Corresponding author.
Code and benchmark: https://github.com/xxtars/EmotionHallucer.

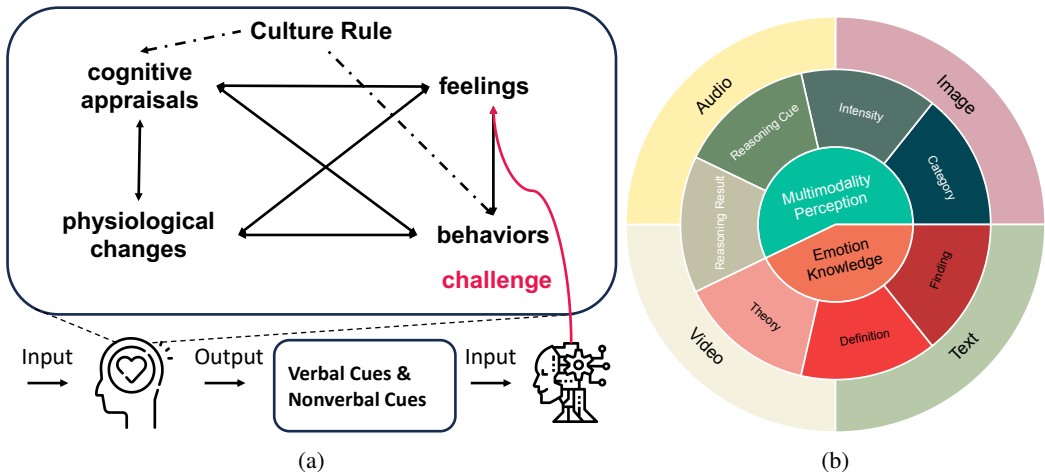

Figure 1: Emotion understanding differences and the EmotionHallucer. **(a)** The difference between how humans and MLLMs understand emotions. Based on the component process model (Scherer, 2009) and a dynamic systems approach (Lewis, 2005), human emotion understanding involves dynamic interactions among cognitive appraisals, physiological changes, feelings, and behaviors. In contrast, MLLMs rely on data-driven learning from external behavioral cues, which limits their ability to accurately infer the underlying emotional states. **(b)** EmotionHallucer is organized along two main dimensions, Emotion Knowledge and Multimodality Perception, and includes seven subcategories across four modalities.

# 1 INTRODUCTION

Emotion understanding is one of the most fundamental yet challenging tasks in AI (Koelstra et al., 2011; Hakak et al., 2017), and has attracted significant attention from the research community (Nandwani and Verma, 2021; Li and Deng, 2020; El Ayadi et al., 2011; Ezzameli and Mahersia, 2023; Rahdari et al., 2019; Xing et al., 2024). Much of the existing work has focused on independent sub-tasks across different modalities: in the text modality, tasks such as sentiment analysis (Wankhade et al., 2022) and emotion cause detection (Lee et al., 2010); in the image modality, facial expression recognition (Li and Deng, 2020; Yuan et al., 2024; Liu et al., 2024c; Hu et al., 2025) and affective scene analysis (Zhao et al., 2021); in the speech modality, speech emotion recognition (Wani et al., 2021); and in the video modality, multimodal emotion recognition (Abdullah et al., 2021), dynamic facial expression recognition (Zhao and Liu, 2021), and body gesture-based emotion recognition (Liu et al., 2021), among others.

Recently, MLLMs have demonstrated remarkable capabilities in textual and visual understanding (Alayrac et al., 2022; Achiam et al., 2023), and have begun to play an increasingly important role in emotion understanding (Zhang et al., 2023b; Lian et al., 2023b; 2025; Yuan et al., 2025; Cheng et al., 2024; Xing et al., 2024). Nevertheless, despite their advanced capabilities, MLLMs often generate incorrect or ungrounded responses when operating based on textual or visual inputs (Li et al., 2023b; Tong et al., 2024; Petryk et al., 2024). This issue of providing misleading information is commonly termed "hallucination" (Rohrbach et al., 2018). Hallucination is generally categorized into two types (Bai et al., 2024): (1) factuality hallucination, where outputs conflict with real-world facts; and (2) faithfulness hallucination, where outputs diverge from input instructions, or provided context, or exhibit internal inconsistencies. In response to these challenges, increasing emphasis has been paid to analyzing and mitigating hallucinations in MLLMs. However, existing hallucination benchmarks are designed primarily for general-purpose tasks (Wang et al., 2024), leaving hallucinations in emotion understanding tasks largely unexplored.

Human emotion arises from a combination of innate biological mechanisms and lifelong social learning (Zeidner et al., 2003), which makes emotions challenging to model, as shown in Figure 1a. Unlike humans, MLLMs rely on data-driven learning and they lack the embodied and experiential grounding that humans use to interpret emotions naturally and intuitively. Fortunately, ***learning how emotion develops can help us understand more about emotion itself*** (Shiota and Kalat, 2017).

Moreover, decades of research in psychology have offered rich insights into how humans perceive, process, and reason about emotions (Niedenthal and Ric, 2017), offering a valuable source of knowledge to support more reliable emotion understanding in MLLMs.

Motivated by these observations, we first consider how emotion-related hallucinations should be defined and categorized. Unlike general hallucinations, emotion hallucinations tend to be more complex, as emotion understanding involves not only objective perception but also psychological and sociological reasoning (Zaki and Ochsner, 2012). In view of this complexity, we propose **EmotionHallucer**, the first benchmark specifically designed to evaluate emotion hallucinations. EmotionHallucer targets two key aspects: hallucinations related to knowledge of emotion psychology (focusing on factuality hallucination) and hallucinations in real-world multimodal emotion understanding (focusing on faithfulness hallucination), as illustrated in Figure 1b. To ensure reliable evaluation and reduce confounding factors (Li et al., 2023b; Zhang et al., 2023a), we adopt a binary QA based evaluation framework (Li et al., 2023b; Wang et al., 2024). Specifically, we construct adversarial QA pairs (Tong et al., 2024), where each pair consists of a basic question and an intentionally hallucinated question to test the models. To mitigate language bias, we balance "yes" and "no" answers, and provide concise explanations to reduce misinterpretation.

By evaluating 41 LLMs and MLLMs on EmotionHallucer, our analysis produces three main findings: **first**, most current models exhibit substantial issues with emotion hallucinations; **second**, closed-source models outperform open-source ones in detecting emotion hallucinations, and reasoning capability provides additional advantages; and **third**, existing models perform better in emotion psychology knowledge than in multimodal emotion perception. Building on these findings, we propose **PEP-MEK**, a plug-and-play framework that incorporates both modality-specific and emotional knowledge to mitigate emotion hallucinations. Experimental results show that applying PEP-MEK leads to a significant performance improvement in results on EmotionHallucer, with an average improvement of 9.90%. We believe this framework can support future research and development in the detection and mitigation of emotion hallucinations in MLLMs. Our main contributions are summarized as follows:

- We present, to the best of our knowledge, the first hallucination benchmark that evaluates emotion psychology knowledge and multimodal perception.

- We conduct a comprehensive evaluation of 41 LLMs and MLLMs, from which we derive three key findings.

- Building on these insights, we propose **PEP-MEK**, and demonstrate through experiments its effectiveness and potential in mitigating emotional hallucinations.

## 2 RELATED WORK

### 2.1 HALLUCINATION IN NATURAL LANGUAGE PROCESSING

Generative models in Natural Language Processing (NLP), particularly Large Language Models (LLMs), have demonstrated impressive performance across a wide range of language generation tasks. However, a major challenge remains: these models may occasionally produce text that is inaccurate, irrelevant, or illogical, a phenomenon commonly referred to as "hallucination" in NLP. Hallucination typically refers to instances where the generated content is nonsensical or deviates from the intended meaning or source material (Filippova, 2020). In the NLP community, this issue is empirically categorized into two types (Bai et al., 2024): (1) factuality hallucination, which highlights inconsistencies between generated output and verifiable real-world facts, often manifesting as factual errors or fabrication; and (2) faithfulness hallucination, which refers to deviations from user instructions, input context, or internal consistency within the generated content. Within specialized research domains, opinions diverge regarding the value of factuality hallucinations. Some studies suggest that such hallucinations can be beneficial (Maynez et al., 2020; Thomson and Reiter, 2020), arguing that the additional information they introduce may enhance the perceived informational value of the output.

## 2.2 HALLUCINATION IN COMPUTER VISION

Recent advances in vision-language modeling have led to impressive performance across various generative tasks (Alayrac et al., 2022; Li et al., 2023a; Achiam et al., 2023). Alongside these developments, increasing attention has been given to the issue of hallucination in this domain. The concept of object hallucination in image captioning, along with the CHAIR metric, was first introduced by Rohrbach et al. (2018). To provide a more robust evaluation framework, POPE (Li et al., 2023b) proposed a binary VQA benchmark specifically aimed at detecting object hallucinations, offering greater reliability than CHAIR. Subsequent studies have broadened the scope of hallucination research to include relationships, attributes, counting, OCR, and other visual phenomena (Sun et al., 2023; Wang et al., 2023a; Guan et al., 2024; Cui et al., 2023; Chen et al., 2024; Liu et al., 2024b). More recently, research has extended to video-based hallucinations, reflecting the growing complexity of multimodal understanding (Zhang et al., 2024; Wang et al., 2024). However, to the best of our knowledge, and despite the emergence of general hallucination benchmarks, no dedicated benchmark has yet been developed to evaluate hallucinations related to emotion understanding. To fill this gap, we introduce the first benchmark specifically designed to assess emotion hallucinations.

## 2.3 EMOTION MLLMS

With the rapid advancement of LLMs, a growing body of research has begun to explore their potential for emotion understanding. These models facilitate the integration of multimodal information, making complex emotional reasoning increasingly feasible. Representative research in this direction includes work presenting the AffectGPT (Lian et al., 2025), EMER (Lian et al., 2023b), Emotion-LLAMA (Cheng et al., 2024), and Omni-Emotion (Yang et al., 2025), which investigate how MLLMs can be adapted for emotion recognition and reasoning. In parallel, other studies have focused on more domain-specific emotion understanding tasks (Li et al., 2024a; Xing et al., 2024; Li et al., 2025), contributing to increasingly flexible and specialized frameworks. However, despite these advancements, the critical issue of hallucination in emotion understanding remains largely underexplored. One key reason for this gap is the lack of a dedicated benchmark to assess hallucinations in emotion understanding. In this work, we address this limitation by introducing the first benchmark specifically designed to evaluate hallucination in emotion understanding.

## 3 THE $\mathcal{E}motion\mathcal{H}allucer$ BENCHMARK

### 3.1 BENCHMARK CONSTRUCTION

To evaluate hallucination in emotion understanding, we divide our benchmark into two primary categories: emotion psychology knowledge and real-world multimodal emotion perception. This design yields seven specific evaluation settings spanning four modalities, as illustrated in Figure 2. For the emotion knowledge dimension, we collected and curated factual statements from an authoritative textbook in emotion psychology (Shiota and Kalat, 2017). For the multimodal perception dimension, we leveraged several widely used datasets across different modalities: SOUL (Deng et al., 2023) for text, Twitter15 and Twitter17 (Zhang et al., 2018) for images, RAVDESS (Livingstone and Russo, 2018) for speech, and MER 2023 (Lian et al., 2023a) and Social-IQ 2.0 (Zadeh et al., 2019; Wilf and collaborators, 2023) for video. These diverse sources allowed us to construct hallucination instances that reflect both knowledge-based and perceptual challenges in emotion understanding. The detailed annotation procedure is described in **Appendix** A.

### 3.1.1 EMOTION PSYCHOLOGY KNOWLEDGE HALLUCINATION

To the best of our knowledge, EmotionHallucer is the first hallucination benchmark that evaluates LLMs and MLLMs on their understanding of emotion psychology knowledge, with a focus on core theories from affective science (Scherer, 2009; Lewis, 2005). We began by selecting a set of well-established and unambiguous statements from an authoritative textbook on emotion psychology (Shiota and Kalat, 2017), which serve as ground truth. Based on these statements, we constructed hallucinated counterparts that intentionally contradict, distort, or misrepresent the original content. This setting enables examination of the susceptibility of models to hallucination within

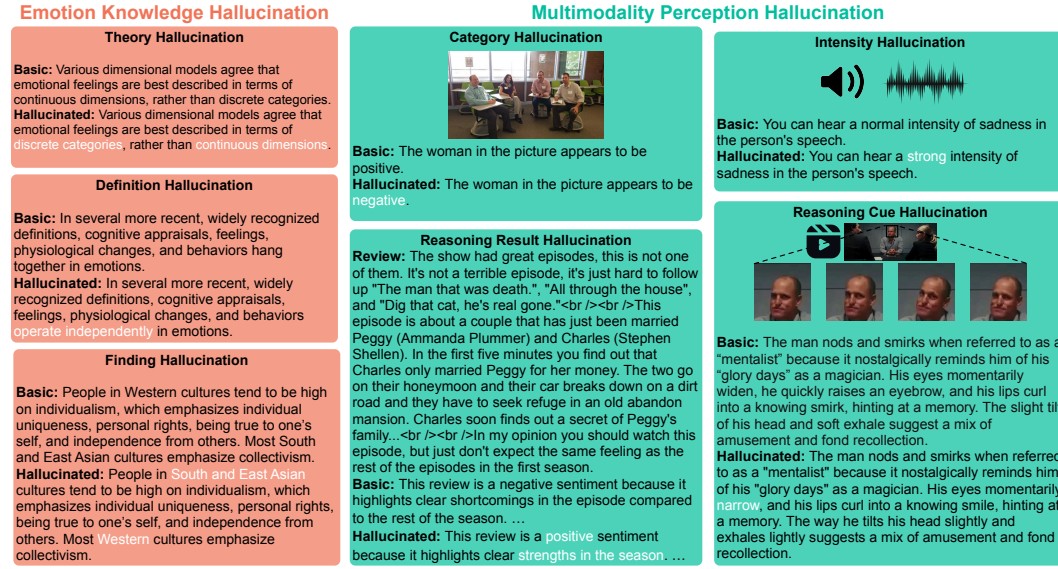

Figure 2: Example tasks in EmotionHallucer. Each pairs consists of a basic question, used to test the basic ability of MLLMs, and a hallucinated question, containing hallucinated content to evaluate the model's ability to detect hallucination. Emotion Knowledge Hallucination targets emotion psychology knowledge (Scherer, 2009; Lewis, 2005; Shiota and Kalat, 2017), whereas Multimodality Perception Hallucination centers on real-world emotion understanding (Deng et al., 2023; Zhang et al., 2018; Livingstone and Russo, 2018; Lian et al., 2023a; Zadeh et al., 2019; Wilf and collaborators, 2023). More details can be found in **Appendix** D.

the domain of emotion knowledge, providing a framework to assess both knowledge grounding and hallucination behaviors in the context of human emotion theory.

**Theory.** Our work introduces an Emotion Psychology Theory Hallucination setting in the context of emotion psychology theory (Cannon, 1927; Scherer, 2009; Lewis, 2005), as illustrated in Figure 2. To support this setting, we collected a set of core statements derived from well-established theoretical frameworks in emotion psychology, covering a range of foundational perspectives (Cannon, 1927; Scherer, 2009; Lewis, 2005; Shiota and Kalat, 2017). Based on these statements, and through an annotation process, we constructed 81 question-answer pairs that serve as test instances for evaluating hallucination in emotion theory knowledge.

**Definition.** We also constructed an Emotion Psychology Definition Hallucination setting, which targets the definitions of widely accepted concepts in emotion psychology (Berkowitz, 1999), as shown in Figure 2. To support this setting, we collected a set of definitions for key emotion-related terms from authoritative academic sources, ensuring that each definition reflects the consensus within the field. This process resulted in 133 question-answer pairs, which serve as evaluation instances for assessing hallucinations related to definitional knowledge in affective science.

**Finding.** We further constructed an Emotion Psychology Finding Hallucination setting, which focuses on empirical findings in emotion research that describe observed phenomena but have not yet been formalized into complete theoretical frameworks (Kagitcibasi, 1997), as shown in Figure 2. This category includes well-documented results such as cross-cultural differences in emotional expression (Hareli et al., 2015), developmental variations between infants and adults (Best et al., 2013), and other empirical observations reported in the literature. This setting yields 178 question-answer pairs, which enable the evaluation of hallucination behaviors related to non-theoretical but empirically grounded knowledge in emotion psychology.

### 3.1.2 MULTIMODALITY PERCEPTION HALLUCINATION

Beyond emotion psychology knowledge, we also constructed a Real-World Multimodal Emotion Perception Hallucination setting, which focuses on assessing hallucination risks in emotion un-

derstanding tasks involving multimodal inputs (e.g., text, audio, and visual signals), as shown in Figure 2. To build this setting, we collected samples from widely used emotion understanding datasets (Deng et al., 2023; Zhang et al., 2018; Livingstone and Russo, 2018; Lian et al., 2023a; Zadeh et al., 2019; Wilf and collaborators, 2023). We then filtered and refined these samples to construct instances in which hallucinated descriptions contradict or misinterpret the emotional content conveyed by the inputs. This setting enables the evaluation of hallucinations in realistic, multimodal environments, where emotion understanding requires integrating ambiguous and context-dependent cues from multiple sources (Zadeh et al., 2017).

**Category.** Following prior work on object hallucination (Rohrbach et al., 2018), and as shown in Figure 2, we introduced an Emotion Category Hallucination setting, which refers to the incorrect generation or identification of emotion categories (Dzedzickis et al., 2020). Specifically, this type of hallucination refers to cases where models generate a nonexistent or inappropriate emotion category. Additionally, the emotion categories in this setting are not limited to binary sentiment labels commonly used in text-based tasks (e.g., positive/negative), but also cover discrete basic emotion categories frequently adopted in visual and multimodal emotion recognition tasks (e.g., happiness, anger, sadness).

**Intensity.** We further introduced an Emotion Intensity Hallucination setting, which concerns the misrepresentation of the strength of emotional expressions, as shown in Figure 2. Unlike traditional approaches in affective computing that represent emotions using continuous valence-arousal dimensions (Russell, 1980), we adopted discrete intensity descriptors (e.g., mild/slightly, normal, strong) that are more aligned with the representational capabilities of LLMs. In this setting, hallucination arises when the model exaggerates, downplays, or otherwise inaccurately describes the intensity of the emotional state conveyed by the input.

**Reasoning Result.** Following previous works (Lian et al., 2023a; Cheng et al., 2024), we introduced a Reasoning Result Hallucination setting. This type of hallucination arises when the model accurately extracts emotional cues from multimodal inputs (e.g., facial expressions, vocal tone, or text), but still produces an incorrect emotional interpretation, as shown in Figure 2. In this setting, the hallucination does not stem from the misidentification of the input signals themselves, but from incorrect emotional reasoning. This design reflects the critical distinction between perception and reasoning in emotion understanding, highlighting that accurate signal recognition does not necessarily guarantee appropriate emotional conclusions.

**Reasoning Cue.** Following previous works (Lian et al., 2023a; Cheng et al., 2024), we introduced a Reasoning Cue Hallucination setting, as shown in Figure 2. In this setting, hallucination arises when the model overlooks, misinterprets, or fabricates key multimodal signals (e.g., missing the angry tone in speech, misreading a facial expression, or inferring unsupported emotional context from text), leading to unreliable reasoning processes. Crucially, Reasoning Cue Hallucination captures cases where the reasoning pathway itself is flawed due to incorrect cue selection or interpretation, independent of the correctness of the final emotion judgment.

## 3.2 BENCHMARK STATISTICS

**Quantitative Analysis.** Table 1 provides an overview of EmotionHallucer. In total, the benchmark comprises 2,742 questions, with an average length of 31.6 words. EmotionHallucer covers four modalities: text, image, audio, and video. We further categorized the data collected from MER 2023Lian et al. (2023a) and Social-IQ 2.0Zadeh et al. (2019); Wilf and collaborators (2023) into short video and long video subsets. More description can be found in **Appendix** A.3.

**Qualitative Analysis.** To provide a more intuitive understanding of EmotionHallucer, key terms are presented using a word cloud, as shown in Figure 3. The visualization highlights frequently occurring concepts such as emotion, voice, facial expression, and tone, which correspond closely to the core modalities and reasoning components in our benchmark. These results suggest that EmotionHallucer effectively focuses on the multimodal cues essential for emotion hallucination.

## 3.3 EVALUATION METRIC

**Hallucination Evaluation.** Following previous work (Tong et al., 2024; Wang et al., 2024), we adopted a QA-based benchmark for the following reasons: (1) Susceptibility to external factors.

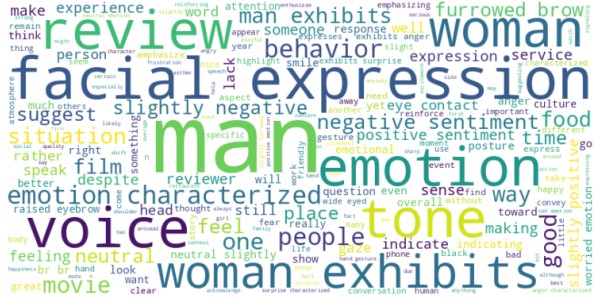

Figure 3: Word cloud of EmotionHallucer.

Table 1: Dataset statistics of EmotionHallucer.

| Statistic | Count |
|---|---|
| Questions | 2,742 |
| Images | 150 |
| Audios | 368 |
| Videos | 230 |
| Avg Question len | 31.6 |
| Avg Knowledge Text len | 19.9 |
| Avg Review Text len | 108.7 |
| Avg Image resolution | $579.5 \times 466.5$ |
| Avg Audio len | 4.2 |
| Avg Short Video len | 4.3 |
| Avg Short Video resolution | $870.0 \times 476.9$ |
| Avg Long Video len | 60.0 |
| Avg Long Video resolution | $637.9 \times 360.0$ |

Caption-based evaluations, like BLEU (Papineni et al., 2002) and ROUGE (Lin, 2004), are sensitive to external variables such as prompt design and caption length (Li et al., 2023b), which can distort evaluation outcomes. (2) Evaluation complexity. Existing approaches such as CHAIR (Rohrbach et al., 2018) rely on intricate, manually designed parsing rules, which complicate the evaluation process and hinder scalability. (3) LLM Hallucination Bias: Given the propensity of LLMs to produce hallucinated outputs, using their own generations for self-evaluation may compromise the reliability and objectivity of the results (Wang et al., 2023a).

To reduce these evaluation biases, EmotionHallucer utilizes an adversarial evaluation framework inspired by (Tong et al., 2024; Wang et al., 2024). For each evaluation instance, we constructed a pair of complementary questions: a basic question, which tests the model's core perception and reasoning abilities, and a hallucinated question, which introduces intentionally fabricated content to test the model's robustness against hallucination. A response is considered correct only if the model answers both questions accurately as a pair. This dual-question design enables a more rigorous assessment of whether a model can detect and resist hallucinations without compromising its performance on fundamental tasks.

**Bias Evaluation.** In addition to the accuracy, we calculated the Yes Percentage Difference (Pct. Diff) and False Positive Ratio (FP Ratio) (Guan et al., 2024; Wang et al., 2024) to reveal the bias in MLLMs. Specifically, the Yes Percentage Difference is calculated as

$$d_y = \frac{\left|\{Pred(m,q) = \text{"yes"}\}_{(m,q) \in V}\right| - \left|\{GT(m,q) = \text{"yes"}\}_{(m,q) \in V}\right|}{|V|}, \quad (1)$$

where $V$ is the set of question pairs, $m$ represents additional modality information such as image, audio, or video (for text-only questions, this component is absent), $q$ refers to the question itself, and $GT(m,q)$ is the ground truth. A smaller $d_y$ indicates that the number of "yes" responses from models is closer to the ground truth, revealing less language bias. The False Positive Ratio is calculated as

$$r_{fp} = \frac{\left|\{Pred(m,q) = \text{"yes"}\}_{(m,q) \in W}\right|}{|W|}, \quad (2)$$

where $W$ is the set of wrongly answered question pairs. $r_{fp}$ gives the percentage of "yes" in all wrongly predicted answers. A value closer to 50% indicates less bias from the models.

## 4 EXPERIMENTS

In this section, we evaluate a range of widely used LLMs and MLLMs on EmotionHallucer. The models are grouped according to the input modalities they support. Implementation details and additional analyses are provided in **Appendix** C.

### 4.1 MAIN BENCHMARK RESULTS

**Multimodality.** The EmotionHallucer benchmark encompasses four modalities: text, image, audio, and video. However, most existing MLLMs lack the ability to process all four modalities simultaneously, with many limited to two modalities (e.g., text-image or text-video). To ensure a fair and consistent evaluation, we report both aggregate results for models capable of handling all four

Table 2: Performance comparison on EmotionHallucer with additional "Yes/No bias" analysis.

| Methods | Model | Yes/No Bias | | Accuracy on EmotionHallucer | | |
|---|---|---|---|---|---|---|
| | Size | Pct. Diff (∼0) | FP Ratio (∼0.5) | Basic ↑ | Hallucinated ↑ | Overall ↑ |
| *Open-source* | | | | | | |
| Qwen2.5-Omni (Xu et al., 2025) | 7B | -0.05 | 0.44 | 52.81 | 63.46 | 18.65 |
| Emotion-LLaMA (Cheng et al., 2024) | 7B | 0.20 | 0.71 | 72.88 | 33.45 | 15.43 |
| *Closed-source* | | | | | | |
| Gemini-2.5-Flash (Google DeepMind, 2025) | - | 0.01 | 0.51 | 69.41 | 68.15 | 45.06 |
| Gemini-2.5-Pro (Google DeepMind, 2025) | - | 0.01 | 0.52 | 70.30 | 67.56 | 44.17 |

Table 3: Performance comparison on EmotionHallucer-NoAudio with additional "Yes/No bias" analysis. More results can be seen in Table 17 of the **Appendix**.

| Methods | Model | Yes/No Bias | | Accuracy on EmotionHallucer–NoA | | |
|---|---|---|---|---|---|---|
| | Size | Pct. Diff (∼0) | FP Ratio (∼0.5) | Basic ↑ | Hallucinated ↑ | Overall ↑ |
| *Open-source* | | | | | | |
| LLaVA (Liu et al., 2023) | 34B | -0.05 | 0.45 | 50.25 | 59.52 | 10.27 |
| Llama3.2-vision (Grattafiori et al., 2024) | 11B | 0.21 | 0.78 | 83.05 | 41.28 | 29.91 |
| Video-ChatGPT (Maaz et al., 2023) | 7B | 0.09 | 0.59 | 61.91 | 44.67 | 18.44 |
| Emotion-LLaMA (Cheng et al., 2024) | 7B | 0.12 | 0.63 | 66.55 | 42.43 | 18.86 |
| Qwen2.5-VL (Bai et al., 2025) | 72B | 0.08 | 0.63 | 78.08 | 62.15 | 43.02 |
| Qwen2.5-Omni (Xu et al., 2025) | 7B | 0.11 | 0.65 | 72.39 | 49.74 | 25.44 |
| *Closed-source* | | | | | | |
| QvQ-Max (Qwen Team, 2025) | - | 0.07 | 0.63 | 78.18 | 63.39 | 47.98 |
| GPT-4o (Hurst et al., 2024) | - | -0.01 | 0.48 | 67.10 | 69.49 | 40.98 |
| GPT-5 (OpenAI, 2025b) | - | -0.06 | 0.40 | 67.10 | 78.17 | 49.35 |
| Gemini-2.5-Flash (Google DeepMind, 2025) | - | 0.06 | 0.61 | 78.55 | 66.80 | 50.56 |
| Gemini-2.5-Pro (Google DeepMind, 2025) | - | 0.07 | 0.64 | 81.31 | 67.01 | 51.58 |

modalities and modality-specific results for models restricted to certain subsets. Additional analyses on model performance and modality-specific analysis are provided in **Appendix** C.3.

As shown in Table 2, we compare models across all modalities available on EmotionHallucer. While the strongest closed-source Gemini models outperform their open-source counterparts, their overall performance remains suboptimal, reflecting the inherent challenges of emotional understanding and hallucination. Notably, all open-source models fail to exceed the 25% accuracy expected from random guessing, also underscoring the current limitations of MLLMs in handling emotional reasoning. In the "Yes/No Bias" evaluation, general-purpose MLLMs demonstrate better neutrality, showing less tendency toward overconfident affirmations. In contrast, the emotion-specific model Emotion-LLaMA performs significantly worse in this regard. This may be attributed to its fine-tuning focus on emotional content. It may also be because the model is relatively outdated compared to recent MLLMs, potentially lacking the latest advances.

Given that most popular MLLMs are optimized for vision modalities and lack support for audio-only inputs, we developed EmotionHallucer–NoAudio, a benchmark subset that excludes the audio modality (see Table 3). Due to space constraints, we report results for 11 representative models. Consistent with earlier findings, closed-source models outperform their open-source counterparts in both overall accuracy and the "Yes/No Bias" evaluation. Gemini-2.5-Pro achieves the best performance, followed closely by Gemini-2.5-Flash. Among open-source models, Qwen2.5 VL performs the best and notably exceeds the random-guessing baseline.

**Unimodality.** To enable a more fine-grained understanding of MLLM behavior across tasks and modalities, we report performance on each individual modality. As shown in Figure 4, models perform best on emotion knowledge, with accuracy decreasing steadily across perception-based tasks, from text to image, and further to audio and video. This trend may be attributed to two main factors: (1) current training is predominantly focused on text data, enhancing models' performance on knowledge-oriented tasks; and (2) a lack of high-quality emotional annotations in modalities,

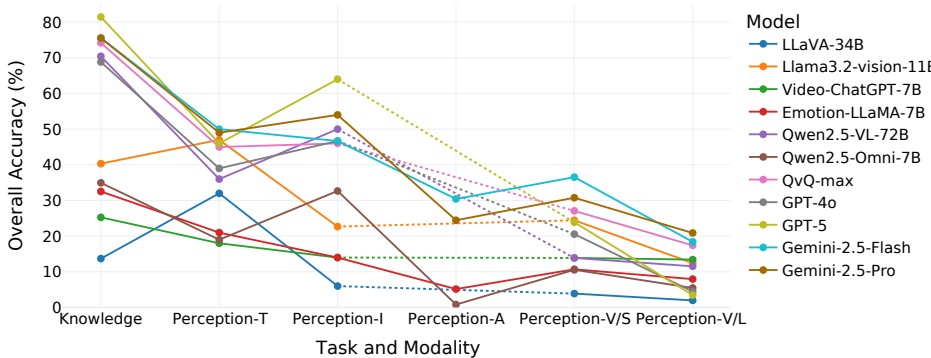

Figure 4: Unimodal performance of partial selected models. T, I, A, V/S, and V/L stand for Text, Image, Audio, Short Video, and Long Video, respectively. Additional models and implementation details are provided in **Appendix** C.

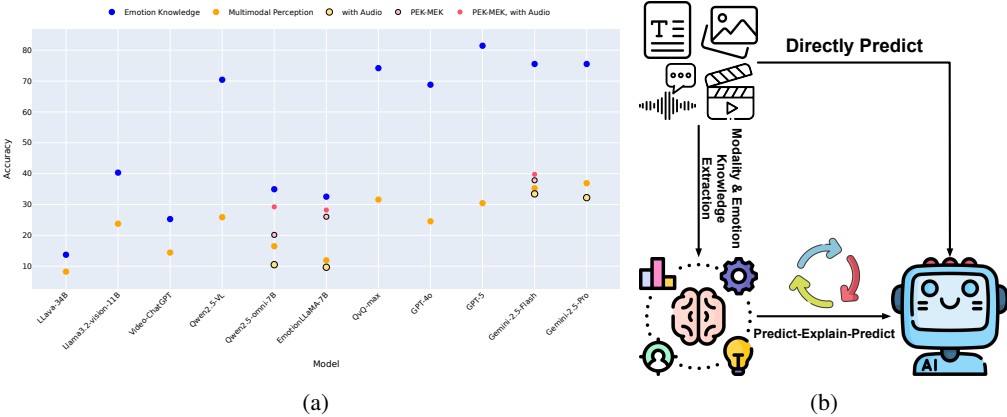

(a)                                                                                    (b)

Figure 5: Emotion hallucination analysis and our proposed framework. **(a)** Comparison of hallucination detection in emotion knowledge and multimodal perception. **(b)** The PEP-MEK Framework.

which limits the models' ability to learn fine-grained emotion understanding. These results underscore a critical future direction: improving cross-modal emotion understanding through better data quality and balanced modality training. Additional detailed results and analysis are provided in **Appendix** C.

# 5    PREDICT-EXPLAIN-PREDICT WITH MODALITY AND EMOTION KNOWLEDGE

We observed that most models perform significantly worse under the multimodal perception hallucination setting compared to the emotion knowledge hallucination setting in EmotionHallucer. Motivated by this observation, we propose a simple yet effective explanation-based method to mitigate hallucination issues arising from multimodal emotion perception errors.

## 5.1    EMOTION KNOWLEDGE VS. MULTIMODAL PERCEPTION HALLUCINATION DETECTION

Model performance on emotion knowledge and multimodal perception is shown in Figure 5a. A key observation is that most models achieve significantly lower accuracy on multimodal perception, highlighting a substantial gap compared to their performance on structured emotion knowledge. This result suggests that while current models handle structured knowledge reasonably well, they struggle with real-world emotion understanding. Accordingly, this section focuses on enhancing model robustness against multimodal perception hallucinations. Among the evaluated models, QvQ-

Table 4: Results of the PEP-MEK Framework.

| Methods | Model Size | Yes/No Bias | | Accuracy on EmotionHallucer-P | | |
|---|---|---|---|---|---|---|
| | | Pct. Diff (∼0) | FP Ratio (∼0.5) | Basic ↑ | Hallucinated ↑ | Overall ↑ |
| *Open-source* | | | | | | |
| Qwen2.5-Omni (Xu et al., 2025) | 7B | -0.18 | 0.30 | 35.51 | 71.96 | 10.49 |
| *+PEP-MEK* | | -0.19 | 0.29 | 37.49 | 74.87 | 20.15 |
| Emotion-LLaMA (Xu et al., 2025) | 7B | 0.28 | 0.77 | 77.34 | 22.02 | 9.65 |
| *+PEP-MEK* | | -0.08 | 0.41 | 46.35 | 62.86 | 26.03 |
| *Closed-source* | | | | | | |
| Gemini-2.5-Flash (Google DeepMind, 2025) | - | 0.00 | 0.50 | 61.54 | 62.05 | 33.44 |
| *+PEP-MEK* | | -0.07 | 0.40 | 56.66 | 71.17 | 37.84 |

Max outperforms Qwen2.5-VL, which we attribute to differences in its reasoning paradigms. Given that LLMs are generally more adept at fact detection than hallucination detection (Ji et al., 2023), we hypothesize that incorporating a more structured and targeted reasoning framework, enriched with emotion-specific knowledge, may further reduce emotion hallucination in MLLMs.

## 5.2 PEP-MEK FRAMEWORK

Building on these findings, we propose a novel framework, the Predict-Explain-Predict with Modality and Emotion Knowledge (PEP-MEK) framework, designed to enhance MLLMs' performance in detecting multimodal emotion hallucinations. Figure 5b illustrates the framework of the PEP-MEK. Rather than relying on direct predictions, PEP-MEK introduces an intermediate reasoning stage to improve model transparency and decision reliability in emotion understanding. First, PEP-MEK uses prompts to guide the model in autonomously extracting modality-specific and emotional knowledge from the input (text, image, audio, video). This extracted information is then combined with the original input for an initial prediction. Next, the model generates an explanation for its prediction using both the input and the extracted knowledge. Then, the explanation is incorporated to refine the prediction. This process enables the model to reason more effectively and reduce hallucinations.

The effectiveness of PEP-MEK evaluated on three representative MLLMs with support for all modalities: a general-purpose model (Qwen2.5-Omni), an emotion-specific model (Emotion-LLaMA), and a closed-source model (Gemini-2.5-Flash), is shown in Table 4. The results demonstrate that PEP-MEK consistently improves performance, yielding an average accuracy gain of 9.9% across models. Notably, Emotion-LLaMA benefits the most from PEP-MEK, achieving a 16.38% increase in overall accuracy and a substantial reduction in bias-related metrics. Full implementation details, ablation study, and qualitative examples are provided in **Appendix** B.

## 6 CONCLUSION

In this work, we introduced EmotionHallucer, the first benchmark designed to detect emotion hallucinations in MLLMs. Our adversarial evaluation strategy provides a rigorous assessment of both emotion understanding and hallucination susceptibility. By evaluating hallucinations from two complementary perspectives, emotion psychology knowledge and multimodal emotion perception, EmotionHallucer enables fine-grained analysis of large models. Testing of 41 LLMs and MLLMs revealed the pervasiveness of emotion hallucinations, especially in tasks requiring perception across modalities. To address these challenges, we propose PEP-MEK, a framework that showed good potential for reducing emotion hallucinations and improving model robustness. These findings highlight critical limitations in current emotion MLLMs and point to promising directions for advancing emotion-aware, multimodal reasoning in future research.

ACKNOWLEDGMENTS

G. Zhao was supported by Research Council of Finland (former Academy of Finland) Academy Professor project EmotionAI (grants 336116, 345122, 359894), HPC project FaceCanvas (grant number 364905), the University of Oulu & Research Council of Finland Profi 7 (grant 352788), and EU HORIZON-MSCA-SE-2022 project ACMod (grant 101130271).

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

APPENDIX

To facilitate deeper understanding and reproducibility, we organize the appendix as follows:

- **Appendix** A presents additional information on benchmark construction and annotation procedures, and statistics.
- **Appendix** B details the implementation and ablation study of PEP-MEK.
- **Appendix** C includes experiment setup and additional experimental results along with in-depth analysis.
- **Appendix** D provides more examples.
- **Appendix** E discusses the limitations of our work.
- **Appendix** F discusses the directions for future improvement.
- **Appendix** G is the Ethics Statement.
- **Appendix** H is the Reproducibility Statement.
- **Appendix** I is the LLM Usage Statement.

## A    BENCHMARK COLLECTION, ANNOTATION, AND STATISTICS

Here, we provide annotation details to better illustrate our data sources as well as each modality and category. The annotation process begins with the collection of knowledge from emotion psychology and existing emotion understanding datasets. This foundational step ensures that our annotation is grounded in well-established emotional concepts and tasks. Next, we engage a group of trained annotators to carefully filter and review the initial QA. Their task is to ensure that each QA pair is accurate, coherent, and well-grounded in the corresponding source content. Following this, for each basic question, annotators are instructed to generate a corresponding hallucinated question. To ensure high annotation quality, we adopt a cross-review verification mechanism. Each question (basic and hallucinated) is independently reviewed by a second annotator. Only when both annotators reach consensus is the item included in the final benchmark. The overall annotation workflow is shown in Figure 6.

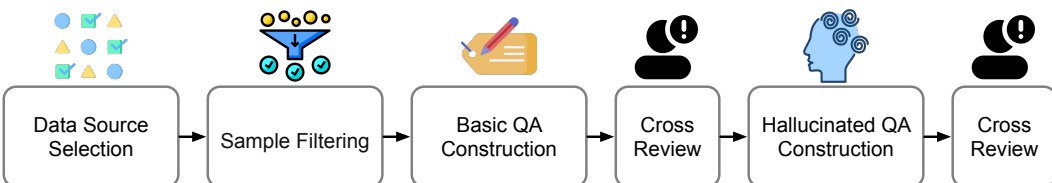

Figure 6: Overview of the annotation workflow. The process consists of four key stages: selecting relevant data sources, filtering for appropriate samples, constructing basic QA pairs, and generating hallucinated QA variants by applying diverse hallucination strategies. Each QA construction step is followed by a cross-review phase to ensure quality and consistency across annotators. This unified pipeline is applied across all modalities, including text, image, audio, and video.

### A.1    COLLECTION AND ANNOTATION ACROSS MODALITIES

We first present our data collection and annotation procedures from a modality perspective to provide a clearer understanding of our data sources and the original objectives of each dataset.

### A.1.1    TEXT

To ground our annotation process in established affective science, we selected a leading textbook in the field of emotion psychology: Emotion (3rd Edition) by Michelle N. Shiota and James W. Kalat (Shiota and Kalat, 2017). This book offers a comprehensive overview of key emotional theories (Cannon, 1927; Scherer, 2009; Lewis, 2005), physiological and cognitive underpinnings of emotions (Berkowitz, 1999), as well as cross-cultural perspectives (Kagitcibasi, 1997; Hareli et al.,

2015; Best et al., 2013). It is widely used in academic settings and is recognized for its balanced integration of classical theories (e.g., James-Lange (Cannon, 1927), Schachter-Singer) and contemporary research findings. The text also emphasizes real-world applications and experimental paradigms (Scherer, 2009; Lewis, 2005), making it a suitable foundation for constructing psychologically grounded emotion-related tasks. We collected a subset of statements from the book that conveyed clear emotion psychology viewpoints and insights. These statements were carefully selected through manual review to ensure their clarity and relevance. For each selected statement, the basic question was taken directly from the original text, preserving the emotional concept or psychological mechanism as presented in the source. After the basic QA pair was established, annotators were instructed to generate a hallucinated version. Common strategies included addition, deletion, or distortion of key concepts or the intension and extension of a given inference, while ensuring that the question remained grammatically fluent and superficially plausible. As a result, 784 basic and hallucination QA pairs were generated.

In the real-world text modality emotion understanding, we incorporate existing datasets to better examine emotion perception hallucinations in real-world textual contexts. One such dataset is SOUL (Deng et al., 2023), which is specifically designed to evaluate fine-grained understanding of sentiment, stance, and opinion in natural language. In its original task format, SOUL presents a review text alongside multiple statements that reflect potential interpretations of the opinion expressed. Each statement is annotated as either correct or incorrect, depending on its emotional alignment with respect to the original review. This setup goes beyond traditional sentiment analysis by emphasizing deeper dimensions of subjective understanding, such as emotional appropriateness, implicit attitude, and contextual nuance. In our annotation process, for each review text, we collected all associated statements and manually refined them into a single, well-grounded summary statement. This statement was used as the basic question in our QA format. This approach allowed our QA pairs to inherit SOUL's emphasis on deeper dimensions of subjective understanding, thus enabling more fine-grained evaluation of MLLMs' capabilities as regards emotion understanding and hallucination detection. As a result, 200 basic and hallucination QA pairs were generated.

### A.1.2 IMAGE

We utilized the Twitter15 and Twitter17 datasets (Zhang et al., 2018), which were originally developed for named entity recognition and sentiment analysis in social media contexts. These datasets consist of tweets accompanied by images and structured annotations, including entity labels and overall tweet-level sentiment polarity (positive, negative, neutral). The data reflects real-world, noisy, and multimodal content, making it a valuable resource for emotion perception tasks.

In our annotation process, we focused on images. Notably, we observed that many images contain multiple people, often displaying divergent emotional expressions within the same instance, as shown in Figure 10. This introduces a unique challenge for MLLMs, as it requires fine grained emotion understanding and the ability to disambiguate multi-actor sentiment based on visual cues. To reflect this, we reformulated each image into a QA format, selecting or synthesizing emotionally consistent description as basic questions, and constructing hallucinated variants by introducing emotionally incongruent elements, such as misattributing emotions to the wrong person or exaggerating emotional intensity. As a result, we generated 300 basic and hallucination QA pairs and 150 images.

### A.1.3 AUDIO

We incorporated the RAVDESS dataset (Livingstone and Russo, 2018), which provides a multi-modal collection of emotional expressions through both facial and vocal modalities. The dataset consists of high-quality recordings in North American English, featuring actors performing scripted speech and song with controlled emotional expressions. Emotions span a wide range (including calm, happy, sad, angry, fearful and more), and are further labeled with varying levels of intensity, making RAVDESS a rich resource for studying graded emotional cues.

In our annotation process, we used only the audio modality as input, focusing on how emotion is conveyed through vocal tone and prosody rather than linguistic content. For each selected audio clip, annotators generated a basic question that captured the emotional tone perceived from the speech alone, based on the original emotion labels provided in the dataset. Hallucinated variants were then constructed by subtly misrepresenting the emotional valence or intensity, such as exaggerating,

downplaying, or inverting the expressed emotion, allowing us to assess the sensitivity of MLLMs to audio signals without relying on semantic content. As a result, we generated 736 basic and hallucination QA pairs and 368 audio clips.

### A.1.4 VIDEO

Videos contain both visual and auditory information, and they represent the most natural and comprehensive modality for capturing human emotional experiences. Motivated by this, we selected two data sources, MER 2023Lian et al. (2023a) and Social-IQ 2.0Zadeh et al. (2019); Wilf and collaborators (2023), to evaluate emotion understanding hallucination in realistic, multimodal contexts. Both datasets involve video-based social situations, where emotion understanding relies not only on speech and text, but also on facial expressions, gestures, and interpersonal dynamics.

MER 2023 focuses on multimodal emotion recognition on sentence-length inputs. Each instance consists of a short video clip accompanied by aligned textual transcripts, from which the model is expected to identify the expressed emotion based on the combined audiovisual and textual cues. It emphasizes how emotions are conveyed through brief, but richly multimodal expressions.

Social-IQ 2.0, in contrast, is designed to assess social intelligence in multimodal contexts, requiring models to understand and reason about complex human interactions. Each video clip in the dataset portrays a real-world social scenario sourced from platforms like YouTube, often involving interpersonal communication, emotional exchanges, or goal-directed behavior. Importantly, each video is paired with multiple questions, each targeting a different aspect of the scene, such as the emotional state of a person, their underlying intentions, or the appropriateness of their response. It transcends traditional discrete emotion concepts and aligns more closely with the diverse emotional experiences encountered in real-world contexts. Each question is accompanied by several candidate answers, only one of which is correct, thereby framing the task as a multiple-choice question-answering problem.

In our annotation process, for MER, we first expanded the original emotion label into a basic reasoning process that incorporates both verbal and non-verbal cues. This reasoning was then formulated into a basic question that captures the emotion and its underlying justifications. Each annotator was subsequently tasked with generating a hallucinated version of the question by subtly altering key aspects of the emotional reasoning, for example, by misattributing the cause of the emotion, exaggerating its intensity, or neglecting relevant multimodal cues. As a result, we got 360 basic and hallucination QA pairs and 180 videos.

For Social-IQ 2.0, the annotation process was more complex. We preserved the original multiple-question-per-video structure to maintain the richness of the reasoning context. First, we filtered out ambiguous or poorly defined questions. For the remaining high-quality questions, we refined and expanded each one, along with its correct answer, into a more detailed version, which served as the basic question in our QA format. Annotators then created hallucinated variants by introducing subtle distortions in social or emotion understanding, such as misinterpreting intentions, assigning incorrect emotional reactions. This enabled us to evaluate MLLMs' ability to handle hallucinations in multimodal contexts. As a result, we generated 402 basic and hallucination QA pairs and 50 videos.

### A.2 ANNOTATION ACROSS CATEGORIES

### A.2.1 THEORY

Throughout the development of emotion psychology, various emotional theories have emerged, each reflecting different perspectives on the nature, origin, and structure of emotions (Cannon, 1927; Scherer, 2009; Lewis, 2005; Shiota and Kalat, 2017). In this setting, we focus on evaluating the capability of MLLMs to detect hallucinations in these theoretical frameworks, rather than judging the correctness or scientific validity of the theories themselves.

To clearly signal the basis of each item, we typically prefaced the question with phrases such as "According to [Theory Name] . . . " or "[Theory Name] argues that . . . ", followed by a concise statement derived from the theory. When constructing hallucinated questions, we deliberately manipulated key aspects of the original statement, such as reversing causal relationships, disrupting conceptual se-

quences, or modifying the intension or extension of key emotional constructs. This strategy allows us to assess whether the model can factually understand the theoretical content.

### A.2.2 DEFINITION

In this setting, we focus on evaluating MLLMs' capability to detecting hallucinations in commonly accepted definitions from emotion psychology (Berkowitz, 1999), which can help models better align with human emotion understanding. To probe their sensitivity, we typically preface the question with phrases such as "[Concept] is . . . ", followed by a definition in psychology.

When constructing hallucinated questions, we deliberately manipulate critical aspects of the original definitions, introducing subtle distortions in the intension (i.e., the core conceptual meaning) or the extension (i.e., the range of phenomena the concept applies to) of emotional constructs.

For example, a widely accepted definition of anxiety might state: "Anxiety is a general expectation that something bad might happen, without identifying any particular danger." To construct a hallucinated version, we might alter it to: "Anxiety is a general expectation that something good might happen, without identifying any particular danger." This version subtly but significantly distorts the core meaning of anxiety by replacing negative anticipation with positive anticipation, which undermines the emotional construct's essential nature.

Such hallucinated definitions are designed to appear structurally and linguistically plausible, yet semantically incorrect. They serve as a diagnostic tool to assess whether MLLMs can distinguish between valid and flawed emotional concept definitions.

### A.2.3 FINDING

In this setting, we focus on evaluating MLLMs' capability to detect hallucinations in a set of empirical findings about emotion. Unlike formalized theories, these findings refer to observed regularities or patterns in emotional expression, perception, and regulation, often cross-cultural or physiological in nature, without necessarily forming a comprehensive theoretical framework (Kagitcibasi, 1997; Hareli et al., 2015; Best et al., 2013; Smith, 1989).

When constructing hallucinated questions, we deliberately altered the scope or the core semantics of the original statement. These manipulations may involve reversing cultural norms, misattributing physiological phenomena, or distorting the contextual boundaries of emotion.

For example, consider the empirical statement: "Japanese consider it inappropriate to show negative emotion to a high-status person." A hallucinated version would state: "Japanese consider it appropriate to show negative emotion to a high-status person." While structurally similar, this altered version directly contradicts established cross-cultural observations, thereby testing whether the model can detect such culturally incongruent assertions.

This type of adversarial probing helps reveal how well MLLMs understand fine-grained emotion, particularly in contexts involving cultural norms, physiological responses, or socio-emotional regulation. Such questions may be challenging even for humans unfamiliar with the specific context, yet the patterns they rely on often stem from deeply ingrained biological or sociocultural tendencies. Therefore, the ability of a model to distinguish between accurate and distorted emotional knowledge can serve as a strong indicator of its deeper emotional competence and alignment with human psychological realities.

### A.2.4 CATEGORY

In this setting, we focused on real-world emotion understanding tasks, among which emotion recognition serves as a fundamental component (Dzedzickis et al., 2020). In current standard setups, this typically includes text-based sentiment analysis tasks, which often rely on coarse sentiment labels such as positive or negative. However, we also considered discrete basic emotion categories, such as happiness, anger, and sadness, which are widely adopted in visual and multimodal emotion recognition tasks. Beyond these categories, we incorporated insights from recent work such as OV-MER (Lian et al., 2024), which proposes a more fine-grained and naturalistic taxonomy of emotions. This framework extends beyond basic emotions by incorporating more nuanced, everyday emotional expressions (e.g., relieved, anxious, content) to better capture the richness of human affective states.

In this context, we constructed hallucinated examples by intentionally distorting the original emotion label annotations. Specifically, we substituted the original emotion with a semantically related or contrasting category. These substitutions can be adjacent in meaning (e.g., replacing angry with serious) or diametrically opposed (e.g., replacing angry with happy), depending on the intended degree of challenge.

Such manipulations allowed us to probe whether MLLMs can detect inconsistencies in emotional labeling, and whether they possess a sufficiently fine-grained understanding of affective semantics to distinguish between closely related or contrasting emotions. This process resulted in 334 question-answer pairs.

### A.2.5   INTENSITY

In our setting, intensity refers to the strength or magnitude of an emotional state as expressed in natural language. Rather than relying on continuous valence-arousal scales commonly used in traditional affective computing (Russell, 1980), we adopted discrete verbal descriptors (e.g., mild, normal, strong), which align more closely with everyday language and the reasoning capabilities of LLMs.

In this context, hallucinations occur when the model misjudges emotional intensity, either by exaggerating, underestimating, or otherwise misrepresenting the degree of emotion expressed in the input. For instance, changing "He seemed slightly annoyed by the interruption" to "He seemed very annoyed by the interruption." This form of distortion challenges the model's ability to capture nuanced emotional gradients and detect subtle affective cues, which is essential for applications requiring fine-grained emotional understanding, such as empathetic communication, psychological assessment, and emotion-aware content generation. This process resulted in 325 question-answer pairs.

### A.2.6   REASONING RESULT

In the framework, reasoning result refers to the final emotional interpretation that a model generates after perceiving and integrating multimodal emotional cues such as facial expressions, vocal tone, and textual context. Unlike perception-level errors, which involve failures in detecting or extracting relevant signals, reasoning result hallucinations arise when the model correctly identifies input cues but still draws an incorrect or implausible emotional conclusion based on faulty reasoning or inference.

For example, given a scenario where a person has a furrowed brow, a tense voice, and says "I can't believe this happened again," a correct interpretation might be frustration. A hallucinated reasoning result would be labeling this as excitement or anger, despite all emotional cues being correctly perceived. In this case, the failure lies not in perception but in misinterpreting the emotional meaning of those cues.

This distinction emphasizes that accurate recognition of emotional signals is a necessary but insufficient condition for successful emotion understanding. Proper emotional inference requires coherent reasoning over the perceived inputs, making reasoning result hallucinations a key diagnostic for evaluating models' deeper emotional competence. This process resulted in 181 question-answer pairs.

### A.2.7   REASONING CUE

In addition, we defined a reasoning cue hallucination, which targets failures in identifying, attending to, or correctly interpreting the emotional cues necessary for sound emotional reasoning—regardless of whether the final emotion classification is ultimately correct or not. In this setting, hallucinations occur when the model overlooks, misinterprets, or fabricates critical multimodal signals, such as failing to detect an angry tone in speech, misreading a sad facial expression as neutral, or drawing unsupported emotional implications from text.

We considered two primary forms of cue hallucination: (1) cases where a specific cue is modified while the final emotional result remains unchanged, thereby testing the model's robustness to misleading or missing signals; and (2) cases where multiple key cues are altered simultaneously,

often accompanied by a shift in the final emotion prediction, which challenges the model's entire reasoning chain, from perception to conclusion.

This setting highlights the importance of the reasoning pathway itself, rather than just the final output. A model may occasionally reach the correct emotion label by chance or superficial pattern matching, but reasoning cue hallucination reveals whether the underlying process is semantically justified and grounded in valid emotional evidence. As such, this setting is crucial for assessing models' interpretability and trustworthiness in real-world emotion understanding scenarios. This process resulted in 159 question-answer pairs.

### A.3 MORE BENCHMARK STATISTICS

As shown in Table 1, for the emotion knowledge text, lengths range from 4 to 56 words, with an average of 19.9 words. In the case of real-world review textsDeng et al. (2023) (see task details in **Appendix** A), the length ranges from 12 to 199 words, averaging 108.7 words. For real-world images (Zhang et al., 2018), resolutions range from $320 \times 194$ to $600 \times 1024$ pixels, with an average resolution of $579.5 \times 466.5$ pixels. The real-world audio clips (Livingstone and Russo, 2018) vary from 3.0 to 6.3 seconds in length, with an average duration of 4.2 seconds. For real-world videos, short videos (Lian et al., 2023a) have an average resolution of $870.0 \times 476.9$ pixels, ranging from $576 \times 256$ to $1280 \times 720$ pixels. Their durations range from 0.52 to 38.29 seconds, with an average of 4.29 seconds. In contrast, long videos (Zadeh et al., 2019; Wilf and collaborators, 2023) have a fixed duration of 60 seconds. Their resolution averages $637.9 \times 360.0$ pixels, ranging from $534 \times 360$ to $640 \times 360$ pixels.

## B IMPLEMENTATION DETAIL AND ABLATION STUDY OF PEP-MEK

### B.1 IMPLEMENTATION DETAIL OF PEP-MEK

As illustrated in Figure 5b, there are two main components of the PEP-MEK: modality and emotion knowledge extraction and predict-explain-predict.

For the modality and emotion knowledge extraction, we asked the model to extract the modality (Yin et al., 2024) and emotion knowledge. Figure 7 shows the prompt we designed to guide modality and emotion knowledge extraction. It considers both modality-specific knowledge (e.g., visual cues such as what can be seen in an image, or auditory cues such as what can be heard in speech) and emotion-specific knowledge (e.g., facial expressions, body posture, vocal prosody, emotion types and definitions, overall emotional atmosphere, and inferred causes). By structuring the prompt, we aimed to ensure that the model captures contextually grounded and psychologically meaningful emotional signals across modalities.

Next, we prompted the model to generate an initial answer by combining the extracted modality and emotion knowledge with the target question. As shown in Figure 8, the prompt instructs the model to base its judgment on the input content (e.g., video, image, audio, or text), and, if needed, to incorporate the accompanying structured knowledge to refine its answer. The model is explicitly asked to respond with only one word, YES or NO, to enforce clarity in evaluation, without generating additional explanations.

We then prompted the MLLM to provide an explanation for its initial answer by referencing the extracted modality and emotion knowledge, the target question, and the initial answer. As shown in Figure 9, the model was asked to first generate a detailed explanation of its reasoning, then assess the factual accuracy and logical soundness of its own explanation, and finally re-affirm its conclusion with a concise binary answer (YES or NO). This structured response format allows us to evaluate not only the model's decision, but also the justification process and self-consistency behind it.

### B.2 ABLATION STUDY ON PEP-MEK

#### B.2.1 DIFFERENT COMPONENTS IN PEP-MEK

As shown in Table 5, we evaluate the contributions of different components in PEP-MEK. MEK denotes incorporating modality and emotion knowledge to perform a prediction, while MEK+Explain

Please extract useful modality and emotional knowledge from the given text/image/audio/video. Your goal is to gather interpretable features that can support accurate emotion prediction, explanation, and refinement. Structure your response according to the following components:

**1. Overall Scene Mood and Context**

- Describe the general emotional atmosphere (e.g., joyful, tense, melancholic, peaceful).
- Identify contextual elements in the environment (e.g., indoor/outdoor, weather, time of day, symbolic objects like candles, rain, broken glass).
- What kind of situation or event might the scene represent (e.g., birthday party, farewell, conflict)?
- For audio/text: consider background sounds (e.g., music, ambient noise) or narrative tone (e.g., hopeful, sarcastic, ominous).

**2. Human Presence and Character Analysis**

- How many people are in the scene (visually or described in text/audio)?
- For each individual:
  - Physical characteristics (e.g., age, gender, clothing style, if visible or described)
  - Facial expression (e.g., happy, sad, fearful, angry, neutral, from video/image)
  - Head pose and gaze direction (e.g., direct gaze, looking away, tilted head)
  - Body posture and hand gesture (e.g., open arms, clenched fists, self-touch, leaning in or away)
  - Voice tone and prosody (e.g., trembling, rising pitch, flat tone, fast or slow pace)
  - Verbal emotional cues (e.g., exclamations, emotional word choice, metaphorical language)
  - Speech content (e.g., direct expression of emotion: "I'm scared," or indirect hints: "I don't know what to do anymore.")

**3. Social and Emotional Interactions**

- Describe the relationships or interactions between people:
  - Are they physically close or distant?
  - Is there eye contact, mirroring expressions, or body synchronization?
  - Is any touch present (e.g., hugging, pushing, hand-holding)?
  - Do they seem emotionally aligned or in conflict?
  - In audio/text: Are their voices overlapping? Do they interrupt, agree, or show empathy?

**4. Emotion Type, Intensity, and Diversity**

- For each person, specify the likely dominant emotion and estimate its intensity (e.g., mild sadness vs. intense grief).
- Are there multiple emotions present in the video/audio/text, possibly conflicting ones?

**5. Emotion Knowledge and Reasoning Support**

- For each identified emotion:
  - Provide a short definition and its typical visual, auditory, or linguistic cues. Example: "Fear is a response to perceived threat, often expressed by widened eyes, raised eyebrows, tense voice, and avoidance language."
  - If possible, suggest potential causes or triggers of these emotions based on the multimodal context.

Figure 7: Prompt for modality and emotion knowledge extraction.

Please provide a clear response to the question below by watching the video (reading the text, watching the image, listening the audio).
If necessary, you can also use the accompanying modality and emotion knowledge to help refine your answer.
Modality and emotion knowledge: {knowledge}
Question: {question}
Please answer with ONLY ONE WORD: YES or NO. Do not provide any explanation or additional output.

Figure 8: Prompt for modality and emotion knowledge extraction.

First, please provide a detailed explanation for your initial answer to the question. Then, verify both the factual accuracy of your explanation and the logic behind your answer. Finally, give a concise response to the question by answering with 'YES' or 'NO'.
Modality and emotion knowledge: {knowledge}
Question: {question}
Initial Answer: {initial_answer}
Answer Format:
1. [Explanation]
2. [Verification]
3. [Final Answer]

Figure 9: Prompt for modality and emotion knowledge extraction.

(i.e., PEP-MEK) represents making a second prediction based on explanations generated from all inputs. Incorporating MEK consistently improves performance across all models, indicating that grounding predictions with structured emotional knowledge helps suppress spurious responses. Adding the explanation component yields further gains, highlighting the role of explicit reasoning in enhancing prediction faithfulness. Together, these results validate the effectiveness of both components and show that combining MEK with explanation-based reasoning forms a complementary strategy to mitigate emotion hallucinations.

Table 5: Ablation study on EmotionHallucer-P showing the effect of different components in PEP-MEK.

| Methods | Input | Yes/No Bias | | Accuracy on EmotionHallucer-P | | |
| --- | --- | --- | --- | --- | --- | --- |
| | | Pct. Diff (∼0) | FP Ratio (∼0.5) | Basic ↑ | Hallucinated ↑ | Overall ↑ |
| Qwen2.5-Omni | Original input | -0.18 | 0.30 | 35.51 | 71.96 | 10.49 |
| | + MEK | -0.26 | 0.20 | 29.39 | 81.93 | 15.58 |
| (Xu et al., 2025) | + MEK + Explain | -0.19 | 0.29 | 37.49 | 74.87 | 20.15 |
| Emotion-LLaMA | Original input | 0.28 | 0.77 | 77.34 | 22.02 | 9.65 |
| | + MEK | -0.14 | 0.36 | 38.64 | 65.87 | 19.12 |
| (Cheng et al., 2024) | + MEK + Explain | -0.08 | 0.41 | 46.35 | 62.86 | 26.03 |
| Gemini-2.5-Flash | Original input | 0.00 | 0.50 | 61.54 | 62.05 | 33.44 |
| | + MEK | -0.08 | 0.39 | 54.95 | 71.37 | 35.84 |
| (Google DeepMind, 2025) | + MEK + Explain | -0.07 | 0.40 | 56.66 | 71.17 | 37.84 |

### B.2.2 EMOTION KNOWLEDGE IN PEP-MEK

We further conduct an ablation study to evaluate the role of prompt design in extracting Modality and Emotion Knowledge (MEK). In particular, we replace the MEK prompt with a generic modality knowledge (MK) prompt that removes emotion-specific guidance. As shown in Table 6, this sub-

stitution consistently reduces performance across all models. These results indicate that emotion-grounded prompts are essential for capturing emotion-relevant information, which in turn is critical for detecting emotion-related hallucinations.

Table 6: Ablation study on EmotionHallucer-P evaluating the role of emotion knowledge in PEP-MEK.

| Methods | Input | Yes/No Bias | | Accuracy on EmotionHallucer-P | | |
| --- | --- | --- | --- | --- | --- | --- |
| | | Pct. Diff (∼0) | FP Ratio (∼0.5) | Basic ↑ | Hallucinated ↑ | Overall ↑ |
| Qwen2.5-Omni | Original input | -0.18 | 0.30 | 35.51 | 71.96 | 10.49 |
| | + PEP-MK | -0.23 | 0.24 | 31.93 | 78.38 | 13.21 |
| (Xu et al., 2025) | + PEP-MEK | -0.19 | 0.29 | 37.49 | 74.87 | 20.15 |
| Emotion-LLaMA | Original input | 0.28 | 0.77 | 77.34 | 22.02 | 9.65 |
| | + PEP-MK | -0.07 | 0.43 | 42.66 | 56.95 | 16.80 |
| (Cheng et al., 2024) | + PEP-MEK | -0.08 | 0.41 | 46.35 | 62.86 | 26.03 |
| Gemini-2.5-Flash | Original input | 0.00 | 0.50 | 61.54 | 62.05 | 33.44 |
| | + PEP-MK | -0.09 | 0.38 | 51.26 | 69.75 | 30.55 |
| (Google DeepMind, 2025) | + PEP-MEK | -0.07 | 0.40 | 56.66 | 71.17 | 37.84 |

### B.2.3 WALL-CLOCK LATENCY AND TOKEN-COST

To assess the computational overhead introduced by the PEP-MEK framework, we measure both token cost and wall-clock latency for each component (Baseline, Description, PEP-MEK-first, and PEP-MEK-explain&second) across all five modalities, as these metrics are fundamentally modality-dependent. The results are summarized in Table 7. These measurements were obtained using the Qwen2.5-Omni-7B model via the Aliyun Bailian API. Token cost was collected by enabling `stream_options={''include_usage'': True}`, while wall-clock time was measured as the elapsed time from sending the request to receiving the final streamed chunk. We note that wall-clock latency also reflects network delay and API scheduling overhead, and thus should be interpreted as an approximation of user-perceived latency rather than pure computational time.

The PEP-MEK components introduce moderate computational overhead but yield consistent performance improvements across modalities. Both the first stage and the second stage show stable accuracy gains relative to the baseline. The benefit is particularly pronounced in long-video scenarios, where explanation-guided reasoning (PEP-MEK-explain&second) outperforms the Description stage, indicating that structured multi-stage reasoning becomes increasingly important as emotional cues unfold over extended temporal sequences. Furthermore, comparisons in Figure 9 show that PEP-MEK achieves stronger performance than alternative strategies operating under similar computational budgets, demonstrating the effectiveness of investing computation toward emotion-specific reasoning rather than naive prompting. We emphasize that for safety-sensitive applications involving emotional interpretation, robustness is often more critical than inference speed. The modest latency increase of PEP-MEK therefore represents a meaningful and practical trade-off.

### B.2.4 PERFORMANCE ACROSS SUBCATEGORIES

Beyond modality-level efficiency, we further analyze the impact of PEP-MEK on the four hallucination subcategories, as shown in Table 8. We observe stable and consistent improvements for both *+MEK* and *+MEK + Explain* across all evaluated models. These gains appear consistently across subcategories, indicating that PEP-MEK enhances not only basic perceptual grounding, but also higher-level emotional reasoning. Importantly, the improvements are not confined to a single failure type but generalize across diverse hallucination categories, demonstrating that PEP-MEK mitigates hallucinations in a broadly applicable manner rather than overfitting to specific cases.

### B.2.5 COMPARATION TO GENERAL METHODS

To further validate the effectiveness of PEP-MEK, we compare Qwen2.5-Omni (Xu et al., 2025) with several widely used hallucination mitigation strategies. For CoT (Wei et al., 2022), we apply the prompt "Let's think step by step" to guide reasoning. In majority voting (Wang et al., 2022),

Table 7: Wall-clock latency and token-cost comparison for each PEP-MEK component across modalities. Each cell reports Overall Accuracy / Token Cost / Latency (s).

| Modality | Baseline | Description | PEP-MEK-first | PEP-MEK-explain&second |
|---|---|---|---|---|
| Text | 19.00 / 274 / 0.86 | – / 1357 / 10.09 | 31.00 / 1357 / 10.09 | 36.00 / 1314 / 3.90 |
| Image | 32.67 / 462 / 2.47 | – / 1499 / 11.33 | 37.33 / 1193 / 8.69 | 40.00 / 1403 / 8.80 |
| Audio | 0.82 / 252 / 57.48 | – / 1103 / 74.54 | 2.99 / 769 / 49.68 | 5.43 / 995 / 20.27 |
| Video-S | 13.19 / 2717 / 35.09 | – / 3584 / 23.65 | 27.08 / 3307 / 2.67 | 29.17 / 3533 / 3.91 |
| Video-L | 5.47 / 18389 / 191.43 | – / 19878 / 157.47 | 6.47 / 19591 / 45.46 | 17.91 / 19812 / 22.71 |

Table 8: Overall accuracy comparison of PEP-MEK components across the four hallucination sub-categories: Category, Intensity, Reasoning-Result, and Reasoning-Cue.

| Model | Original Input | +MEK | +MEK + Explain |
|---|---|---|---|
| Qwen2.5-Omni | 14.37/2.67/18.92/6.04 | 18.86/6.87/25.95/11.54 | 21.56/9.92/30.27/21.98 |
| Emotion-LLaMA | 11.08/4.58/12.81/5.03 | 18.26/15.27/30.05/14.57 | 25.45/22.90/33.50/23.54 |
| Gemini-2.5-Flash | 41.32/27.86/39.38/20.97 | 42.51/32.06/40.32/26.37 | 43.71/37.02/40.82/27.86 |

we sample three outputs (temperature = 1.2, top_p = 1.0) to match PEP-MEK's inference budget. Combining CoT with majority voting yields the self-consistency variant (Wang et al., 2022). For RAG (Lewis et al., 2020), because our questions are not directly suited for conventional retrieval, we design a psychology-grounded variant: the model generates a query to an emotion psychology textbook, retrieves a relevant answer, and then synthesizes its prediction.

As shown in Table 9, PEP-MEK consistently outperforms all alternatives. While these general strategies mitigate hallucinations to some extent, their lack of emotion-specific grounding limits effectiveness in our benchmark. We also observe that psychology-based RAG often biases retrieval toward facial-expression–related cues, overlooking other emotional signals. Importantly, these strategies are not mutually exclusive with PEP-MEK; integrating structured emotion reasoning with retrieval, voting, or self-consistency offers a promising direction for future research.

Table 9: Performance comparison of Qwen2.5-Omni with different reasoning and augmentation strategies on EmotionHallucer-P.

| Methods | Yes/No Bias | | Accuracy on EmotionHallucer-P | | |
|---|---|---|---|---|---|
| | Pct. Diff (∼0) | FP Ratio (∼0.5) | Basic ↑ | Hallucinated ↑ | Overall ↑ |
| Qwen2.5-Omni (Xu et al., 2025) | -0.18 | 0.30 | 35.51 | 71.96 | 10.49 |
| + CoT (Wei et al., 2022) | -0.23 | 0.24 | 31.58 | 77.94 | 14.79 |
| + Majority Voting (Wang et al., 2022) | -0.11 | 0.38 | 42.73 | 64.29 | 10.78 |
| + Self-consistency (Wang et al., 2022) | -0.29 | 0.17 | 27.35 | 85.60 | 17.31 |
| + RAG (Lewis et al., 2020) | -0.13 | 0.36 | 41.60 | 66.92 | 13.40 |
| + PEP-MEK | -0.19 | 0.29 | 37.49 | 74.87 | 20.15 |

## B.3 QUALITATIVE ANALYSIS OF PEP-MEK

As shown in Figure 10, we present an example that benefits from the PEP-MEK approach, where the generated content originates from the Qwen2.5-Omni model. It is evident that in the initial response, the model exhibited a hallucination issue, failing to accurately interpret the facial expressions of individuals in the image. However, with the integration of PEP-MEK, the model was guided to attend to more discriminative visual details, for instance, recognizing that most individuals are smiling, except for the person wearing a red tie, who has a neutral or serious expression. This targeted guidance significantly improved the model's emotion understanding of the image. Ultimately, the model identified the key feature and produced the correct answer, demonstrating the effectiveness of PEP-MEK in mitigating hallucinations and enhancing vision and emotion reasoning accuracy.

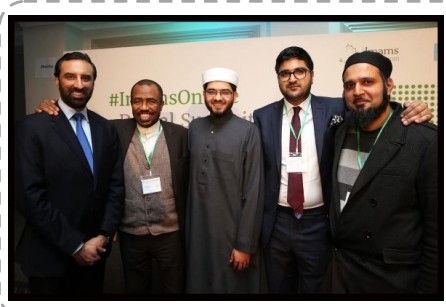

**Hallucinated:** The person with the red tie has a happy expression.
**Initial Answer:** YES ❌
**PEP-MEK:**
Explanation: The description states that the person with the red tie has a happy expression. In the provided image, most individuals are smiling, which aligns with the concept of happiness. However, the description specifically mentions only one individual (the person with the red tie), and his expression seems natural. Therefore, the description does not fully capture the collective emotion of the entire group but focuses on a single person.
Verification: 1. Emotion in Image: Most individuals in the image are smiling, indicating happiness. 2. Focus of Description: The description focuses on a specific individual (the person with the red tie) rather than the entire group. 3. Accuracy: The description only mentions one person out of several, rather than describing the expressions of the majority of the group.
Final Answer: NO ✅

Figure 10: Qualitative analysis of PEP-MEK. In the observed image, everyone except the person wearing the red tie is smiling, displaying a happy expression. PEP-MEK noticed this detail during the explanation and ultimately provided the correct answer.

## C    MORE EXPERIMENT DETAILS

### C.1    SETUPS FOR BASELINES

In our experiment, we selected several well-known LLMs and MLLMs for comparison, as shown in Table 10. We provide a list of model names, model sizes, supported input and output modalities, all organized by release date to enable a more detailed analysis. Additionally, we annotate whether each model is a reasoning model like DeepSeek-R1 (Guo et al., 2025). To ensure a fair comparison, we adopted the default hyper-parameters of these models. All models with fewer than 235B parameters were run locally on either a single NVIDIA A100 or four NVIDIA A100 GPUs. The remaining models were accessed via APIs provided by their developers.

### C.2    RESULT OF UNIMODALITY

We present a more fine-grained evaluation and analysis of the model's performance for each individual modality.

#### C.2.1    EMOTION KNOWLEDGE

As shown in Table 11, we report the performance of various models on the EmotionHallucer-EK, which focuses on emotion knowledge hallucination. These results are further visualized in Figure 11. From the visualization, several key trends emerge: (1) model performance has steadily improved over time; (2) closed-source models generally outperform open-source models; (3) the incorporation of reasoning capabilities in open-source models significantly narrows the gap with closed-source models; (4) MLLMs generally perform worse than pure LLMs on this task; and (5) within the same model family, larger parameter sizes tend to correlate with better performance. Overall, we observe a clear upward trend in model performance on emotion knowledge hallucination tasks, with many models achieving relatively high accuracy. This highlights the growing potential of leveraging LLMs for applications in emotion psychology, and points to future directions for further enhancing open models, particularly in multimodal emotion reasoning.

#### C.2.2    MULTIMODALITY PERCEPTION: TEXT

As shown in Table 12, we report the performance of various models on the EmotionHallucer-PT, which focuses on hallucinations in real-world emotion understanding from text. The results are further visualized in Figure 12. From the figure, several key observations emerge: (1) model performance has gradually improved over time; (2) unlike in the knowledge setting, the performance gap between closed-source and open-source models is relatively small; (3) the incorporation of reasoning capabilities into open-source models does not lead to better performance; (4) within the same model family, larger parameter sizes do not necessarily correlate with higher accuracy; (5) although both PT and EK benchmarks are based on text, models perform significantly worse on PT, suggesting that LLMs are more adept at retrieving structured emotional knowledge than interpreting nuanced

Table 10: Comparison of representative MLLMs. For closed-source models, we report the API version date. I, V, A, and T represent Image, Video, Audio, and Text, respectively.

| Model | Date | Model Size (B) | Input | Output | Reasoning |
|---|---|---|---|---|---|
| *open-source* | | | | | |
| LLaVA (Liu et al., 2023) | 04.2023 | 7&13&34 | I + T | T | No |
| Video-ChatGPT (Maaz et al., 2023) | 06.2023 | 7 | V + T | T | No |
| Mistral (Jiang et al., 2023) | 08.2023 | 7 | T | T | No |
| Qwen (Bai et al., 2023) | 09.2023 | 7&14 | T | T | No |
| Chat-UniVi (Jin et al., 2024) | 11.2023 | 7 | I + V + T | T | No |
| LLaMA-VID (Li et al., 2024c) | 11.2023 | 7 | I + V + T | T | No |
| Video-LLaVA (Lin et al., 2023) | 11.2023 | 7 | I + V + T | T | No |
| Onellm (Han et al., 2024) | 12.2023 | 7 | I + A + V + T | T | No |
| Mixtral (Jiang et al., 2024) | 01.2024 | 8×7 | T | T | No |
| Llama3 (Grattafiori et al., 2024) | 04.2024 | 8 | T | T | No |
| Emotion-LLaMA (Cheng et al., 2024) | 06.2024 | 7 | V + T | T | No |
| Llama3.1 (Grattafiori et al., 2024) | 07.2024 | 8&70 | T | T | No |
| Qwen2 (Yang et al., 2024a) | 07.2024 | 7 | Text | T | No |
| Llama3.2 (Grattafiori et al., 2024) | 09.2024 | 3 | T | T | No |
| Llama3.2-vision (Grattafiori et al., 2024) | 09.2024 | 11 | I + T | T | No |
| Llama3.3 (Grattafiori et al., 2024) | 12.2024 | 70 | T | T | No |
| Phi4 (Abdin et al., 2024) | 12.2024 | 14 | T | T | No |
| Qwen2.5 (Yang et al., 2024b) | 12.2024 | 3&7&14&32&72 | T | T | No |
| DeepSeek-V3 (Liu et al., 2024a) | 12.2024 | - | T | T | No |
| DeepSeek-R1 (Guo et al., 2025) | 01.2025 | 7&8&14&32&70&671 | T | T | Yes |
| Qwen2.5-VL (Bai et al., 2025) | 02.2025 | 32&72 | I + V + T | T | Yes |
| QwQ (Team, 2025) | 03.2025 | 32 | T | T | Yes |
| Gemma3 (Team et al., 2025) | 03.2025 | 4&12&27 | I + T | T | No |
| Mistral-small3.1 (Mistral AI, 2024) | 03.2025 | 24 | I + T | T | No |
| Qwen2.5-Omni (Xu et al., 2025) | 03.2025 | 7 | I+ V+ A + T | A + T | No |
| Qwen3 (Qwen Team, 2024) | 04.2025 | 4&8&14&30&32&235 | T | T | Yes |
| Kimi-Audio (Ding et al., 2025) | 04.2025 | 7 | A + T | A + T | No |
| *closed-source* | | | | | |
| GPT-4o (Hurst et al., 2024) | 08.2024 | - | I + T | T | No |
| Qwen-Audio-Turbo (Chu et al., 2024) | 12.2024 | - | A | T | No |
| Qwen-Plus (Yang et al., 2024b) | 01.2025 | - | T | T | No |
| Qwen-Max (Yang et al., 2024b) | 01.2025 | - | T | T | No |
| Qwen-VL-Plus (Bai et al., 2025) | 01.2025 | - | T+I+V | T | No |
| QwQ-Plus (Team, 2025) | 03.2025 | - | T | T | Yes |
| QvQ-Max (Qwen Team, 2025) | 03.2025 | - | I + T | T | Yes |
| Gemini-2.5-Pro (Google DeepMind, 2025) | 03.2025 | - | I+ V+ A + T | T | Yes |
| Qwen-VL-Max (Bai et al., 2025) | 04.2025 | - | T+I+V | T | No |
| Gemini-2.5-Flash (Google DeepMind, 2025) | 04.2025 | - | I+ V+ A + T | T | Yes |
| GPT-4.1 (OpenAI, 2024) | 04.2025 | - | I + T | T | No |
| GPT-OSS (OpenAI, 2025a) | 08.2025 | - | T | T | No |
| GPT-5 (OpenAI, 2025b) | 08.2025 | - | I + T | T | Yes |
| Qwen3-Next (Qwen, 2025) | 09.2025 | - | T | T | Yes |

emotions in real-world language; and (6) PEP-MEK demonstrates its effectiveness, often achieving performance that matches or exceeds the best-performing models of the same period.

We hypothesize that this performance gap stems from two primary factors: (1) current LLMs struggle to detect subtle emotional shifts and cues in naturalistic language, making them prone to hallucination in text emotion understanding tasks; and (2) existing models may lack sufficient exposure to pretraining tasks that require fine-grained emotion reasoning in everyday contexts. These findings highlight a critical challenge for future work: enhancing MLLMs' capacity to understand subtle emotional shifts in textual content.

### C.2.3 MULTIMODALITY PERCEPTION: IMAGE

As shown in Table 13, we report the performance of various models on the EmotionHallucer-PI, which focuses on hallucinations in real-world emotion understanding from images. The results

Table 11: Performance comparison on EmotionHallucer-EK (Emotion Knowledge).

| Methods | Model Size | Yes/No Bias | | Accuracy on EmotionHallucer-EK | | |
| --- | --- | --- | --- | --- | --- | --- |
| | | Pct. Diff (∼0) | FP Ratio (∼0.5) | Basic ↑ | Hallucinated ↑ | Overall ↑ |
| *Open-source* | | | | | | |
| Mistral (Jiang et al., 2023) | 7B | 0.26 | 0.89 | 92.47 | 39.52 | 34.68 |
| Qwen (Bai et al., 2023) | 7B | 0.21 | 0.81 | 87.63 | 45.70 | 37.37 |
| Qwen (Bai et al., 2023) | 14B | 0.11 | 0.69 | 82.26 | 60.48 | 46.51 |
| Mixtral (Jiang et al., 2024) | 8x7B | 0.05 | 0.42 | 62.10 | 72.04 | 41.13 |
| Llama3 (Grattafiori et al., 2024) | 8B | 0.40 | 0.98 | 98.39 | 18.55 | 17.47 |
| Llama3.1 (Grattafiori et al., 2024) | 8B | 0.26 | 0.90 | 93.28 | 41.14 | 37.37 |
| Llama3.1 (Grattafiori et al., 2024) | 70B | 0.22 | 0.94 | 96.77 | 51.88 | 49.73 |
| Qwen2 (Yang et al., 2024a) | 7B | 0.18 | 0.86 | 93.01 | 56.18 | 50.54 |
| Llama3.2 (Grattafiori et al., 2024) | 3B | 0.19 | 0.78 | 84.95 | 46.24 | 34.68 |
| Llama3.3 (Grattafiori et al., 2024) | 70B | 0.20 | 0.92 | 95.97 | 56.18 | 53.76 |
| Phi4 (Abdin et al., 2024) | 14B | 0.12 | 0.79 | 91.40 | 66.67 | 60.22 |
| Qwen2.5 (Yang et al., 2024b) | 3B | -0.13 | 0.27 | 58.87 | 84.95 | 45.70 |
| Qwen2.5 (Yang et al., 2024b) | 7B | 0.17 | 0.82 | 90.32 | 57.26 | 50.54 |
| Qwen2.5 (Yang et al., 2024b) | 14B | -0.01 | 0.46 | 80.91 | 83.60 | 66.40 |
| Qwen2.5 (Yang et al., 2024b) | 32B | 0.09 | 0.75 | 91.13 | 73.92 | 67.47 |
| Qwen2.5 (Yang et al., 2024b) | 72B | 0.11 | 0.83 | 94.35 | 72.31 | 68.82 |
| DeepSeek-V3 (Liu et al., 2024a) | 671B | 0.10 | 0.76 | 90.59 | 69.89 | 62.10 |
| DeepSeek-R1 (Guo et al., 2025) | 7B | 0.19 | 0.80 | 87.90 | 50.27 | 43.55 |
| DeepSeek-R1 (Guo et al., 2025) | 8B | 0.15 | 0.78 | 87.63 | 56.99 | 48.92 |
| DeepSeek-R1 (Guo et al., 2025) | 14B | 0.08 | 0.71 | 88.17 | 71.24 | 62.63 |
| DeepSeek-R1 (Guo et al., 2025) | 32B | 0.08 | 0.75 | 91.67 | 75.00 | 68.82 |
| DeepSeek-R1 (Guo et al., 2025) | 70B | 0.09 | 0.76 | 91.40 | 72.58 | 65.86 |
| DeepSeek-R1 (Guo et al., 2025) | 671B | 0.08 | 0.78 | 93.82 | 77.96 | 73.12 |
| QwQ (Team, 2025) | 32B | 0.07 | 0.75 | 92.74 | 73.66 | 73.39 |
| Qwen3 (Qwen Team, 2024) | 4B | 0.05 | 0.63 | 86.02 | 76.61 | 65.32 |
| Qwen3 (Qwen Team, 2024) | 8B | 0.04 | 0.63 | 86.56 | 78.23 | 67.20 |
| Qwen3 (Qwen Team, 2024) | 14B | 0.04 | 0.61 | 86.83 | 70.30 | 68.82 |
| Qwen3 (Qwen Team, 2024) | 30B | 0.03 | 0.60 | 88.17 | 81.99 | 72.58 |
| Qwen3 (Qwen Team, 2024) | 32B | 0.06 | 0.73 | 93.01 | 80.91 | 76.08 |
| Qwen3 (Qwen Team, 2024) | 235B | 0.05 | 0.67 | 91.13 | 81.72 | 74.18 |
| GPT-OSS (OpenAI, 2025a) | 20B | 0.04 | 0.62 | 87.63 | 79.84 | 70.97 |
| GPT-OSS (OpenAI, 2025a) | 120B | 0.04 | 0.66 | 91.40 | 83.06 | 76.34 |
| Qwen3-Next-Instruct (Qwen, 2025) | 80B | 0.02 | 0.55 | 85.22 | 81.99 | 68.55 |
| Qwen3-Next-Thinking (Qwen, 2025) | 80B | -0.04 | 0.36 | 82.26 | 90.05 | 73.66 |
| LLaVA (Liu et al., 2023) | 7B | 0.41 | 0.97 | 97.85 | 16.40 | 15.59 |
| LLaVA (Liu et al., 2023) | 13B | 0.23 | 0.85 | 90.32 | 43.82 | 36.29 |
| LLaVA (Liu et al., 2023) | 34B | 0.43 | 1.00 | 99.73 | 13.71 | 13.71 |
| Video-ChatGPT (Maaz et al., 2023) | 7B | 0.26 | 0.83 | 86.56 | 34.95 | 25.27 |
| Chat-UniVi (Jin et al., 2024) | 7B | 0.50 | 1.00 | 100.00 | 0.00 | 0.00 |
| LLaMA-VID (Li et al., 2024c) | 7B | 0.50 | 1.00 | 100.00 | 0.00 | 0.00 |
| Video-LLaVA (Lin et al., 2023) | 7B | 0.50 | 1.00 | 100.00 | 0.00 | 0.00 |
| Onellm (Han et al., 2024) | 7B | -0.01 | 0.48 | 66.40 | 68.82 | 40.05 |
| Emotion-LLaMA (Cheng et al., 2024) | 7B | -0.04 | 0.45 | 59.68 | 67.20 | 32.53 |
| Mistral-small3.1 (Mistral AI, 2024) | 24B | 0.12 | 0.77 | 90.05 | 66.94 | 59.14 |
| Llama3.2-vision (Grattafiori et al., 2024) | 11B | 0.24 | 0.88 | 92.20 | 45.16 | 40.32 |
| Qwen2.5-VL (Bai et al., 2025) | 32B | 0.25 | 0.97 | 98.39 | 48.39 | 47.85 |
| Qwen2.5-VL (Bai et al., 2025) | 72B | 0.09 | 0.79 | 93.55 | 75.54 | 70.43 |
| Gemma3 (Team et al., 2025) | 4B | 0.28 | 0.90 | 93.01 | 36.56 | 32.26 |
| Gemma3 (Team et al., 2025) | 12B | 0.19 | 0.85 | 91.94 | 54.30 | 49.19 |
| Gemma3 (Team et al., 2025) | 27B | 0.20 | 0.92 | 96.24 | 55.38 | 52.69 |
| Qwen2.5-Omni (Xu et al., 2025) | 7B | 0.28 | 0.96 | 97.58 | 41.40 | 39.78 |
| *Closed-source* | | | | | | |
| GPT-4o (Hurst et al., 2024) | - | 0.10 | 0.78 | 92.74 | 73.66 | 68.82 |
| Qwen-Plus (Yang et al., 2024b) | - | 0.17 | 0.89 | 94.89 | 60.48 | 58.06 |
| Qwen-Max (Yang et al., 2024b) | - | 0.14 | 0.84 | 93.28 | 66.59 | 61.02 |
| QwQ-Plus (Team, 2025) | - | 0.07 | 0.76 | 93.55 | 79.30 | 75.81 |
| QvQ-Max (Qwen Team, 2025) | - | 0.05 | 0.67 | 90.59 | 81.18 | 74.19 |
| Gemini-2.5-Pro (Google DeepMind, 2025) | - | 0.02 | 0.58 | 88.98 | 84.68 | 75.54 |
| Gemini-2.5-Flash (Google DeepMind, 2025) | - | 0.03 | 0.61 | 90.05 | 84.14 | 75.54 |
| GPT-4.1 (OpenAI, 2024) | - | 0.08 | 0.81 | 94.62 | 77.69 | 74.19 |
| GPT-5 (OpenAI, 2025b) | - | 0.02 | 0.59 | 91.94 | 88.17 | 81.45 |

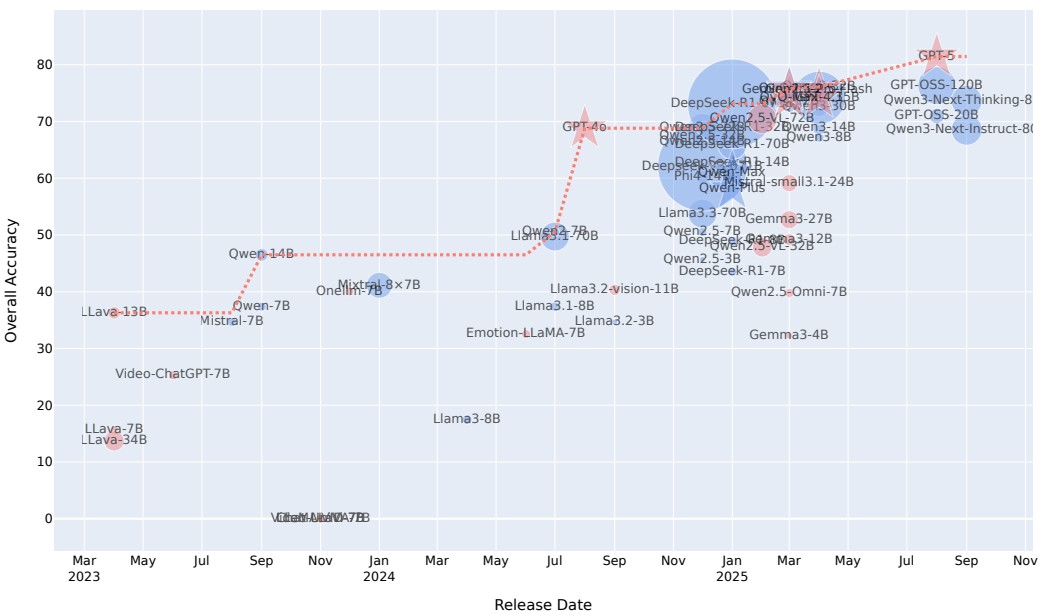

Figure 11: Performance comparison on EmotionHallucer-EK. Blue denotes LLMs, red denotes MLLMs. Circles indicate known parameter sizes, while stars represent unknown parameter sizes. Red dashed line denotes the top-performing model of the current month.

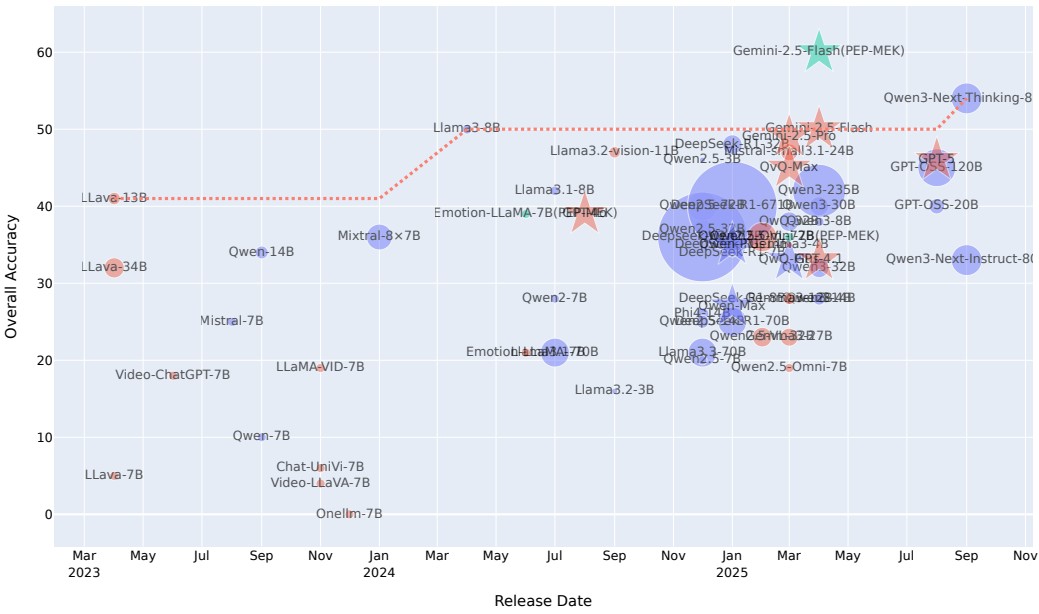

Figure 12: Performance comparison on EmotionHallucer-PT. Blue denotes LLMs, red denotes MLLMs. Circles indicate known parameter sizes, while stars represent unknown parameter sizes. Red dashed line denotes the top-performing model of the current month.

Table 12: Performance comparison on EmotionHallucer-PT (Perception Text).

| Methods | Model Size | Yes/No Bias | | Accuracy on EmotionHallucer-PT | | |
|---|---|---|---|---|---|---|
| | | Pct. Diff ($\sim$0) | FP Ratio ($\sim$0.5) | Basic ↑ | Hallucinated ↑ | Overall ↑ |
| *Open-source* | | | | | | |
| Mixtral (Jiang et al., 2024) | 7B | 0.37 | 0.99 | 99.00 | 25.00 | 25.00 |
| Qwen (Bai et al., 2023) | 7B | 0.12 | 0.60 | 53.00 | 29.00 | 10.00 |
| Qwen (Bai et al., 2023) | 14B | 0.07 | 0.41 | 53.00 | 67.00 | 34.00 |
| Mixtral (Jiang et al., 2024) | 8x7B | 0.14 | 0.33 | 46.00 | 73.00 | 36.00 |
| Llama3 (Grattafiori et al., 2024) | 8B | 0.04 | 0.58 | 75.00 | 66.00 | 50.00 |
| Llama3.1 (Grattafiori et al., 2024) | 8B | 0.11 | 0.65 | 75.00 | 53.00 | 42.00 |
| Llama3.1 (Grattafiori et al., 2024) | 70B | 0.40 | 1.00 | 100.00 | 21.00 | 21.00 |
| Qwen2 (Yang et al., 2024a) | 7B | 0.34 | 0.95 | 96.00 | 28.00 | 28.00 |
| Llama3.2 (Grattafiori et al., 2024) | 3B | 0.39 | 0.06 | 17.00 | 95.00 | 16.00 |
| Llama3.3 (Grattafiori et al., 2024) | 70B | 0.40 | 1.00 | 100.00 | 21.00 | 21.00 |
| Phi4 (Abdin et al., 2024) | 14B | 0.36 | 0.99 | 99.00 | 26.00 | 26.00 |
| Qwen2.5 (Yang et al., 2024b) | 3B | 0.19 | 0.19 | 50.00 | 88.00 | 46.00 |
| Qwen2.5 (Yang et al., 2024b) | 7B | 0.39 | 0.98 | 98.00 | 20.00 | 20.00 |
| Qwen2.5 (Yang et al., 2024b) | 14B | 0.38 | 1.00 | 100.00 | 25.00 | 25.00 |
| Qwen2.5 (Yang et al., 2024b) | 32B | 0.29 | 0.94 | 96.00 | 38.00 | 37.00 |
| Qwen2.5 (Yang et al., 2024b) | 72B | 0.30 | 1.00 | 100.00 | 40.00 | 40.00 |
| DeepSeek-V3 (Liu et al., 2024a) | -B | 0.28 | 0.93 | 95.00 | 38.00 | 36.00 |
| DeepSeek-R1 (Guo et al., 2025) | 7B | 0.30 | 0.93 | 95.00 | 35.00 | 34.00 |
| DeepSeek-R1 (Guo et al., 2025) | 8B | 0.34 | 0.97 | 98.00 | 29.00 | 28.00 |
| DeepSeek-R1 (Guo et al., 2025) | 14B | 0.32 | 0.98 | 99.00 | 35.00 | 35.00 |
| DeepSeek-R1 (Guo et al., 2025) | 32B | 0.23 | 0.93 | 96.00 | 49.00 | 48.00 |
| DeepSeek-R1 (Guo et al., 2025) | 70B | 0.37 | 0.99 | 99.00 | 25.00 | 25.00 |
| DeepSeek-R1 (Guo et al., 2025) | 671B | 0.28 | 0.94 | 96.00 | 40.00 | 40.00 |
| QwQ (Team, 2025) | 32B | 0.31 | 1.00 | 100.00 | 38.00 | 38.00 |
| Qwen3 (Qwen Team, 2024) | 4B | 0.35 | 0.99 | 99.00 | 28.00 | 28.00 |
| Qwen3 (Qwen Team, 2024) | 8B | 0.29 | 0.94 | 99.00 | 38.00 | 38.00 |
| Qwen3 (Qwen Team, 2024) | 14B | 0.33 | 0.95 | 96.00 | 30.00 | 28.00 |
| Qwen3 (Qwen Team, 2024) | 30B | 0.26 | 0.89 | 93.00 | 41.00 | 40.00 |
| Qwen3 (Qwen Team, 2024) | 32B | 0.33 | 0.97 | 98.00 | 32.00 | 32.00 |
| Qwen3 (Qwen Team, 2024) | 235B | 0.23 | 0.89 | 93.00 | 45.00 | 42.00 |
| GPT-OSS (OpenAI, 2025a) | 20B | 0.26 | 0.89 | 93.00 | 41.00 | 40.00 |
| GPT-OSS (OpenAI, 2025a) | 120B | 0.23 | 0.86 | 91.00 | 46.00 | 45.00 |
| Qwen3-Next-Instruct (Qwen, 2025) | 80B | 0.29 | 0.92 | 94.00 | 35.00 | 33.00 |
| Qwen3-Next-Thinking (Qwen, 2025) | 80B | 0.10 | 0.68 | 82.00 | 62.00 | 54.00 |
| LLaVA (Liu et al., 2023) | 7B | 0.47 | 1.00 | 100.00 | 5.00 | 5.00 |
| LLaVA (Liu et al., 2023) | 13B | 0.21 | 0.18 | 46.00 | 88.00 | 41.00 |
| LLaVA (Liu et al., 2023) | 34B | 0.32 | 0.96 | 97.00 | 33.00 | 32.00 |
| Video-ChatGPT (Maaz et al., 2023) | 7B | 0.32 | 0.17 | 19.00 | 83.00 | 18.00 |
| Chat-UniVi (Jin et al., 2024) | 7B | 0.47 | 0.01 | 6.00 | 99.00 | 6.00 |
| LLaMA-VID (Li et al., 2024c) | 7B | 0.20 | 0.68 | 64.00 | 25.00 | 19.00 |
| Video-LLaVA (Lin et al., 2023) | 7B | 0.47 | 0.98 | 98.00 | 4.00 | 4.00 |
| Onellm (Han et al., 2024) | 7B | 0.50 | 1.00 | 100.00 | 0.00 | 0.00 |
| Emotion-LLaMA (Cheng et al., 2024) | 7B | 0.28 | 0.22 | 21.00 | 78.00 | 21.00 |
| +PEP-MEK | | 0.14 | 0.32 | 46.00 | 75.00 | 39.00 |
| Mistral-small3.1 (Mistral AI, 2024) | 24B | 0.20 | 0.84 | 91.00 | 51.00 | 47.00 |
| Llama3.2-vision (Grattafiori et al., 2024) | 11B | 0.02 | 0.53 | 69.00 | 65.00 | 47.00 |
| Qwen2.5-VL (Bai et al., 2025) | 32B | 0.39 | 1.00 | 100.00 | 23.00 | 23.00 |
| Qwen2.5-VL (Bai et al., 2025) | 72B | 0.32 | 0.98 | 99.00 | 36.00 | 36.00 |
| Gemma3 (Team et al., 2025) | 4B | 0.29 | 0.91 | 94.00 | 36.00 | 35.00 |
| Gemma3 (Team et al., 2025) | 12B | 0.35 | 0.97 | 98.00 | 28.00 | 28.00 |
| Gemma3 (Team et al., 2025) | 27B | 0.39 | 1.00 | 100.00 | 23.00 | 23.00 |
| Qwen2.5-Omni (Xu et al., 2025) | 7B | 0.41 | 1.00 | 100.00 | 19.00 | 19.00 |
| +PEP-MEK | | 0.28 | 0.91 | 94.00 | 38.00 | 36.00 |
| *Closed-source* | | | | | | |
| GPT-4o (Hurst et al., 2024) | - | 0.28 | 0.94 | 96.00 | 39.00 | 39.00 |
| Qwen-Plus (Yang et al., 2024b) | - | 0.31 | 0.96 | 97.00 | 35.00 | 35.00 |
| Qwen-Max (Yang et al., 2024b) | - | 0.36 | 0.99 | 99.00 | 27.00 | 27.00 |
| QwQ-Plus (Team, 2025) | - | 0.33 | 0.99 | 99.00 | 33.00 | 33.00 |
| QvQ-Max (Qwen Team, 2025) | - | 0.24 | 0.93 | 96.00 | 47.00 | 45.00 |
| Gemini-2.5-Pro (Google DeepMind, 2025) | - | 0.23 | 0.92 | 96.00 | 51.00 | 49.00 |
| Gemini-2.5-Flash (Google DeepMind, 2025) | - | 0.21 | 0.89 | 94.00 | 51.00 | 50.00 |
| +PEP-MEK | | 0.12 | 0.74 | 88.00 | 54.00 | 60.00 |
| GPT-4.1 (OpenAI, 2024) | - | 0.30 | 0.92 | 94.00 | 34.00 | 33.00 |
| GPT-5 (OpenAI, 2025b) | - | 0.18 | 0.80 | 88.00 | 51.00 | 46.00 |

Table 13: Performance comparison on EmotionHallucer-PI (Perception Image).

| Methods | Model Size | Yes/No Bias | | Accuracy on Perception-I | | |
|---|---|---|---|---|---|---|
| | | Pct. Diff (∼0) | FP Ratio (∼0.5) | Basic ↑ | Hallucinated ↑ | Overall ↑ |
| *Open-source* | | | | | | |
| LLaVA (Liu et al., 2023) | 7B | 0.50 | 1.00 | 100.00 | 0.00 | 0.00 |
| LLaVA (Liu et al., 2023) | 13B | 0.06 | 0.44 | 47.33 | 58.67 | 22.00 |
| LLaVA (Liu et al., 2023) | 34B | 0.46 | 0.01 | 6.67 | 99.33 | 6.00 |
| Video-ChatGPT (Maaz et al., 2023) | 7B | 0.04 | 0.54 | 53.33 | 45.33 | 14.00 |
| Chat-UniVi (Jin et al., 2024) | 7B | -0.31 | 0.18 | 19.33 | 82.00 | 14.00 |
| LLaMA-VID (Li et al., 2024c) | 7B | -0.21 | 0.28 | 32.67 | 74.00 | 25.33 |
| Video-LLaVA (Lin et al., 2023) | 7B | 0.46 | 0.99 | 99.33 | 6.67 | 6.67 |
| Onellm (Han et al., 2024) | 7B | 0.48 | 0.99 | 98.67 | 3.33 | 3.33 |
| Emotion-LLaMA (Cheng et al., 2024) | 7B | 0.32 | 0.81 | 80.00 | 15.33 | 14.00 |
| +PEP-MEK | | 0.09 | 0.60 | 63.33 | 44.67 | 28.00 |
| Llama3.2-vision (Grattafiori et al., 2024) | 11B | 0.25 | 0.79 | 82.67 | 33.33 | 22.67 |
| Qwen2.5-VL (Bai et al., 2025) | 32B | -0.02 | 0.47 | 60.67 | 65.33 | 41.33 |
| Qwen2.5-VL (Bai et al., 2025) | 72B | -0.07 | 0.39 | 64.00 | 77.33 | 50.00 |
| Gemma3 (Team et al., 2025) | 4B | -0.14 | 0.35 | 40.00 | 68.00 | 22.67 |
| Gemma3 (Team et al., 2025) | 12B | -0.17 | 0.22 | 46.67 | 80.00 | 39.33 |
| Gemma3 (Team et al., 2025) | 27B | 0.13 | 0.66 | 74.00 | 48.67 | 40.67 |
| Mistral-small3.1 (Mistral AI, 2024) | 24B | -0.44 | 0.02 | 10.67 | 98.00 | 9.33 |
| Qwen2.5-Omni (Xu et al., 2025) | 7B | -0.11 | 0.36 | 48.00 | 70.67 | 32.67 |
| +PEP-MEK | 7B | -0.11 | 0.34 | 54.00 | 76.00 | 40.00 |
| *Closed-source* | | | | | | |
| GPT-4o (Hurst et al., 2024) | - | 0.04 | 0.56 | 72.00 | 64.00 | 46.67 |
| QvQ-Max (Qwen Team, 2025) | - | 0.04 | 0.57 | 72.57 | 64.00 | 46.00 |
| Gemini-2.5-Pro (Google DeepMind, 2025) | - | 0.06 | 0.62 | 82.00 | 70.00 | 56.67 |
| Qwen-VL-Plus (Bai et al., 2025) | - | -0.24 | 0.22 | 34.67 | 82.00 | 26.00 |
| Qwen-VL-Max (Bai et al., 2025) | - | -0.07 | 0.38 | 63.33 | 78.00 | 50.00 |
| Gemini-2.5-Flash (Google DeepMind, 2025) | - | 0.05 | 0.58 | 74.67 | 64.67 | 46.67 |
| +PEP-MEK | - | 0.02 | 0.53 | 73.33 | 69.33 | 50.67 |
| GPT-4.1 (OpenAI, 2024) | - | 0.07 | 0.61 | 77.33 | 64.00 | 54.00 |
| GPT-5 (OpenAI, 2025b) | - | 0.01 | 0.52 | 79.33 | 77.33 | 64.00 |

are further visualized in Figure 13. From the figure, several key observations emerge: (1) model performance has steadily improved over time; (2) closed-source models generally outperform open-source models, though the performance gap is relatively small; (3) the incorporation of reasoning capabilities into open-source models does not lead to clear performance gains; (4) in most cases, larger parameter sizes tend to yield better results; and (5) PEP-MEK demonstrates its effectiveness, often achieving performance that matches or exceeds the best-performing models of the same period.

Furthermore, the lack of significant improvements from reasoning-augmented models may indicate that reasoning paradigms in the visual domain are still in their early stages and require further development and refinement. These findings highlight a critical challenge for future work: enhancing MLLMs' capacity to understand emotional cues embedded in visual content, and to reason more effectively about complex human affect in real-world image contexts, particularly through the continued advancement and adaptation of reasoning paradigms.

### C.2.4 MULTIMODALITY PERCEPTION: AUDIO

As shown in Table 14, we report the performance of various models on the EmotionHallucer-PA benchmark, which focuses on hallucinations in real-world, perception-based emotion understanding from audio. The results are further visualized in Figure 14. From the figure, several key observations emerge: (1) model performance has gradually improved over time, though it remains in the early stages, with most models only slightly exceeding the random guess baseline of 25%; (2) closed-source models significantly outperform open-source models; and (3) PEP-MEK demonstrates strong effectiveness, often achieving performance that surpasses the best models available at the time.

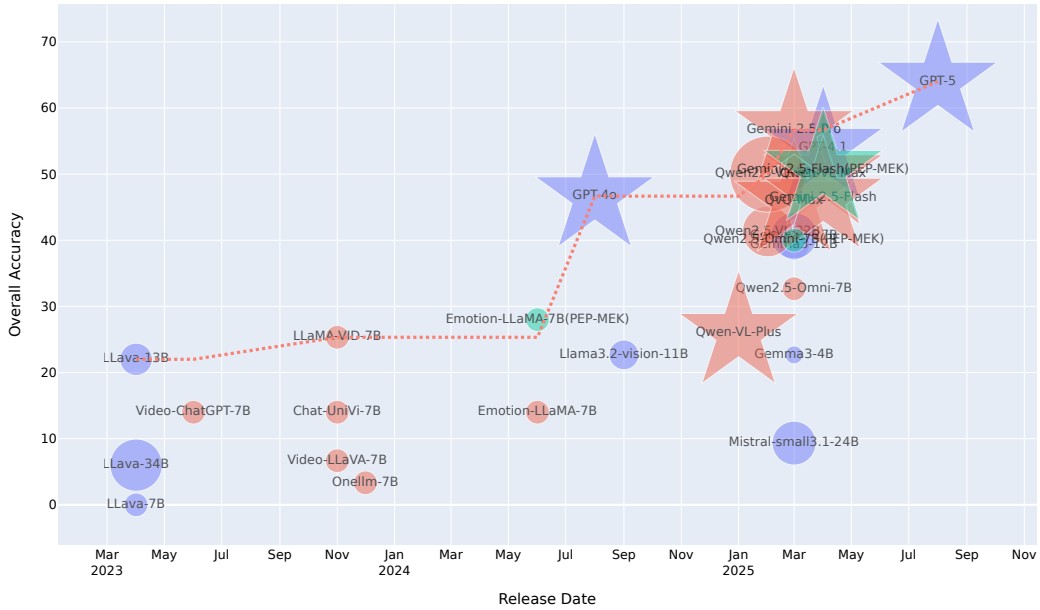

Figure 13: Performance comparison on EmotionHallucer-PI. Blue denotes MLLMs that accept only the current modality (e.g., image), red denotes models capable of handling additional modalities (e.g., video, audio), and cyan denotes the results obtained using PEP-MEK. Circles indicate known parameter sizes, while stars represent unknown sizes. The red dashed line marks the top-performing model of the current month.

Table 14: Performance comparison on EmotionHallucer-PA (Perception Audio).

| Methods | Model Size | Yes/No Bias | | Accuracy on EmotionHallucer-PA | | |
|---|---|---|---|---|---|---|
| | | Pct. Diff (∼0) | FP Ratio (∼0.5) | Basic ↑ | Hallucinated ↑ | Overall ↑ |
| *Open-source* | | | | | | |
| Onellm (Han et al., 2024) | 7B | 0.50 | 1.00 | 100.00 | 0.00 | 0.00 |
| Emotion-LLaMA (Cheng et al., 2024) | 7B | 0.43 | 0.92 | 91.85 | 6.52 | 5.16 |
| +PEP-MEK | | -0.14 | 0.36 | 33.68 | 64.67 | 22.28 |
| Qwen2.5-Omni (Xu et al., 2025) | 7B | -0.49 | 0.01 | 1.36 | 99.46 | 0.82 |
| +PEP-MEK | | -0.45 | 0.04 | 6.25 | 95.92 | 5.43 |
| Kimi-Audio (Ding et al., 2025) | 7B | -0.23 | 0.22 | 36.41 | 81.79 | 19.29 |
| *Closed-source* | | | | | | |
| Qwen-Audio-Turbo (Chu et al., 2024) | - | -0.49 | 0.01 | 2.17 | 99.46 | 1.63 |
| Gemini-2.5-Pro (Google DeepMind, 2025) | - | -0.14 | 0.34 | 41.03 | 69.02 | 24.46 |
| Gemini-2.5-Flash (Google DeepMind, 2025) | - | -0.13 | 0.34 | 45.11 | 71.74 | 30.43 |
| +PEP-MEK | | -0.12 | 0.34 | 48.10 | 72.83 | 34.51 |

We hypothesize that the relatively low performance in the audio modality is due to the current focus of audio-based MLLM research, which primarily targets tasks such as Automatic Speech Recognition (ASR) and semantic audio understanding. For instance, when refusing to answer, GPT-4o-Audio responds with: "I'm sorry, but I can't help with that request." These tasks emphasize transcribing or interpreting spoken content, but often neglect paralinguistic features, such as tone, intensity, and prosody, which are essential for emotion understanding. These findings highlight a critical challenge for future work: enhancing MLLMs' ability to perceive and interpret emotional signals embedded in speech, beyond lexical content. This calls for the development of audio-specific reasoning paradigms and training objectives that better capture the richness of human emotional expression in voice.

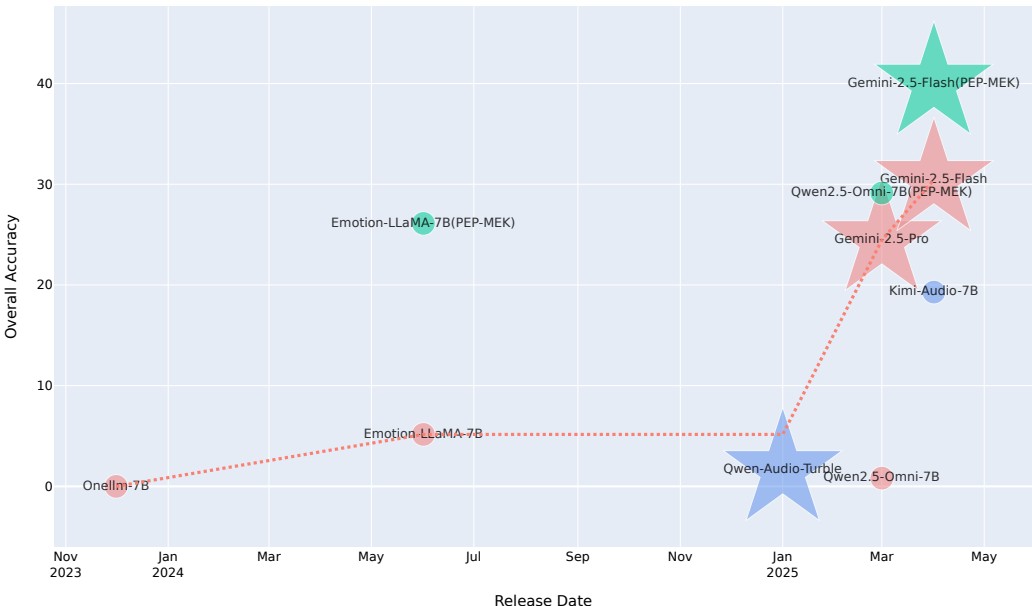

Figure 14: Performance comparison on EmotionHallucer-PA. Blue denotes MLLMs that accept only the current modality (e.g., audio), red denotes models capable of handling additional modalities (e.g., iamge, video), and cyan denotes the results obtained using PEP-MEK. Circles indicate known parameter sizes, while stars represent unknown sizes. The red dashed line marks the top-performing model of the current month.

### C.2.5 MULTIMODALITY PERCEPTION: SHORT VIDEO

As shown in Table 15, we report the performance of various models on the EmotionHallucer-PV/S, which focuses on hallucinations in real-world, perception-based emotion understanding from short videos. The results are further visualized in Figure 15. For models that only support image input, we adopt a key-frame sampling strategy to enable evaluation on video data.

Several key observations emerge from the figure: (1) model performance has steadily improved over time; (2) closed-source models generally outperform open-source models, though the performance gap remains modest; (3) for MLLMs that only support image input, increasing sampled frames typically does not lead to performance improvements; (4) some models, such as LLaMA-VID, Video-LLaVA, and OneLLM, perform particularly poorly on this task; (5) Emotion LLaMA, despite being fine-tuned on emotion recognition tasks, does not outperform general-purpose models; (6) PEP-MEK continues to demonstrate strong performance, often matching or exceeding that of the best models during the same period.

Furthermore, video-based emotion understanding and hallucination detection remains an open and underexplored challenge, with most models still in the early developmental stage. Our results also indicate that supervised finetuning on emotion tasks does not necessarily alleviate hallucination issues. Notably, although Emotion-LLaMA achieves excellent performance on MER, it performs poorly on EmotionHallucer-PV/S, constructed from the same data source but reframed for hallucination detection.

These findings highlight a critical challenge for future work: to develop video-language models with stronger emotion recognition and reasoning capabilities, and to identify effective strategies for mitigating hallucinations introduced by supervised finetuning (Perez et al., 2022; Achiam et al., 2023).

Table 15: Performance comparison on EmotionHallucer-PV/S (Perception Short Video).

| Methods | Model | Yes/No Bias | | Accuracy on EmotionHallucer-PV/S | | |
|---|---|---|---|---|---|---|
| | Size | Pct. Diff ($\sim$0) | FP Ratio ($\sim$0.5) | Basic ↑ | Hallucinated ↑ | Overall ↑ |
| *Open-source* | | | | | | |
| LLava (Liu et al., 2023)/1F | 7B | 0.50 | 1.00 | 100.00 | 0.00 | 0.00 |
| LLava (Liu et al., 2023)/1F | 13B | -0.34 | 0.14 | 19.44 | 87.22 | 11.11 |
| LLava (Liu et al., 2023)/1F | 34B | -0.44 | 0.04 | 7.22 | 96.11 | 3.89 |
| Llama3.2-vision (Grattafiori et al., 2024)/1F | 11B | 0.18 | 0.72 | 76.67 | 40.00 | 24.44 |
| Llama3.2-vision (Grattafiori et al., 2024)/2F | 11B | -0.46 | 0.02 | 5.00 | 97.78 | 3.33 |
| Llama3.2-vision (Grattafiori et al., 2024)/4F | 11B | -0.49 | 0.01 | 1.67 | 99.44 | 1.11 |
| Gemma3 (Team et al., 2025)/1F | 4B | 0.21 | 0.74 | 76.11 | 33.33 | 17.78 |
| Gemma3 (Team et al., 2025)/2F | 4B | 0.24 | 0.74 | 75.00 | 27.22 | 14.44 |
| Gemma3 (Team et al., 2025)/4F | 4B | 0.22 | 0.73 | 75.00 | 31.67 | 15.00 |
| Gemma3 (Team et al., 2025)/1F | 12B | 0.16 | 0.69 | 73.68 | 42.78 | 25.56 |
| Gemma3 (Team et al., 2025)/2F | 12B | 0.15 | 0.66 | 69.44 | 39.44 | 20.56 |
| Gemma3 (Team et al., 2025)/4F | 12B | 0.03 | 0.53 | 58.33 | 52.22 | 20.56 |
| Gemma3 (Team et al., 2025)/1F | 27B | 0.28 | 0.82 | 83.89 | 27.78 | 17.22 |
| Gemma3 (Team et al., 2025)/2F | 27B | 0.22 | 0.73 | 75.00 | 31.11 | 17.22 |
| Gemma3 (Team et al., 2025)/4F | 27B | 0.18 | 0.70 | 72.22 | 36.67 | 20.00 |
| Mistral-small3.1 (Mistral AI, 2024)/1F | 24B | -0.13 | 0.35 | 41.11 | 67.78 | 17.78 |
| Mistral-small3.1 (Mistral AI, 2024)/2F | 24B | -0.23 | 0.24 | 32.78 | 78.89 | 17.22 |
| Mistral-small3.1 (Mistral AI, 2024)/4F | 24B | -0.38 | 0.11 | 12.22 | 88.89 | 6.11 |
| Video-ChatGPT (Maaz et al., 2023) | 7B | -0.09 | 0.41 | 39.44 | 58.33 | 13.89 |
| Chat-UniVi (Jin et al., 2024) | 7B | -0.16 | 0.32 | 39.44 | 71.67 | 26.11 |
| LLaMA-VID (Li et al., 2024c) | 7B | 0.49 | 0.99 | 99.44 | 0.56 | 0.56 |
| Video-LLaVA (Lin et al., 2023) | 7B | 0.50 | 1.00 | 100.00 | 0.00 | 0.00 |
| Onellm (Han et al., 2024) | 7B | 0.50 | 1.00 | 100.00 | 0.00 | 0.00 |
| Emotion-LLaMA (Cheng et al., 2024) | 7B | 0.20 | 0.70 | 70.71 | 31.07 | 10.71 |
| +PEP-MEK | | -0.12 | 0.35 | 45.00 | 70.00 | 26.11 |
| Qwen2.5-VL (Bai et al., 2025) | 32B | 0.32 | 0.95 | 86.11 | 22.92 | 15.97 |
| Qwen2.5-VL (Bai et al., 2025) | 72B | 0.38 | 0.92 | 93.06 | 18.06 | 13.89 |
| Qwen2.5-Omni (Xu et al., 2025) | 7B | 0.37 | 0.91 | 91.67 | 17.36 | 13.19 |
| +PEP-MEK | | -0.09 | 0.38 | 51.39 | 70.14 | 29.17 |
| *Closed-source* | | | | | | |
| GPT-4o (Hurst et al., 2024)/1F | - | 0.00 | 0.50 | 57.78 | 57.22 | 20.56 |
| GPT-4o (Hurst et al., 2024)/2F | - | 0.05 | 0.56 | 62.22 | 52.78 | 24.44 |
| QvQ-Max (Qwen Team, 2025) | - | 0.23 | 0.76 | 79.86 | 34.73 | 27.08 |
| Gemini-2.5-Pro (Google DeepMind, 2025) | - | 0.24 | 0.81 | 85.90 | 36.46 | 30.77 |
| Qwen-VL-Plus (Bai et al., 2025) | - | -0.14 | 0.33 | 43.75 | 72.22 | 20.83 |
| Qwen-VL-Max (Bai et al., 2025) | - | 0.38 | 0.93 | 93.75 | 17.36 | 14.58 |
| Gemini-2.5-Flash (Google DeepMind, 2025) | - | 0.19 | 0.77 | 83.33 | 45.51 | 36.54 |
| +PEP-MEK | | -0.04 | 0.45 | 62.78 | 70.00 | 40.00 |
| GPT-4.1 (OpenAI, 2024)/1F | - | 0.20 | 0.72 | 75.56 | 35.56 | 19.44 |
| GPT-5 (OpenAI, 2025b)/1F | - | 0.01 | 0.51 | 58.33 | 57.22 | 23.89 |

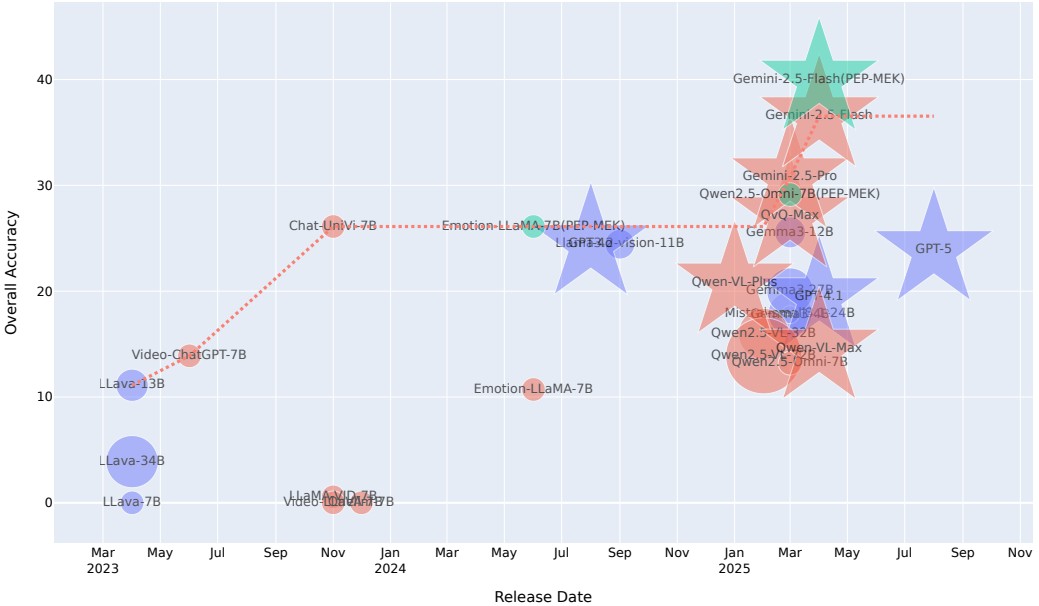

Figure 15: Performance comparison on EmotionHallucer-PV/S. Blue denotes MLLMs that accept only the current modality (e.g., video), red denotes models capable of handling additional modalities (e.g., image, audio), and cyan denotes the results obtained using PEP-MEK. Circles indicate known parameter sizes, while stars represent unknown sizes. The red dashed line marks the top-performing model of the current month.

### C.2.6 MULTIMODALITY PERCEPTION: LONG VIDEO

As shown in Table 16, we report the performance of various models on the EmotionHallucer-PV/L benchmark, which focuses on hallucinations in real-world, perception-based emotion understanding from long-form videos. The results are visualized in Figure 16. For models that only support image input, we adopt a key-frame sampling strategy to approximate video-level evaluation.

From the figure, several key observations emerge: (1) model performance has gradually improved over time, but overall results remain low, indicating that this task is still in its early stages; (2) closed-source models generally outperform open-source ones, though the performance gap is not as large as in other benchmarks; (3) nearly all models perform at or below random guess levels, underscoring the difficulty of emotion understanding in long video contexts; (4) Emotion LLaMA, despite being fine-tuned on emotion recognition tasks, fails to outperform general-purpose models, suggesting limited transferability to hallucination detection; (5) PEP-MEK demonstrates strong and consistent effectiveness, in many cases outperforming all models released in the same timeframe.

Additionally, we note that many MLLMs perform surprisingly poorly on this task, despite their strength on other modalities and tasks. This discrepancy suggests that current MLLMs are not yet equipped to reason about fine-grained emotion states and changes over long temporal spans, and may lack both temporal integration and emotion-specific understanding capabilities.

These findings highlight a critical challenge for future work: to develop temporal-aware, emotion-grounded MLLMs capable of robust reasoning over long-form emotional content. It will also be essential to explore new reasoning paradigms and training strategies that directly target hallucination resilience in dynamic emotional contexts.

### C.3 RESULT OF MULTIMODALITY

We further present the hallucination performance of models on EmotionHallucer-NoAudio in Table 17 and Figure 17, providing an overall view of multimodal capability. From the figure, several key trends can be observed: (1) model performance has consistently improved over time, reflect-

Table 16: Performance comparison on EmotionHallucer-PV/L (Perception Long Video).

| Methods | Model Size | Yes/No Bias | | Accuracy on EmotionHallucer-PV/L | | |
|---|---|---|---|---|---|---|
| | | Pct. Diff (∼0) | FP Ratio (∼0.5) | Basic ↑ | Hallucinated ↑ | Overall ↑ |
| *Open-source* | | | | | | |
| LLava (Liu et al., 2023)/1F | 7B | 0.50 | 1.00 | 100.00 | 0.00 | 0.00 |
| LLava (Liu et al., 2023)/1F | 13B | -0.13 | 0.37 | 38.31 | 63.68 | 5.97 |
| LLava (Liu et al., 2023)/1F | 34B | -0.44 | 0.05 | 6.47 | 95.02 | 1.99 |
| Llama3.2-vision (Grattafiori et al., 2024)/1F | 11B | 0.25 | 0.77 | 79.10 | 29.35 | 12.44 |
| Llama3.2-vision (Grattafiori et al., 2024)/2F | 11B | -0.38 | 0.11 | 12.44 | 89.05 | 3.48 |
| Llama3.2-vision (Grattafiori et al., 2024)/4F | 11B | -0.49 | 0.01 | 0.50 | 98.51 | 0.00 |
| Gemma3 (Team et al., 2025)/1F | 4B | -0.03 | 0.47 | 49.25 | 55.22 | 12.94 |
| Gemma3 (Team et al., 2025)/2F | 4B | -0.12 | 0.37 | 41.79 | 65.67 | 10.95 |
| Gemma3 (Team et al., 2025)/4F | 4B | -0.06 | 0.43 | 49.25 | 61.69 | 14.93 |
| Gemma3 (Team et al., 2025)/1F | 12B | -0.07 | 0.42 | 48.26 | 62.69 | 16.42 |
| Gemma3 (Team et al., 2025)/2F | 12B | -0.10 | 0.39 | 44.28 | 64.18 | 13.43 |
| Gemma3 (Team et al., 2025)/4F | 12B | -0.15 | 0.33 | 40.30 | 70.65 | 13.93 |
| Gemma3 (Team et al., 2025)/1F | 27B | -0.01 | 0.49 | 52.74 | 54.23 | 9.95 |
| Gemma3 (Team et al., 2025)/2F | 27B | 0.00 | 0.50 | 54.23 | 54.73 | 11.44 |
| Gemma3 (Team et al., 2025)/4F | 27B | -0.09 | 0.41 | 44.78 | 62.19 | 11.94 |
| Mistral-small3.1 (Mistral AI, 2024)/1F | 24B | -0.40 | 0.08 | 12.44 | 92.04 | 4.98 |
| Mistral-small3.1 (Mistral AI, 2024)/2F | 24B | -0.45 | 0.05 | 5.47 | 95.52 | 2.49 |
| Mistral-small3.1 (Mistral AI, 2024)/4F | 24B | -0.48 | 0.02 | 2.99 | 98.51 | 1.49 |
| Video-ChatGPT (Maaz et al., 2023) | 7B | 0.17 | 0.66 | 64.18 | 30.85 | 13.43 |
| Chat-UniVi (Jin et al., 2024) | 7B | 0.13 | 0.63 | 62.69 | 36.82 | 18.41 |
| LLaMA-VID (Li et al., 2024c) | 7B | 0.49 | 0.98 | 97.51 | 0.00 | 0.00 |
| Video-LLaVA (Lin et al., 2023) | 7B | 0.50 | 1.00 | 100.00 | 0.50 | 0.50 |
| Onellm (Han et al., 2024) | 7B | 0.50 | 1.00 | 100.00 | 0.00 | 0.00 |
| Emotion-LLaMA (Cheng et al., 2024) | 7B | 0.36 | 0.86 | 86.57 | 14.93 | 7.96 |
| +PEP-MEK | | -0.04 | 0.45 | 52.74 | 60.70 | 24.88 |
| Qwen2.5-VL (Bai et al., 2025) | 32B | -0.17 | 0.31 | 36.32 | 71.14 | 9.95 |
| Qwen2.5-VL (Bai et al., 2025) | 72B | -0.16 | 0.32 | 38.81 | 70.65 | 11.44 |
| Qwen2.5-Omni (Xu et al., 2025) | 7B | -0.36 | 0.12 | 16.42 | 88.06 | 5.47 |
| +PEP-MEK | | -0.06 | 0.43 | 44.28 | 57.21 | 17.91 |
| *Closed-source* | | | | | | |
| GPT-4o (Hurst et al., 2024)/1F | - | -0.41 | 0.08 | 9.95 | 92.04 | 4.48 |
| GPT-4o (Hurst et al., 2024)/2F | - | -0.37 | 0.12 | 15.42 | 88.56 | 5.97 |
| QvQ-Max (Qwen Team, 2025) | - | -0.05 | 0.45 | 49.25 | 58.71 | 17.41 |
| Gemini-2.5-Pro (Google DeepMind, 2025) | - | -0.03 | 0.46 | 55.72 | 62.19 | 20.90 |
| Qwen-VL-Plus (Bai et al., 2025) | - | -0.49 | 0.01 | 0.50 | 99.00 | 0.00 |
| Qwen-VL-Max (Bai et al., 2025) | - | -0.05 | 0.45 | 49.25 | 59.20 | 11.94 |
| Gemini-2.5-Flash (Google DeepMind, 2025) | - | -0.06 | 0.43 | 48.76 | 60.70 | 18.41 |
| +PEP-MEK | | -0.19 | 0.29 | 37.81 | 75.12 | 21.39 |
| GPT-4.1 (OpenAI, 2024)/1F | - | -0.29 | 0.19 | 24.88 | 82.59 | 10.45 |
| GPT-5 (OpenAI, 2025b)/1F | - | -0.42 | 0.08 | 9.45 | 92.54 | 3.48 |

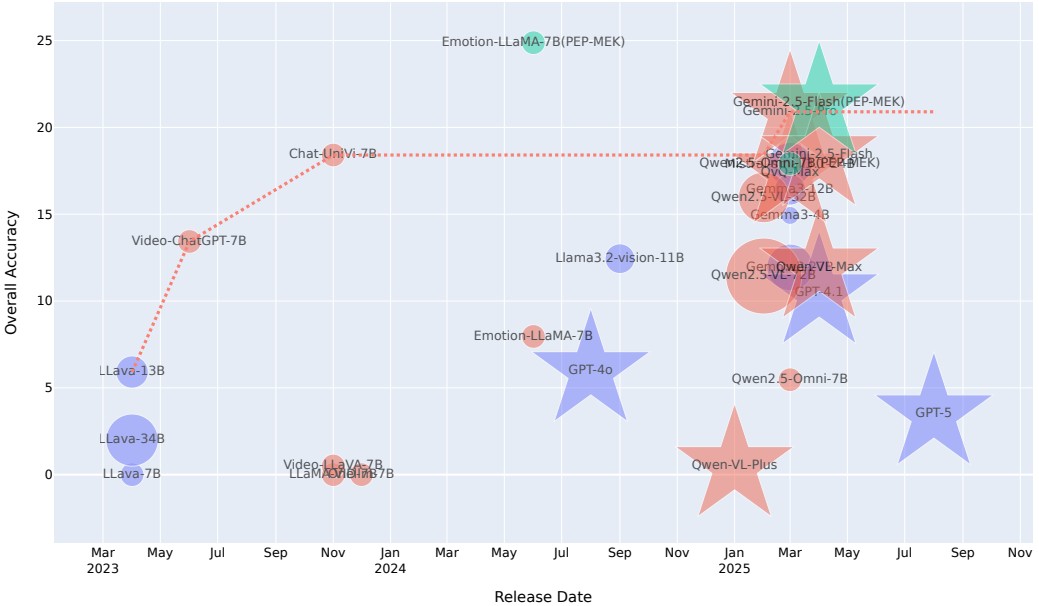

Figure 16: Performance comparison on EmotionHallucer-PV/L. Blue denotes MLLMs that accept only the current modality (e.g., video), red denotes models capable of handling additional modalities (e.g., image, audio), and cyan denotes the results obtained using PEP-MEK. Circles indicate known parameter sizes, while stars represent unknown sizes. The red dashed line marks the top-performing model of the current month.

ing the rapid progress in multimodal learning; (2) recent image-based MLLMs outperform earlier models with video-processing capabilities, suggesting advances in visual understanding even without temporal cues; (3) open-source models still lag behind closed-source counterparts, though the performance gap is gradually narrowing; (4) overall, the Gemini series stands out for both its ability to handle all input modalities and its strong overall performance across tasks.

## D    MORE EXAMPLES OF EMOTIONHALLUCER

In this section, we provide more cases from EmotionHallucer, as shown Figure 18, Figure 19, Figure 20, Figure 21, Figure 22, and Figure 23.  The resources are available at `https://anonymous.4open.science/r/EmotionHallucer`.

## E    LIMITATIONS AND DISCUSSIONS

While our benchmark provides a comprehensive evaluation of emotion hallucinations across modalities, several limitations remain.  (1) Although we take a lot of strategies to make sure the quality of EmotionHallucer, there is noise introduced by human annotations. (2) The benchmark currently uses English as the interaction language with LLMs. Although the underlying multimodal data come from diverse cultural contexts, the current version does not yet systematically evaluate how cross-lingual or cross-cultural variation affects emotion understanding and hallucination. (3) While we observe emotion hallucination phenomena, the underlying causes, such as pretraining biases, modality misalignment, or lack of emotion-specific supervision, remain underexplored. Understanding these root causes is essential for designing more robust and interpretable models. (4) Although we treat emotion understanding and hallucination detection separately, real-world applications often require joint emotion understanding and hallucination awareness, which calls for unified modeling strategies that integrate both capabilities. (5) Open-ended evaluation. Our current benchmark primarily relies on binary QA, which ensures clear ground-truth supervision but remains insufficient to capture the open-ended hallucinations in real-world settings.

Table 17: Performance comparison on EmotionHallucer-NoAudio with additional "Yes/No bias" analysis.

| Methods | Model Size | Yes/No Bias Pct. Diff (~0) | FP Ratio (~0.5) | Accuracy on EmotionHallucer–NoA Basic ↑ | Hallucinated ↑ | Overall ↑ |
|---|---|---|---|---|---|---|
| *Open-source* | | | | | | |
| LLaVA (Liu et al., 2023) | 34B | -0.05 | 0.45 | 50.25 | 59.52 | 10.27 |
| Video-ChatGPT (Maaz et al., 2023) | 7B | 0.09 | 0.59 | 61.91 | 44.67 | 18.44 |
| Chat-UniVi (Jin et al., 2024) | 7B | 0.09 | 0.59 | 60.22 | 42.37 | 11.07 |
| LLaMA-VID (Li et al., 2024c) | 7B | 0.36 | 0.86 | 85.74 | 13.66 | 5.78 |
| Video-LLaVA (Lin et al., 2023) | 7B | 0.49 | 1.00 | 99.70 | 1.50 | 1.50 |
| Onellm (Han et al., 2024) | 7B | 0.31 | 0.85 | 87.34 | 26.02 | 15.35 |
| Emotion-LLaMA (Cheng et al., 2024) | 7B | 0.12 | 0.63 | 66.55 | 42.43 | 18.86 |
| Llama3.2-vision (Grattafiori et al., 2024) | 11B | 0.21 | 0.78 | 83.05 | 41.28 | 29.91 |
| Gemma3 (Team et al., 2025) | 27B | 0.15 | 0.70 | 78.66 | 49.15 | 33.90 |
| Qwen2.5-VL (Bai et al., 2025) | 72B | 0.08 | 0.63 | 78.08 | 62.15 | 43.02 |
| Mistral-small3.1 (Mistral AI, 2024) | 24B | -0.11 | 0.35 | 53.94 | 75.17 | 32.20 |
| Qwen2.5-Omni (Xu et al., 2025) | 7B | 0.11 | 0.65 | 72.39 | 49.74 | 25.44 |
| *Closed-source* | | | | | | |
| QvQ-Max (Qwen Team, 2025) | - | 0.07 | 0.63 | 78.18 | 63.39 | 47.98 |
| GPT-4o (Hurst et al., 2024) | - | -0.01 | 0.48 | 67.10 | 69.49 | 40.98 |
| GPT-4.1 (OpenAI, 2024) | - | 0.05 | 0.58 | 74.58 | 64.71 | 44.47 |
| Gemini-2.5-Flash (Google DeepMind, 2025) | - | 0.06 | 0.61 | 78.55 | 66.80 | 50.56 |
| Gemini-2.5-Pro (Google DeepMind, 2025) | - | 0.07 | 0.64 | 81.31 | 67.01 | 51.58 |
| GPT-5 (OpenAI, 2025b) | - | -0.06 | 0.40 | 67.10 | 78.17 | 49.35 |

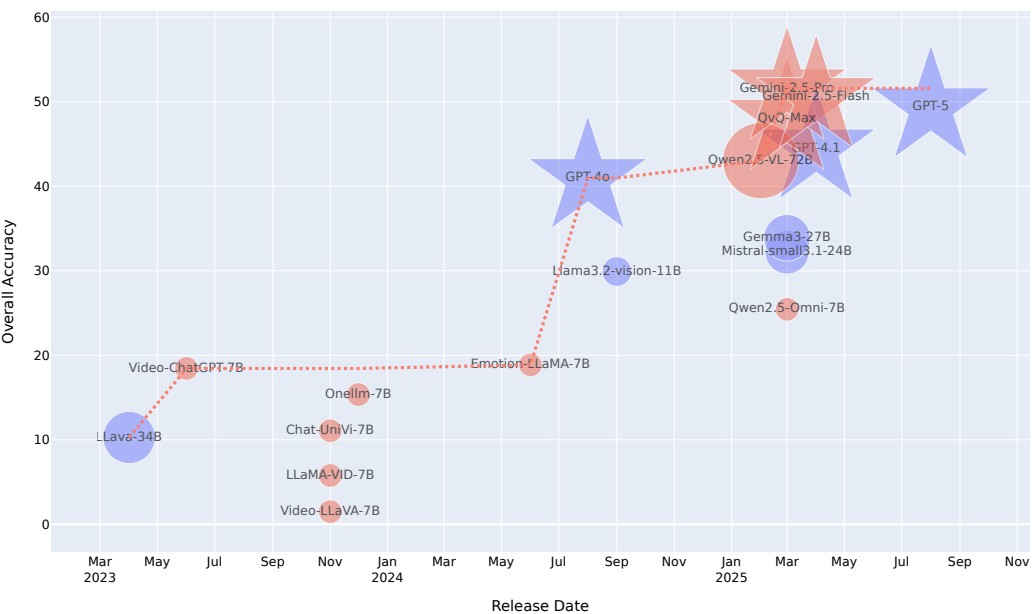

Figure 17: Performance comparison on EmotionHallucer-NoAudio. Blue denotes MLLMs that accept only the image modality, red denotes models capable of handling additional modalities (e.g., video, audio). Circles indicate known parameter sizes, while stars represent unknown sizes. The red dashed line marks the top-performing model of the current month.

**Basic:** Facial feedback hypothesis is the hypothesis that a posed facial expression of emotion can help generate an emotional feeling.

**Hallucinated:** Facial feedback hypothesis is the hypothesis that a posed facial expression of emotion has no effect on emotional feelings and is unrelated to emotional experience.

**Basic:** As suggested by the affect infusion model, the emotion humans feel at any moment influences how they react to other events, even if they are unrelated to whatever evoked their emotion.

**Hallucinated:** As suggested by the affect infusion model, the emotion humans feel at any moment influence how they react to other events, but only if those events are related to whatever evoked their emotion.

Figure 18: Two examples of basic-hallucinated pair for Theory Hallucination.

**Basic:** Appraisal is cognitive evaluation of what a stimulus or situation means for one's goals, concerns, and well-being.

**Hallucinated:** Appraisal is cognitive evaluation of what a stimulus or situation means for one's emotional state.

**Basic:** Anxiety is a general expectation that something bad might happen, without identifying any particular danger.

**Hallucinated:** Anxiety is a general expectation that something good might happen, without identifying any particular danger.

Figure 19: Two examples of basic-hallucinated pair for Definition Hallucination.

**Basic:** "Happiness" means different things to people in different cultures, even setting aside issues of language. People in many Asian cultures prefer a sense of contentment and knowledge that one has done one's duty toward family and community, whereas North Americans place greater value on excitement and achievement.
**Hallucinated:** "Happiness" means different things to people in different cultures, even setting aside issues of language. People in many North American cultures prefer a sense of contentment and knowledge that one has done one's duty toward family and community, whereas Asian culture place greater value on excitement and achievement.

**Basic:** Simply believing that you have some control over a situation can effectively reduce stress levels.

**Hallucinated:** Simply believing that you have no control over a situation can effectively reduce stress levels.

Figure 20: Two examples of basic-hallucinated pair for Definition Hallucination.

**Category Hallucination**

**Basic:** The person exhibits a worried emotion.

**Hallucinated:** The person exhibits an angry emotion.

Figure 21: An example of basic-hallucinated pair for Category Hallucination.

**Intensity Hallucination**

**Review Text:** As a poker enthusiast I was looking forward to seeing this movie - Especially as it had Scotty Nyugen in it.

Basically, Scotty Nyugens short spots in this film are all it has going for it.

The characters are unlikeable and annoying, the soundtrack is awful and the plot, well, there isn't one.

I honestly got a headache and found myself reading the barcode number on the DVD box after twenty minutes I was THAT bored. Its actually ashame that Nyugen was in this movie as otherwise I wouldn't have wasted $16 buying it off Ebay.

Take it from me - AVOID like 7 2 offsuit!!! Dire. :(

**Basic:** This review is a very negative sentiment, because …

**Hallucinated:** This review is a slightly negative sentiment, because …

Figure 22: An example of basic-hallucinated pair for Intensity Hallucination.

**Reasoning Hallucination**

**Basic:** The man does not display obvious pride or shame about his retail job, instead adopting a defensive stance. From his firm posture and slight nod, it is evident that, despite considering it a step down, he remains composed. Even as his tone carries a hint of defensiveness, his brief pause before speaking suggests he acknowledges the perceived step down.

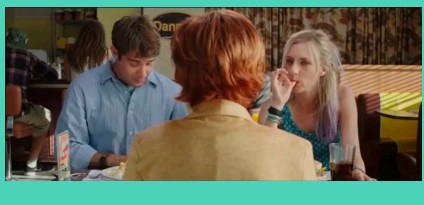

**Hallucinated:** The man displays obvious shame about his retail job. From his somewhat anxious posture and slight nod, it is evident that he considers it a step down and struggles to remain composed. His tone carries a hint of embarrassment, and his brief pause before speaking suggests he is unsure how to respond.

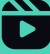

**Basic:** The red-haired woman looks to the side at 0:45 because she is calling the man's girlfriend a mean name and avoids looking at him. As she speaks, her gaze shifts away, her body angling slightly to avoid direct confrontation.

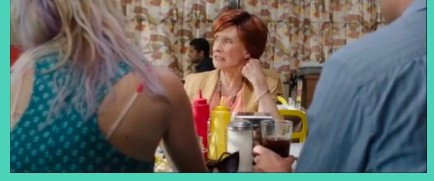

**Hallucinated:** The red-haired woman looks to the side at 0:45 because she is calling the man's girlfriend a mean name and avoids looking at him. As she speaks, her gaze shifts away, her breathing quickens, and her body angles slightly to avoid direct confrontation.

Figure 23: An example of basic-hallucinated pair for Reasoning Result and Cue Hallucination.

# F FUTURE WORK

In this section, we outline several directions for extending EmotionHallucer based on the limitations identified in our current design and empirical findings. Rather than treating these limitations as isolated issues, we view them as stepping stones toward a progressively richer and more comprehensive framework for evaluating emotion understanding and emotion hallucination in MLLMs. Each proposed direction—ranging from open-ended evaluation, multilingual and cross-cultural extensions, and deeper mechanistic probing to fully integrated joint reasoning—builds on the diagnostic foundation established in this work. These future developments will enable EmotionHallucer to evolve from an initial fine-grained benchmark into a broader evaluation suite capable of supporting advanced research on emotionally reliable and trustworthy MLLMs.

**Open-ended evaluation.** Open-ended setting represents an important next step for extending EmotionHallucer beyond binary QA. Although our current binary QA format provides a controlled and reliable diagnostic setting, it cannot fully capture the expressive richness of free-form emotional descriptions. To examine its extensibility, we further conducted a pilot open-ended evaluation where MLLMs (Gemini-2.5-Flash in our case) were asked to generate free-form emotion descriptions, and a judge model (GPT-4o in our case) was used to evaluate whether the generated descriptions aligned with the basic or hallucinated references. We present the independent confusion matrices in Figure 25, and in Figure 26 we report pair-level consistency (a pair is counted as consistent only if the description for the basic item is judged aligned with the basic reference and the description for its hallucinated counterpart is judged not aligned). The overall consistency between binary QA and open-ended evaluation is reasonably aligned, suggesting that our binary paradigm may serve as a reliable proxy for open-ended settings.

Beyond these preliminary observations, open-ended evaluation itself introduces additional challenges. Judge models often struggle to evaluate long and diverse free-form responses, especially when emotion cues are subtle, indirect, or embedded in implicit contextual reasoning. Their judgments can also be sensitive to prompt wording and instruction framing, echoing findings from prior work on evaluator instability (Wang et al., 2023a;b; Li et al., 2024b; Krumdick et al., 2025). These issues collectively limit the reliability of fully automated open-ended assessment. One natural solution is to incorporate human-rated open-ended evaluations, where annotators directly assess the correctness, grounding, and hallucination tendency of free-form emotional descriptions. However, high-quality human rating for multimodal emotional reasoning is costly and difficult to scale—particularly when responses require fine-grained interpretation of emotion cues or long-form multimodal contexts. Given these constraints, we believe that structured open-ended annotation provides a more sustainable and reliable path forward. Instead of rating entire responses holistically, annotators label key interpretable components—such as entities, activated versus non-activated emotional cues, emotion category and intensity, and the reasoning pathway, as shown in Figure 24. This structured protocol is grounded in long-standing frameworks from psychology research, which emphasize that emotional understanding emerges from identifiable verbal and nonverbal cues as well as their integration (Scherer, 2003; Knapp et al., 1972). Leveraging these well-established theoretical foundations ensures that the annotation components are not arbitrary, but follow empirically supported principles of human emotional processing. This approach enables fine-grained identification of hallucinations at both the span level and the holistic reasoning level. Moreover, structured annotations naturally align with the cognitive stages used in EmotionHallucer, offering a principled extension of the current binary QA framework toward richer open-ended evaluation. We emphasize that this structured scheme is a preliminary design intended to illustrate a feasible direction; in future iterations of EmotionHallucer, we plan to refine, expand, and further validate this annotation protocol to support more comprehensive open-ended assessments.

**Cross-lingual or cross-cultural evaluation.** Building on our current English-based interaction protocol, a natural next step is to extend EmotionHallucer into multilingual and cross-cultural settings. A more comprehensive cross-lingual evaluation requires systematically examining how MLLMs' emotion understanding and hallucination patterns change when the interaction language itself varies, inspired by Ponti et al. (2020). Additionally, languages differ in structural properties, affective lexicalization, and pragmatic conventions—factors that directly influence how to interpret emotion cues, resolve ambiguity, and map linguistic expressions to affective states. Understanding these language-driven shifts is essential for assessing whether emotional reasoning remains stable across different linguistic interfaces.

Beyond interaction language, emotion communication itself is known to vary across cultures, including differences in display rules, emotion appraisal patterns, and nonverbal expressive norms (Mesquita and Frijda, 1992). Future versions of EmotionHallucer will explore culturally grounded extensions—such as parallel multimodal samples across cultural groups, culturally specific emotional frameworks, and cross-cultural reasoning tasks—to evaluate whether MLLMs can adapt to culturally shaped emotional cues.

**Deeper mechanistic understanding.** While EmotionHallucer reveals clear patterns of emotion hallucination, the deeper causal mechanisms behind these failures—such as pretraining corpus biases, modality misalignment, and insufficient emotion-specific supervision—remain largely unexplored. Progress toward understanding these root causes requires evaluation foundations that make it possible to probe models from multiple complementary angles. In this sense, the current binary QA forms the core foundation for controlled evaluation, while the planned structured open-ended evaluation and cross-lingual/cross-cultural extensions will serve as additional pillars that enable deeper mechanistic analysis rather than endpoints themselves. Together, these foundational tools will support future efforts to move beyond surface error characterization toward principled insights into how emotional information is encoded, transformed, and sometimes distorted within modern MLLMs.

**Joint emotion understanding and hallucination.** A coherent integration of emotional perception, inference, and hallucination-awareness is an important long-term goal for MLLMs. However, achieving robust joint reasoning first requires a phase-by-phase understanding of where each type of error originates. EmotionHallucer adopts this early-stage decomposition to establish such a foundation. Building on this basis, future extensions of EmotionHallucer can move toward joint formulations where emotion understanding and hallucination mitigation are assessed within a unified reasoning process. Our planned structured open-ended annotation scheme is particularly promising in this regard: by explicitly labeling entities, activated and non-activated cues, emotional categories, intensities, and reasoning pathways, it provides the representational scaffolding needed to connect perception, inference, and hallucination detection within a single coherent framework. This opens the door to future benchmarks and models that more closely mirror real-world emotion reasoning, where understanding and self-monitoring operate hand in hand.

## G  ETHICS STATEMENT

This work complies with ethical standards for AI research. All datasets used in this study are publicly available and were originally released under appropriate research or academic licenses. No private or sensitive personal data were involved. Our benchmark focuses on evaluating emotional hallucinations in language and multimodal models; however, we acknowledge the subjective and culturally nuanced nature of emotion, and we caution against overinterpreting model outputs in high-stakes or sensitive applications. Furthermore, hallucinations in emotional reasoning may lead to miscommunication or emotional misjudgment, particularly in domains such as mental health, education, or human-computer interaction. We encourage future work to incorporate robust safety checks, human oversight, and culturally inclusive evaluations when deploying such models in real-world scenarios.

## H  REPRODUCIBILITY STATEMENT

We have taken several steps to ensure the reproducibility of our work. Our benchmark is constructed from publicly available datasets, and we provide a detailed description of the annotation and processing pipeline in the appendix. The prompts used for MLLMs are fully documented in the appendix.

## I  LLM USAGE STATEMENT

In this paper, we used LLMs to assist with writing (language polishing). In addition, we employed several LLMs and MLLMs for evaluation purposes. The paper provides a complete list of the models and prompts used in these evaluations.

```
{
  "entity": "...",
  "perceptual_cues": {
    "verbal_cues": {
      "lexical_emotion_words": {
        "evidence": ["devastated", "heartbroken"]
      },
      "emotion_related_content": {
        "summary": "The speaker talks about losing an important relationship."
      }
    },
    "nonverbal_cues": {
      "visual": {
        "facial_expression": {
          "description": "teary eyes, downcast gaze"
        },
        "head_position": {
          "description": "head tilted downward"
        },
        "shoulder_posture": {
          "description": ""
        },
        "torso_posture": {
          "description": "leaning forward slightly"
        },
        "arm_gesture": {
          "description": ""
        },
        "hand_gesture": {
          "description": ""
        },
        ...
      },
      "acoustic": {
        "voice_prosody": {
          "description": "low pitch, slow tempo, long pauses"
        },
        "voice_quality": {
          "description": "trembling, weak voice"
        }
      }
    }
  },
  "emotion_inference": {
    "category": "sadness",
    "intensity": "high"
  },
  "reasoning": {
  "inferred_emotion": {
    "category": "sadness",
    "intensity": "high"
  },
  "supporting_cues": {
    "verbal": [
      "verbal_cues.lexical_emotion_words",
      "verbal_cues.emotion_related_content"
    ],
    "visual": [
      "nonverbal_cues.visual.facial_expression",
      "nonverbal_cues.visual.head_position",
      "nonverbal_cues.visual.torso_posture"
    ],
    "acoustic": [
      "nonverbal_cues.acoustic.voice_prosody",
      "nonverbal_cues.acoustic.voice_quality"
    ]
  },
  "conflicting_or_absent_cues": {
    "visual": [
      "nonverbal_cues.visual.shoulder_posture",
      "nonverbal_cues.visual.arm_gesture",
      "nonverbal_cues.visual.hand_gesture"
    ],
    "acoustic": []
  },
  "explanation": "The sadness inference is supported by ..."
  }
}
```

Figure 24: Envisioned structured annotation format for open-ended evaluation.



(a) Perception-T      (b) Perception-I      (c) Perception-V/S      (d) Perception-V/L

Figure 25: Confusion matrices for hallucination detection in the open-ended setting.

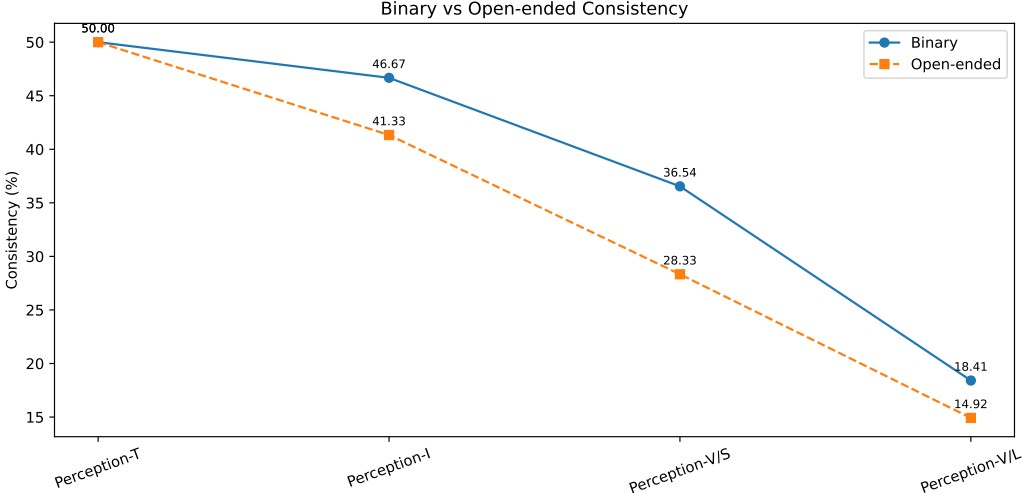

Figure 26: Binary vs open-ended consistency of hallucination detection across different modalities.

