# OpenReview forum: "EmotionHallucer: Evaluating Emotion Hallucinations in Multimodal Large Language Models"
_ICLR.cc/2026/Conference — ICLR 2026 Poster_

### Official Review · Reviewer_jLHy · 2025-10-20

**Soundness:** 4
**Presentation:** 3
**Contribution:** 4
**Rating:** 8
**Confidence:** 5

**Summary:**

This paper introduces EmotionHallucer, the first benchmark designed to evaluate emotion hallucinations in MLLMs. The benchmark assesses two complementary aspects, emotion psychology knowledge and multimodal emotion perception. Evaluating 41 models using an adversarial binary QA framework, the paper reports three main findings and propose PEP-MEK, a framework that integrates modality-specific and emotional reasoning to mitigate emotion hallucination. PEP-MEK achieves an average 9.9% improvement across selected models. Therefore, EmotionHallucer provides a benchmark and valuable insights for advancing emotionally reliable MLLMs.

**Strengths:**

1.	The paper introduces the first study of emotion hallucinations in MLLMs, addressing a crucial yet previously unexplored aspect of emotion understanding.

2.	EmotionHallucer is designed based on emotion psychology and real-world emotion perception, spanning four modalities and multiple evaluation settings. The adversarial QA framework provides a controlled protocolto assess emotional reasoning errors.

3.	The benchmark evaluates a large number of models, providing comprehensive insights into the current state of MLLMs.

4.	PEP-MEK is compatible with both open- and closed-source models through standard APIs, offering a plug-and-play approach for emotion hallucination mitigation.

5.	The paper is well-structured, easy to follow.

**Weaknesses:**

1.	While the adversarial binary QA framework provides strong objectivity and control, and the authors have additionally performed consistency checks between binary and open-ended results, this setting may still not fully capture open-ended emotion hallucinations in real-world generative scenarios. It would be valuable to further discuss how this limitation could be addressed, and what directions the authors plan to explore in future work.

2.	The benchmark primarily focuses on English-language. Although some data sources contain diverse cultural content, explicitly incorporating cross-cultural emotion understanding tasks would further enhance its generalizability.

3.	While this trade-off is reasonable given the improvement in reliability, a more detailed discussion on the efficiency–accuracy balance and potential optimization strategies would further strengthen the work.

**Questions:**

See Weakness.

---

> ### Author Response · Authors · 2025-11-19
> **Response to Weaknesses 1-2**
>
> We thank the reviewer for the valuable comments and appreciate the AC, SAC, and PC for managing the review process. We are grateful for the reviewer’s positive remarks on the strengths of our work, and we respond to the weaknesses and questions point by point below.
>
> **Weakness 1**: We thank the reviewer for this thoughtful comment. We agree that real-world generative scenarios often involve open-ended emotion understanding, and a controlled binary QA format cannot fully capture this complexity. Our decision to adopt the adversarial QA framework is intentional: it provides strong objectivity, minimal confounds, and fine-grained diagnostic power, enabling us to isolate where emotion hallucinations emerge across perceptual and reasoning stages. This level of control is difficult to achieve in open-ended settings, where generation variability and stylistic differences can obscure the underlying failure mode. Importantly, we have already performed binary-open-ended consistency checks (Appendix E), where we evaluate models under a free-form setting using LLM judges. The observed consistency in hallucination patterns indicates that the binary QA formulation serves as a reliable proxy for open-ended emotional reasoning. Still, we fully acknowledge that open-ended hallucinations may manifest in richer and more diverse forms. To address this limitation, we plan to extend EmotionHallucer in two directions. First, we will incorporate free-form open-ended tasks where models generate emotional interpretations, causal explanations, and multimodal reasoning narratives, allowing us to capture naturally occurring hallucinations in generative outputs. Second, we are developing a structured open-ended annotation scheme in which annotators label entities, activated emotional cues, emotion categories and intensities, and reasoning pathways. This structure preserves open-ended expressiveness while enabling fine-grained, interpretable, and span-level hallucination diagnosis.
>
> **Weakness 2**: We thank the reviewer for raising this important concern. We acknowledge the reviewer's valid points regarding language limitations. As a first pioneering benchmark in the field of emotion hallucination, our priority is to establish a rigorous, controlled, and reproducible foundation.  Our current benchmark uses English as the interaction language with MLLMs primarily for two reasons: first, to ensure a controlled and reproducible comparison across the wide spectrum of MLLMs, many of which are primarily optimized for English, and second, because English currently serves as a lingua franca in scientific research and AI development, which facilitates the initial adoption and dissemination of our benchmark within the research community. However, this does not mean that the emotion content itself is culturally narrow.
> The multimodal data in EmotionHallucer, drawn from social media posts, YouTube videos, TV shows, and cinematic content, naturally spans diverse cultural backgrounds. These sources encompass variations in emotion display rules, conversational dynamics, and situational emotional cues, providing rich cultural coverage. In addition, the Emotion Psychology Knowledge part incorporates insights from cross-cultural emotion research. This grounding ensures that our EmotionHallucer is not tied to a single cultural framework and remains widely applicable. That said, we fully acknowledge that systematically examining how cultural and linguistic differences influence emotional interpretation and hallucination patterns goes beyond the scope of the current release and remains an important direction for future work. We plan to extend EmotionHallucer to the following: multilingual settings covering diverse languages, culture-grounded emotional frameworks with culturally specific cues and appraisal norms, parallel multimodal samples enabling direct cross-cultural comparison, and cross-lingual emotional reasoning tasks that test how MLLMs adapt to culturally grounded emotion expressions. We believe these extensions will enhance the benchmark’s generalizability and offer deeper insights into how language, culture, and multimodal emotion reasoning interact in modern MLLMs.

---

> ### Author Response · Authors · 2025-11-19
> **Response to Weaknesses 3**
>
> **Weaknesses 3**: We thank the reviewer for this constructive comment. We agree that the efficiency–accuracy trade-off deserves further clarification. Our aim with PEP-MEK is to prioritize reliability and interpretability in emotional reasoning, and the multi-stage procedure naturally increases inference time. However, this design allows the model to explicitly re-ground its predictions using emotion-specific cues, substantially reducing hallucination rates and improving the stability of outputs, an essential objective for safety-critical emotional reasoning tasks. In the future, we will explore several optimization strategies. First, we plan to develop lighter-weight variants of the pipeline that selectively trigger deeper reasoning only when the model’s initial prediction is uncertain or inconsistent. Second, by incorporating confidence estimation or adaptive prompting, we can dynamically adjust the depth of reasoning and explicitly allow the model to answer “I don’t know” when the emotional content is uncertain. Third, by leveraging EmotionHallucer’s fine-grained categorization, we can identify which specific components of the emotional reasoning pipeline. This enables us to diagnose whether the root cause stems from pretraining biases, insufficient multimodal alignment, or lack of emotion-specific supervision during instruction tuning and post training. Such targeted diagnostic signals can guide improvements to model architectures and training strategies, enabling more principled interventions that reduce hallucination without introducing substantial computational overhead.

---

### Official Review · Reviewer_QF5h · 2025-10-30

**Soundness:** 3
**Presentation:** 3
**Contribution:** 4
**Rating:** 6
**Confidence:** 5

**Summary:**

This paper introduces EmotionHallucer, a novel benchmark designed to evaluate emotion-related hallucinations in multimodal large language models (MLLMs). The benchmark spans four modalities (text, image, audio, video) and is organized around two main dimensions: emotion psychology knowledge and multimodal emotion perception. The authors also propose PEP-MEK, a reasoning-enhanced framework aimed at mitigating emotion hallucinations. Extensive experiments on 41 LLMs and MLLMs reveal widespread emotion hallucination issues, particularly in multimodal perception tasks, and demonstrate the effectiveness of PEP-MEK in improving model robustness.

**Strengths:**

- Novel Benchmark: EmotionHallucer is the first comprehensive benchmark targeting emotion hallucinations, with a well-designed adversarial evaluation protocol.
- Multimodal Coverage: The benchmark spans text, image, audio, and video, enabling a holistic assessment of MLLMs’ emotion understanding capabilities.
- Large-Scale Evaluation: Experiments on 41 models offer broad insights into current limitations and trends.
- Practical Mitigation Framework: PEP-MEK demonstrates consistent improvements across models and modalities, offering a simple yet effective approach to reducing emotion hallucinations.

**Weaknesses:**

- Limited Cross-Lingual and Cultural Analysis: The benchmark is limited to English, and there is no discussion of how cultural or linguistic differences might affect emotion hallucination patterns. This limits the generalizability of the findings.
- Superficial Error Analysis: While the paper reports performance drops in open-set and multimodal settings, it does not deeply investigate the root causes of failures (e.g., which types of cues are most often misinterpreted).
- Benchmark Design Limitations: The binary QA format, while useful for controlled evaluation, may not fully capture the complexity of open-ended emotion understanding in real-world scenarios.
- Lack of Human Baseline: The absence of human performance comparison makes it difficult to gauge the practical significance of the model results.

**Questions:**

- Have the authors considered extending EmotionHallucer to include non-English or cross-cultural emotional expressions? If so, what challenges do they anticipate?
- Could the authors provide more detailed analysis or examples of cases where PEP-MEK fails? Understanding its limitations could help guide future improvements.
- How might the benchmark be adapted to support more open-ended emotion generation or reasoning tasks, beyond binary QA?

---

> ### Author Response · Authors · 2025-11-19
> **Response to Weaknesses 1-2 and Questions 1**
>
> We thank the reviewer for the valuable comments and appreciate the AC, SAC, and PC for managing the review process. We are grateful for the reviewer’s positive remarks on the strengths of our work, and we respond to the weaknesses and questions point by point below.
>
> **Weakness 1** and **Question 1**: We thank the reviewer for raising this important concern. We acknowledge the reviewer's valid points regarding language limitations. As a first pioneering benchmark in the field of emotion hallucination, our priority is to establish a rigorous, controlled, and reproducible foundation.  Our current benchmark uses English as the interaction language with MLLMs primarily for the following two reasons: first, to ensure a controlled and reproducible comparison across the wide spectrum of MLLMs, many of which are primarily optimized for English; and second, because English currently serves as a lingua franca in scientific research and AI development, which facilitates the initial adoption and dissemination of our benchmark within the research community. However, this does not mean that the emotion content itself is culturally narrow.
> The multimodal data in EmotionHallucer, drawn from social media posts, YouTube videos, TV shows, and cinematic content, naturally spans diverse cultural backgrounds. These sources encompass variations in emotion display rules, conversational dynamics, and situational emotional cues, providing rich cultural coverage. In addition, the Emotion Psychology Knowledge part incorporates insights from cross-cultural emotion research. This grounding ensures that our EmotionHallucer is not tied to a single cultural framework and remains widely applicable. That said, we fully acknowledge that systematically examining how cultural and linguistic differences influence emotional interpretation and hallucination patterns goes beyond the scope of the current release and remains an important direction for future work. We plan to extend EmotionHallucer to the following: multilingual settings covering diverse languages, culture-grounded emotional frameworks with culturally specific cues and appraisal norms, parallel multimodal samples enabling direct cross-cultural comparison, and cross-lingual emotional reasoning tasks that test how MLLMs adapt to culturally grounded emotion expressions. A key challenge is that emotion categories, display rules, and multimodal cues vary substantially across cultures, making it non-trivial to construct culturally aligned annotations and comparable multimodal samples. We believe these extensions will enhance the benchmark’s generalizability and offer deeper insights into how language, culture, and multimodal emotion reasoning interact in modern MLLMs.
>
> **Weakness 2**: We thank the reviewer for this helpful observation. We agree that understanding which specific emotion cues lead to hallucinations is essential for deeper mechanistic interpretation. A full mechanistic investigation of all cue types is beyond the scope of this initial release. As noted in our responses to Weakness 3 and Question 3, our planned structured open-ended annotation scheme will allow future versions of the benchmark to systematically capture entity grounding, activated or non-activated emotional cues, emotion categories, intensities, and reasoning pathways, providing the necessary infrastructure for comprehensive cue-level analysis. Nevertheless, our current results already reveal several initial insights (as shown in Table 1 and Table 2 in our response to Reviewer aDze). In the audio and video modalities, we observe that existing MLLMs struggle with prosodic cues and temporal emotional dynamics, suggesting insufficient grounding of vocal rhythm, intonation, and evolving affective states over time. At the same time, we find that current MLLMs perform poorly on the intensity and reasoning-cue subcategories. We believe this reflects the fact that MLLMs do not process emotion information with the same fine-grained sensitivity as humans. We believe this may be because they struggle both with accurately perceiving subtle emotion cues and with resolving conflicts among cues once they are detected.

---

> ### Author Response · Authors · 2025-11-19
> **Response to Weaknesses 3-4 and Questions 3**
>
> **Weakness 3** and **Question 3**: We thank the reviewer for raising these insightful points. We agree that real-world emotional understanding is often open-ended, dynamic, and a binary QA format alone cannot fully capture the complexity. However, as discussed in our paper, the controlled binary setting is intentionally designed as a diagnostic foundation, enabling precise isolation of perception-level vs. inference-level hallucinations, strict control of semantic content, and reproducible comparisons across heterogeneous MLLMs, which open-ended generation inherently lacks due to its variability and unconstrained output space. As a result, the two settings are complementary: the binary format provides fine-grained, artifact-free diagnostic signals, while open-ended tasks capture more naturalistic generative behaviors. Additionally, our analysis shows that this controlled setting reflects real-world tendencies rather than imposing an artificial task. As presented in Appendix E, the Binary vs. Open-Ended Consistency study demonstrates that models exhibit highly similar hallucination patterns across formats, indicating that the binary QA design is a stable and interpretable proxy for naturalistic emotional failures. We acknowledge, however, that extending toward richer real-world scenarios is a valuable direction. In future work, we plan to introduce the following two complementary open-ended extensions: 1. Free-form open-ended emotion understanding, where models produce naturalistic emotional interpretations, causal explanations, and multimodal reasoning narratives, enabling evaluation of generative behavior beyond binary constraints. 2. Structured annotation, where human raters identify entities, activated or non-activated emotional cues, emotion categories, intensities, and reasoning pathways. Such expansions will allow us to localize hallucinations at both span-level and holistic levels in open-ended setting, better reflecting the complexity of real-world emotion understanding. These extensions will allow EmotionHallucer to evolve from a controlled diagnostic benchmark into a broader evaluation suite that captures the full spectrum of emotional reasoning behaviors encountered in real-world MLLM deployments.
>
> **Weakness 4**: We thank the reviewer for this helpful comment. While human performance can be informative in some settings, a direct human baseline for our binary QA pairs is not suitable for the present benchmark. Unlike LLMs, humans cannot be evaluated in a strict single-turn, single-item setting without exposure to both items in a pair. Humans will compare the basic and hallucinated versions, and once they realize that exactly one option is incorrect, this introduces a strong paired-task-level prior that fundamentally alters their reasoning process. Such priors may artificially inflate human accuracy and make the results incomparable to LLMs, whose evaluation is conducted without any pairwise context. Constructing a more complex human-evaluation pipeline to avoid this prior (e.g., multi-stage randomization, distraction items, or pair obfuscation) would impose significant cognitive load. This would undermine the comparability between human judgments and the model setting, which is central to the benchmark’s purpose. That said, we fully agree that human judgments are valuable for assessing more natural forms of emotion understanding. Instead of forcing an artificial human baseline on the binary QA format, we plan to incorporate human evaluation in open-ended settings, where humans can provide free-form emotional interpretations or structured ratings. Such settings avoid the priors inherent to forced binary comparisons and offer a much more meaningful way to align human and model performance in future extensions of EmotionHallucer.

---

> ### Author Response · Authors · 2025-11-19
> **Response to Questions 2**
>
> **Question 2**: We thank the reviewer for the suggestion. In response to the reviewer's question, we have conducted further analysis and found that most failure cases remain concentrated in the audio and video modalities (as shown in Table 1 in our response to Reviewer aDze). Specifically, we observe that (1) in audio-only settings, models often misinterpret or under-detect prosodic cues such as stress, rhythm, or emotional tone, (2) in short videos, models sometimes fail to integrate visual and acoustic signals into a coherent emotional interpretation, and (3) in long videos, emotion temporal grounding and emotion-dynamic tracking remain challenging, leading to inconsistent or drifted predictions. While PEP-MEK substantially reduces hallucinations by enforcing emotion-knowledge consistency, it can still fail when the underlying perceptual cues extracted by the model are incomplete or incorrect. Typical failure cases occur when the model (a) grounds on an incorrect emotional cue during the Description step, or (b) provides an explanation that is logically coherent but based on the wrong perceptual evidence. These limitations point toward future improvements such as strengthening perceptual grounding modules, enhancing long-range emotional dynamic modeling, and developing adaptive explanation strategies that adjust their depth based on modality confidence.

---

### Official Review · Reviewer_abcp · 2025-11-01

**Soundness:** 2
**Presentation:** 2
**Contribution:** 3
**Rating:** 4
**Confidence:** 4

**Summary:**

EmotionHallucer introduces the first systematic benchmark for evaluating emotion-related hallucinations in multimodal large language models (MLLMs). The benchmark is grounded in emotion psychology and real-world multimodal perception and uses adversarial binary QA pairs (basic vs. hallucinated versions) across modalities (text, image, audio, video). The authors evaluate 41 MLLMs (open- and closed-source) on multiple subtests (e.g., perception-level, psychology/knowledge-level, reasoning-result), report quantitative metrics (including bias/FP measures and separate accuracy for basic vs. hallucinated items), and analyze model behaviors. They find that many models remain vulnerable to emotion hallucinations, that closed-source models often outperform open-source ones, and that models are typically better at explicit emotion-knowledge tasks than at grounded multimodal perception and inference. To improve detection, the paper proposes PEP-MEK, a multimodal+emotion-knowledge augmentation framework that boosts hallucination detection and is evaluated via ablations. The paper provides dataset construction, annotation procedures, limitations, and plans to release resources on GitHub.

**Strengths:**

### A novel benchmark (EmotionHallucer) specifically targeting emotion hallucinations in MLLMs:
- Covers multiple modalities and multiple diagnostic levels (perception, emotion knowledge, reasoning results).
- Uses adversarially constructed basic vs. hallucinated QA pairs to probe hallucination propensity.
### Large-scale empirical evaluation and analysis:
- Systematic evaluation of 41 MLLMs (both open- and closed-source) with detailed metrics (Pct. Diff, FP Ratio, separate Basic vs. Hallucinated accuracy, overall scores).
- Insights showing systematic weaknesses (e.g., multimodal perception and reasoning produce more hallucinations; closed-source models typically fare better).
### Proposed mitigation/analysis method (PEP-MEK) and ablation studies:
- PEP-MEK integrates psychology-grounded emotion knowledge and perceptual cues to improve hallucination detection.
- Demonstrated consistent improvements across models and includes ablation studies showing the contribution of emotion knowledge and other components.

**Weaknesses:**

### Annotation noise and scope limited to English and certain datasets:
- The benchmark relies on human annotation (e.g., creating hallucinated variants), admitting annotation noise.
- The dataset is English-only and does not address cross-lingual or cultural variability in emotional expression.
### Partial exploration of root causes:
- While the paper documents hallucination phenomena and correlates them with modality and model class, it does not deeply investigate underlying causes (e.g., pretraining biases, modality misalignment, lack of emotion-specific supervision) or provide mechanistic explanations.
### Separation of evaluation axes and incomplete real-world integration:
- Emotion understanding and hallucination detection are treated separately, whereas practical systems need integrated capabilities (joint perception, inference, and hallucination-awareness).
- Temporal and long-form audio/video integration remain challenging and less explored; the benchmark and methods may not fully capture these complex real-world scenarios.

**Questions:**

SEE WEAKNESS

---

> ### Author Response · Authors · 2025-11-19
> **Response to Weaknesses 1-2**
>
> We thank the reviewer for the valuable comments and appreciate the AC, SAC, and PC for managing the review process. We are grateful for the reviewer’s positive remarks on the strengths of our work, and we respond to the weaknesses and questions point by point below.
>
> **Weakness 1**: We thank the reviewer for raising this important point. We agree that human annotations inevitably introduce some noise. To minimize this risk, we designed a highly controlled, multi-stage annotation pipeline, as detailed in Appendix A. Basic items are sourced directly from well-annotated public datasets, avoiding synthetic biases. Hallucinated items are created through strict minimal edits, modifying only the emotion-relevant semantic component while keeping all other textual attributes stable. Each QA pair is then cross-reviewed by an independent annotator, and only pairs with full agreement are retained; ambiguous cases are discarded. The resulting dataset achieves an 84.6% inter-annotator agreement, demonstrating high reliability despite the inherent subjectivity of the task.
> Regarding the English-only limitation, we acknowledge the reviewer's valid point regarding language limitation. As a first pioneering benchmark in the field of emotion hallucination, our priority is to establish a rigorous, controlled, and reproducible foundation.  Our current benchmark uses English as the interaction language with MLLMs primarily for two reasons: first, to ensure a controlled and reproducible comparison across the wide spectrum of MLLMs, many of which are primarily optimized for English; and second, because English currently serves as a lingua franca in scientific research and AI development, which facilitates the initial adoption and dissemination of our benchmark within the research community. However, this does not mean that the emotion content itself is culturally narrow. The multimodal data in EmotionHallucer, drawn from social media posts, YouTube videos, TV shows, and cinematic content, naturally spans diverse cultural backgrounds. These sources encompass variations in emotion display rules, conversational dynamics, and situational emotional cues, providing rich cultural coverage. In addition, the Emotion Psychology Knowledge part incorporates insights from cross-cultural emotion research. This grounding ensures that our EmotionHallucer is not tied to a single cultural framework and remains widely applicable. That said, we fully acknowledge that systematically examining how cultural and linguistic differences influence emotional interpretation and hallucination patterns goes beyond the scope of the current release and remains an important direction for future work. We plan to extend EmotionHallucer to the following: multilingual settings covering diverse languages, culture-grounded emotional frameworks with culturally specific cues and appraisal norms, parallel multimodal samples enabling direct cross-cultural comparison, and cross-lingual emotional reasoning tasks that test how MLLMs adapt to culturally grounded emotion expressions. We believe these extensions will enhance the benchmark’s generalizability and offer deeper insights into how language, culture, and multimodal emotion reasoning interact in modern MLLMs.
>
> **Weakness 2**: We thank the reviewer for this insightful comment. We agree that understanding the underlying causes of emotion hallucination is an important direction for future research. However, the primary goal of EmotionHallucer is to provide the first systematic benchmark that isolates different forms of emotion hallucination, measuring them across modalities and categories. Establishing this reliable evaluation foundation is what enables targeted mechanistic investigations, moving the field from initial speculation to rigorous, hypothesis-driven research. That said, our analysis provides initial evidence pointing toward several root causes. For example, we observe modality-dependent vulnerability, suggesting potential modality misalignment and weaker emotion understanding beyond text–image signals.  We fully agree that deeper mechanistic analysis (e.g., tracing pretraining corpus biases, probing cross-modal fusion layers, or integrating emotion-specific supervision) would be highly valuable. As future work, we plan to leverage EmotionHallucer as a diagnostic tool to guide such investigations, including probing-based studies, intervention experiments, and emotion-aware fine-tuning diagnostics. We believe that establishing this benchmark is a critical first step toward uncovering the mechanistic sources of emotion hallucination.

---

> ### Author Response · Authors · 2025-11-19
> **Response to Weaknesses 3**
>
> **Weakness 3**: We thank the reviewer for this thoughtful observation. We agree that practical multimodal systems require integrated emotional perception, inference, and hallucination-awareness. We also note that EmotionHallucer is the first attempt to formally measure emotion hallucinations in MLLMs. For a first benchmark, separating emotion understanding and hallucination detection is essential: it provides a clean problem formulation, avoids entangling perception errors with reasoning errors, and establishes the foundational structure upon which more integrated, real-world settings can be built. In this sense, separation is not a limitation but a deliberate design choice to create a principled and interpretable starting point for this new research direction. Our decision to separate these evaluation axes is intentional: the goal of EmotionHallucer is to build the first diagnostic framework that isolates distinct cognitive stages where emotion hallucinations arise. Importantly, this does not prevent integrated evaluation since the paired basic–hallucinated format inherently requires the model to jointly process perception, inference, and grounding to detect hallucinations.
> Regarding the reviewer’s point on temporal and long-form multimodal integration, we fully agree that this remains an open challenge for current MLLMs. Rather than reflecting a limitation of the benchmark, the weak performance of state-of-the-art models highlights precisely why a controlled evaluation like EmotionHallucer is needed. As future work, we plan to extend the benchmark toward longer video sequences, temporally aligned emotion dynamics, multi-turn emotional reasoning, and sustained hallucination tracking, enabling joint assessment of perception, inference, and hallucination-awareness in more realistic scenarios. We believe EmotionHallucer provides the necessary foundation for studying integrated emotional intelligence, and future expansions will move toward richer, more continuous real-world multimodal contexts.

---

### Official Review · Reviewer_aDze · 2025-11-01

**Soundness:** 4
**Presentation:** 3
**Contribution:** 4
**Rating:** 6
**Confidence:** 4

**Summary:**

The paper introduces EmotionHallucer, a benchmark to detect and analyze emotion-related hallucinations in multimodal LLMs. It evaluates two axes:
- Emotion psychology knowledge (theory, definitions, empirical findings)
- Real-world multimodal perception (category, intensity, reasoning cues/results) across text, image, audio, and video.

The benchmark uses adversarial binary QA pairs—a “basic” item and a matched “hallucinated” item—and counts a prediction as correct only if the model answers both correctly, reducing prompt/length confounds seen in captioning metrics and self-evaluation bias.

**Strengths:**

This is the first benchmark that is dedicated to emotion hallucinations, spanning both psychology knowledge and multimodal perception; prior hallucination suites are general-purpose. The seven subcategories (theory/definition/finding; category/intensity/reasoning cue/reasoning result) make the construct very concrete. And, the adversarial paired QA design (basic vs hallucinated) is what I call a neat, low-variance way to test detection of hallucination, beyond typical caption/LLM-judge setups.

The paper is well structured, with clear logic, a well-defined task taxonomy, examples, and an easy-to-follow pipeline; the appendices document the collection/annotation and PEP-MEK details; ethics and reproducibility statements are included.

Non-trivial scale and coverage, with broad evaluation and clear metrics, which is very nice.

**Weaknesses:**

1. Adversarial pair construction & QA artifacts. The process risks introducing superficial cues between the basic and “hallucinated” versions. Report inter-annotator agreement, pair-level quality controls, and checks against annotation artifacts (e.g., spurious lexical markers).

2. Latency/compute overhead and failure cases are not quantified. A wall-clock and token-cost-wise comparison is needed here, along with ablations for each PEP-MEK component and per-subcategory gains.

**Questions:**

1. Again, I would like to know a wall-clock and token-cost-wise comparison, along with ablations for each PEP-MEK component and per-subcategory gains.

2. Bias balancing details. The authors stated that the yes/no is balanced. Can you share the exact balance per subcategory and modality, and how you prevented position or wording biases between paired items?

3. Open-ended evaluation. Beyond the pilot LLM-judge setup, do you plan a human-rated open-ended benchmark slice to validate the binary proxy and reduce judge-model bias?

---

> ### Author Response · Authors · 2025-11-19
> **Response to Weaknesses 1**
>
> We thank the reviewer for the valuable comments and appreciate the AC, SAC, and PC for managing the review process. We are grateful for the reviewer’s positive remarks on the strengths of our work, and we respond to the weaknesses and questions point by point below.
>
> **Weakness 1**: We thank the reviewer for highlighting this important issue. We fully agree that adversarial QA construction must avoid introducing superficial cues, such as sentence length, lexical artifacts, or stylistic patterns, that could allow models to distinguish basic vs. hallucinated items without genuine emotion understanding. Our annotation pipeline (detailed in Appendix A) was designed to prevent such artifacts. First, the basic questions are derived directly from existing, publicly available datasets or authoritative psychology literature, ensuring that they are grounded, coherent, and free from synthetic or stylistic biases. Because annotators do not rewrite these basic items but instead refine existing labeled samples, the basic items naturally avoid artificial patterns. Second, hallucinated items are created via minimal semantic edits. Specifically, annotators modify only the emotion-relevant components, such as emotional category, intensity, or causal reasoning cues, while keeping all unrelated textual attributes (overall syntax, tone, style, length, and vocabulary distribution) unchanged. This “minimal-edit” design prevents the introduction of spurious cues. Third, as shown in Figure 6 in Appendix A, every QA pair undergoes a cross-review process: each pair is independently reviewed by a second annotator, and only pairs where both annotators fully agree on naturalness, correctness, and the absence of annotation artifacts are retained. Ambiguous, borderline, or stylistically inconsistent pairs are discarded. To provide empirical evidence of annotation reliability, we further computed the inter-annotator agreement over the final retained dataset, which reaches 84.6%, demonstrating strong consistency despite the inherent subjectivity of emotional interpretation. This agreement rate reflects not only correctness checks but also the manual screening of unintended lexical or structural artifacts. We believe this careful process supports a reliable and meaningful assessment of MLLMs. Moreover, the experimental results further validate the absence of superficial cues: even the strongest MLLMs consistently struggle with EmotionHallucer. If the pairs contained shallow lexical cues, such performance gaps would not systematically appear across models of different scales and architectures. This converging evidence indicates that the benchmark genuinely evaluates emotional reasoning rather than exploitable surface patterns.

---

> ### Author Response · Authors · 2025-11-19
> **Response to Weaknesses 2 and Questions 1**
>
> **Weaknesses 2** and **Questions 1**:
>
> **Table 1: Wall-clock latency and token-cost comparison for each PEP-MEK component across modalities. The three reported metrics correspond to Overall Accuracy, Token Cost, and Latency.**
>
> | Modality | Baseline            | Description       | PEP-MEK-first        | PEP-MEK-explain&second |
> |-|-|-|-|-|
> | Text     | 19.00/274/0.86       | -/1357/10.09       | 31.00/1357/10.09       | 36.00/1314/3.90          |
> | Image    | 32.67/462/2.47       | -/1499/11.33       | 37.33/1193/8.69        | 40.00/1403/8.80          |
> | Audio    | 0.82/252/57.48       | -/1103/74.54       | 2.99/769/49.68         | 5.43/995/20.27           |
> | Video-S| 13.19/2717/35.09     | -/3584/23.65       | 27.08/3307/2.67        | 29.17/3533/3.91          |
> | Video-L| 5.47/18389/191.43    | -/19878/157.47     | 6.47/19591/45.46       | 17.91/19812/22.71        |
>
> **Table 2: Overall accuracy comparison of PEP-MEK components across the four hallucination subcategories: Category, Intensity, Reasoning-Result, and Reasoning-Cue.**
>
> | Model             | Original input          | + MEK                    | + MEK + Explain          |
> |-|-|-|-|
> | Qwen2.5-Omni      | 14.37/2.67/18.92/6.04    | 18.86/6.87/25.95/11.54    | 21.56/9.92/30.27/21.98    |
> | Emotion-LLaMA     | 11.08/4.58/12.81/5.03    | 18.26/15.27/30.05/14.57   | 25.45/22.90/33.50/23.54   |
> | Gemini-2.5-Flash  | 41.32/27.86/39.38/20.97  | 42.51/32.06/40.32/26.37   | 43.71/37.02/40.82/27.86   |
>
> We thank the reviewer for highlighting this important issue.
>
> As shown in Table 1, we report the token cost and wall-clock latency of each component (Baseline, Description, PEP-MEK-first, and PEP-MEK-explain&second) across all modalities. These measurements were collected using the Qwen2.5-Omni-7B model via the Aliyun Bailian API, and reflect the real execution cost of running each stage in practice. Token cost was obtained by enabling stream_options={"include_usage": True}. Wall-clock time was measured as the elapsed time from sending the request to receive the final streamed chunk. We note that wall-clock latency may not perfectly correspond to pure computational time, as it also reflects network delay, API scheduling latency, and other external factors. Nevertheless, it provides a practical approximation of the real user-perceived cost when deploying these pipelines in real applications. Regarding the per-subcategory gains, we note that token cost and wall-clock latency are fundamentally modality-dependent rather than subcategory-dependent. Therefore, reporting per-subcategory efficiency would not provide meaningful distinctions beyond noise, while modality-level reporting captures the true computational variation. We hope the reviewer finds this clarification reasonable.
>
> Across modalities, we observe that the PEP-MEK components provide stable gains over the baseline. This pattern holds for both perception-aligned (“first”) and inference-aligned (“second”) variants, indicating that structured emotion reasoning benefits all modalities rather than only specific cases. Notably, in more complex scenarios such as Video-Long, the explain stage outperforms the description step, suggesting that explanation-guided reasoning is particularly effective when the emotion cues unfold over long temporal spans. While PEP-MEK introduces additional computational overhead, the increase is moderate and remains practical relative to the substantial accuracy improvements. Furthermore, as shown in Table 7 of Appendix B.2, our approach achieves stronger performance than alternative strategies operating under similar computational cost, demonstrating that the added overhead translates into meaningful emotion hallucination reduction. Additionally, while the latency increases, we argue that for safety-critical tasks like emotion understanding, reliability is paramount over speed.
>
> Beyond modality, we also examined how PEP-MEK affects different hallucination subcategories, as shown in Table 2. We observe stable improvements for both ‘+ MEK’ and ‘+ MEK + Explain’.  These consistent gains across subcategories and models suggest that PEP-MEK strengthens both perceptual grounding and higher-level emotion reasoning in a broadly applicable way, rather than targeting isolated failure modes.
>
> In terms of failure-case quantification, overall accuracy allows us to assess failures in an aggregate manner. Across modalities, we find that most failure cases occur for Audio and Video-Long, indicating that models struggle most with prosodic ambiguity and long-range temporal emotion dynamics. For the subcategories, we further observe that most failures arise from intensity and reasoning-cue errors. This pattern suggests that models often misjudge the subtle variations in emotional intensity and frequently misinterpret or overlook the cues that support an emotion inference.

---

> ### Author Response · Authors · 2025-11-19
> **Response to Questions 2-3**
>
> **Questions 2**: We thank the reviewer for raising this important point. Because EmotionHallucer is designed around paired basic–hallucinated questions, each pair naturally forms a 1:1 balanced yes/no. Therefore, the yes/no balance is not only maintained at the dataset level but is guaranteed within each hallucination subtype and each modality (text, image, audio, and video), ensuring fully balanced evaluation across all components of the benchmark. To clarify the reviewer’s concern, the evaluation protocol does not expose the basic and hallucinated items to the model in the same prompt. Each question is presented independently, and the tested MLLM never sees the paired item. Therefore, there is no opportunity for position bias at the question level. Regarding word position and wording bias, as discussed in our response to Weakness 1, the basic items are sourced directly from well-annotated descriptions or labels in existing public datasets, ensuring groundedness and eliminating the possibility of synthetic stylistic artifacts. For the hallucinated items, we follow a strict minimal-edit protocol: annotators modify only the emotion-relevant semantic component, such as emotional category, intensity, or causal inference, while keeping all other textual attributes (sentence length, syntactic structure, phrasing patterns, and lexical distribution) unchanged. This controlled construction prevents spurious cues from emerging between the basic and hallucinated versions, ensuring that model errors stem from genuine emotion misinterpretation rather than superficial linguistic patterns.
>
> **Questions 3**: We thank the reviewer for raising these insightful points. We agree that real-world emotion understanding is often open-ended, and a binary QA format alone cannot fully capture the complexity. However, as discussed in our paper, the controlled binary setting is intentionally designed as a diagnostic foundation, enabling precise isolation of emotion hallucinations, strict control of semantic content, and reproducible comparisons across MLLMs, which open-ended generation inherently lacks due to its variability and unconstrained output space. As a result, the two settings are complementary: the binary format provides fine-grained, artifact-free diagnostic signals, while open-ended tasks capture more naturalistic generative behaviors. Additionally, our analysis shows that this controlled setting reflects real-world tendencies rather than imposing an artificial task. As presented in Appendix E, the Binary vs. Open-Ended Consistency study demonstrates that models exhibit highly similar hallucination patterns across formats, indicating that the binary QA design is a stable and interpretable proxy for naturalistic emotional failures. We acknowledge, however, that extending toward richer real-world scenarios is a valuable direction. In future work, we plan to introduce the following two complementary open-ended extensions: 1. Free-form human-rated open-ended emotion understanding, where models produce naturalistic understanding, and humans would then rate these responses along multiple dimensions. 2. Structured annotation as mentioned in Appendix E, where human raters identify entities, activated or non-activated emotional cues, emotion categories, intensities, and reasoning pathways. Given the high cost of high-quality human rating, we view structured annotation as a more feasible alternative to fully free-form human ratings. Such annotation will allow us to localize hallucinations at both span-level and holistic levels in open-ended setting, better reflecting the complexity of real-world emotion understanding. These extensions will allow EmotionHallucer to evolve from a controlled diagnostic benchmark into a broader evaluation suite that captures the full spectrum of emotion understanding behaviors encountered in real-world MLLMs.

---

### Official Review · Reviewer_Fm9U · 2025-11-02

**Soundness:** 2
**Presentation:** 2
**Contribution:** 2
**Rating:** 4
**Confidence:** 3

**Summary:**

The paper introduces EmotionHallucer, the first benchmark designed to evaluate emotional hallucinations in multimodal large language models (MLLMs). The findings indicate that most current models exhibit significant issues with emotional understanding, particularly concerning hallucinations. By introducing the PEP-MEK framework, the authors demonstrate an average performance improvement of 9.90% in emotion hallucination detection. The study draws from emotion psychology knowledge and real-world multimodal perception, providing a comprehensive evaluation perspective. Overall, the paper contributes valuable tools and directions for future research in the field of emotional understanding.

**Strengths:**

This study fills a critical gap in the evaluation of emotional hallucinations, offering the first benchmark tailored for MLLM emotional understanding. The introduction of the PEP-MEK framework shows significant effectiveness, enhancing model performance in hallucination detection. The authors provide robust experimental data and statistical evidence to support their conclusions, increasing the paper's credibility. The research methodology integrates insights from emotion psychology, ensuring the scientific validity of the assessments. Additionally, the use of diverse multimodal data sources adds practical relevance to the findings.

**Weaknesses:**

Language Limitation: The study is restricted to English, failing to account for cross-linguistic and cross-cultural variations in emotional expression.
Complex Definitions: The definitions and classifications of emotional hallucinations may be overly intricate, potentially leading to ambiguity in the evaluation process.
Suboptimal Performance: Model performance in processing multimodal data, particularly in audio and video emotional understanding, remains inadequate.
Result Stability: The stability and reliability of certain experimental results need further validation to ensure consistency.
Real-World Reflection: The assessment of existing models may not accurately reflect their performance variations in practical applications.

**Questions:**

In defining emotional hallucinations, how can researchers effectively balance scientific rigor with interpretability?

Is the current benchmark adaptable to different types of multimodal models to ensure consistency in evaluation?

Can the PEP-MEK framework be further optimized for additional emotional understanding tasks beyond those currently evaluated?

---

> ### Author Response · Authors · 2025-11-19
> **Response to Weaknesses 1–3**
>
> We thank the reviewer for the valuable comments and appreciate the AC, SAC, and PC for managing the review process. We are grateful for the reviewer’s positive remarks on the strengths of our work, and we respond to the weaknesses and questions point by point below.
>
> **Weakness 1** (Language Limitation): We thank the reviewer for raising this important concern. We acknowledge the reviewer's valid points regarding language limitations. As a first pioneering benchmark in the field of emotion hallucination, our priority is to establish a rigorous, controlled, and reproducible foundation.  Our current benchmark uses English as the interaction language with MLLMs primarily for two reasons: first, to ensure a controlled and reproducible comparison across the wide spectrum of MLLMs, many of which are primarily optimized for English; and second, because English currently serves as a lingua franca in scientific research and AI development, which facilitates the initial adoption and dissemination of our benchmark within the research community.
> However, this does not mean that the emotion content itself is culturally narrow. The multimodal data in EmotionHallucer, drawn from social media posts, YouTube videos, TV shows, and cinematic content, naturally spans diverse cultural backgrounds. These sources encompass variations in emotion display rules, conversational dynamics, and situational emotional cues, providing rich cultural coverage. In addition, the Emotion Psychology Knowledge part incorporates insights from cross-cultural emotion research. This grounding ensures that our EmotionHallucer is not tied to a single cultural framework and remains widely applicable. That said, we fully acknowledge that systematically examining how cultural and linguistic differences influence emotional interpretation and hallucination patterns goes beyond the scope of the current release and remains an important direction for future work. We plan to extend EmotionHallucer to the following: multilingual settings covering diverse languages, culture-grounded emotional frameworks with culturally specific cues and appraisal norms, parallel multimodal samples enabling direct cross-cultural comparison, and cross-lingual emotional reasoning tasks that test how MLLMs adapt to culturally grounded emotion expressions. We believe these extensions will enhance the benchmark’s generalizability and offer deeper insights into how language, culture, and multimodal emotion reasoning interact in modern MLLMs.
>
> **Weakness 2** (Complex Definitions): We appreciate the reviewer’s thoughtful concern. We would like to clarify that the definitions and classifications of emotion hallucinations in EmotionHallucer are not intended to be unnecessarily complicated; rather, they are designed to reflect the inherent dynamic and rigor of human emotion understanding itself. As illustrated in Figure 1, human emotion involves dynamic interactions among cognitive appraisals, physiological changes, feelings, and behaviors. Emotion hallucinations can therefore manifest at different levels of this pipeline. To capture this natural hierarchy, we separate hallucinations into subtypes grounded in established emotion psychology theories and structured multimodal perception research. Each category corresponds to a distinct cognitive or perceptual step, which enhances the clarity of evaluation by allowing us to pinpoint where the model’s reasoning deviates from human norms. Thus, the taxonomy is not an unnecessary complication, but a scientifically motivated decomposition that enables precise, interpretable, and reproducible assessment of emotion hallucinations.
>
> **Weakness 3** (Suboptimal Performance): We thank the reviewer for this observation. We agree that current MLLMs exhibit notably weaker performance on audio and video emotion understanding. However, we would like to clarify that **this is a reflection of the models’ current limitations, not a limitation of the benchmark itself**. A central motivation behind EmotionHallucer is precisely to reveal how modern MLLMs behave in emotion-related hallucination scenarios. The performance disparities we observe, stronger results on text and image but weaker results on audio and video, directly demonstrate the value of our benchmark. Importantly, these findings also point to natural future directions for both model development and benchmark evolution. In future work, we plan to extend the benchmark toward emotion dynamics in longer video sequences, temporal alignment tasks, and richer prosody-based emotional cues, enabling a more comprehensive assessment of how emotion understanding and hallucination interact over time.

---

> ### Author Response · Authors · 2025-11-19
> **Response to Weaknesses 4–5 and Questions 1**
>
> **Weakness 4** (Result Stability): We thank the reviewer for highlighting the importance of result stability. In our experiments, we ensured reproducible outputs by using deterministic decoding settings, e.g., temperature = 0 and greedy decoding where applicable, to eliminate variability introduced by stochastic sampling, which is standard practice in LLM experiments. Additionally, since EmotionHallucer adopts a binary QA format rather than open-ended generation, model outputs are inherently stable across inference passes and are far less sensitive to decoding randomness. Furthermore, the benchmark design, paired basic/hallucinated items, further reduces potential variance and ensures that the evaluation reflects systematic model behavior. These properties jointly contribute to the inherent stability of the reported results.
>
> **Weakness 5** (Real-World Reflection): We thank the reviewer for raising these insightful points. We agree that real-world emotional understanding is often open-ended, dynamic, and a binary QA format alone cannot fully capture the complexity. However, as discussed in our paper, the controlled binary setting is intentionally designed as a diagnostic foundation, enabling precise isolation of perception-level vs. inference-level hallucinations, strict control of semantic content, and reproducible comparisons across heterogeneous MLLMs, which open-ended generation inherently lacks due to its variability and unconstrained output space. As a result, the two settings are complementary: the binary format provides fine-grained, artifact-free diagnostic signals, while open-ended tasks capture more naturalistic generative behaviors. Additionally, our analysis shows that this controlled setting reflects real-world tendencies rather than imposing an artificial task. As presented in Appendix E, the Binary vs. Open-Ended Consistency study demonstrates that models exhibit highly similar hallucination patterns across formats, indicating that the binary QA design is a stable and interpretable proxy for naturalistic emotional failures. We acknowledge, however, that extending toward richer real-world scenarios is a valuable direction. In future work, we plan to introduce the following two complementary open-ended extensions: 1. Free-form open-ended emotion understanding, where models produce naturalistic emotional interpretations, causal explanations, and multimodal reasoning narratives, enabling evaluation of generative behavior beyond binary constraints. 2. Structured annotation, where human raters identify entities, activated or non-activated emotional cues, emotion categories, intensities, and reasoning pathways. Such expansions will allow us to localize hallucinations at both span-level and holistic levels in open-ended setting, better reflecting the complexity of real-world emotion understanding. These extensions will allow EmotionHallucer to evolve from a controlled diagnostic benchmark into a broader evaluation suite that captures the full spectrum of emotional reasoning behaviors encountered in real-world MLLM deployments.
>
> **Questions 1**: We thank the reviewer for this insightful question. In EmotionHallucer, we balance scientific rigor and interpretability by grounding our definitions in established emotion psychology theories while structuring them into clear, operational, and human-interpretable categories. The taxonomy is derived from well-documented cognitive pathways in emotional processing, as illustrated in Figure 1, ensuring strong theoretical grounding and taking into account structured multimodal perception research. As discussed in our response to Complex Definitions, emotion understanding is inherently multi-stage and dynamic, and our decomposition reflects this natural structure rather than introducing unnecessary complexity. Thus, rigor and interpretability are not opposing goals: the scientifically motivated decomposition provides conceptual clarity, and the task design ensures that each type of emotion hallucination can be directly observed, diagnosed, and reasoned about in practice. This balance enables fine-grained analysis of model failures without sacrificing accessibility or practical utility.

---

> ### Author Response · Authors · 2025-11-19
> **Response to Questions 2-3**
>
> **Questions 2**: We thank the reviewer for this valuable question. EmotionHallucer is designed to be model-agnostic and highly adaptable: the binary QA format, standardized multimodal inputs, and unified prompting procedures allow the benchmark to be applied consistently across models. This design enables fair comparison without requiring model-specific modifications or access to internal weights. In practice, the benchmark has already been applied to 41 diverse MLLMs as shown in Table 8 of Appendix C.1, including open-source and closed-source systems, text, image-text, audio-text, video-text, and video-text-audio models. All models are evaluated under the same QA templates, input organization, and decoding settings, demonstrating that the framework generalizes well across different model families.
>
> **Questions 3**: We thank the reviewer for this thoughtful question. The current PEP-MEK framework is primarily designed to enhance emotional knowledge extraction, and thereby reduce emotion hallucinations. Beyond mitigating hallucinations, we agree that the same multi-stage reasoning paradigm can be extended to broader emotion understanding tasks. Since PEP-MEK decomposes the process into interpretable stages, prediction, explanation, and re-prediction, it naturally supports extensions to more complex scenarios such as emotion-cause reasoning, emotion dynamics across longer interactions, emotion-context consistency, and social commonsense driven emotional inference. The modular design of PEP-MEK makes such adaptations straightforward: each stage can be enriched or specialized for new tasks while maintaining the overall reliability and interpretability of the pipeline. We therefore view PEP-MEK not only as a hallucination mitigation method, but as a generalizable multi-stage reasoning framework for future emotion understanding research. We plan to explore these extensions in subsequent versions of the benchmark.
>
> We sincerely thank the reviewer for these thoughtful comments, which helped us clarify the broader importance of this work. EmotionHallucer is intended not just to highlight an emerging challenge, but to provide the first systematic, psychology-grounded framework for measuring and understanding emotion hallucinations in MLLMs. We believe this contribution fills a critical gap: until now, the community had no standard way to test whether MLLMs hallucinate emotions or to measure how much they improve. Without such a benchmark, it is very hard to evaluate, compare, or meaningfully advance emotionally reliable AI. By coupling this benchmark with concrete improvement strategies such as PEP-MEK, we hope to establish both a diagnostic foundation and a clear path forward for future research. We appreciate the reviewer’s engagement with our work and sincerely hope that our revisions make its motivation and impact more evident.

---

> > ### Comment · Reviewer_Fm9U · 2025-11-24
> >
> > Dear Authors,
> >
> > Thank you for the rebuttal. I note that you plan to make several important modifications. Could you please provide the specific details of these planned changes? For instance, providing the revised sentences or paragraphs and indicating exactly where they will be included would be extremely helpful for evaluating the improved work. I am willing to raise my score if you can provide these details.

---

> > > ### Author Response · Authors · 2025-11-25
> > >
> > > Dear Reviewer Fm9U,
> > >
> > > We sincerely thank the reviewer for the constructive follow-up question and for the willingness to reconsider the score. As noted in our rebuttal, several of the reviewer’s comments (e.g., on definition complexity and model adaptability) have already been clarified conceptually. Beyond these clarifications, we have now integrated all planned future work directly into the updated manuscript (with new content marked in blue). Below, we summarize the concrete changes and indicate where they appear in the revised paper:
> > >
> > > - **Appendix Section E — Limitations and Discussions** (Page 38, Line 2042)
> > >
> > > We revised the discussion of language limitations to clarify that English is used only as the interaction language with MLLMs, while the underlying multimodal data remain culturally diverse. We also added an explanation of how linguistic and cultural factors may influence emotion understanding and hallucination, providing a clearer bridge toward future cross-lingual and cross-cultural extensions.
> > >
> > > (Addresses Weakness 1: Language Limitation)
> > >
> > > - **Appendix Section F — Future Work** (Page 42, Line 2214)
> > >
> > > We created an expanded and standalone Future Work section, which now outlines concrete, actionable extensions triggered by the reviewer’s comments:
> > >
> > > 1. **Open-ended evaluation** (Page 42, Line 2225):
> > >
> > > We added detailed descriptions of our pilot open-ended study and introduced the envisioned structured annotation design for emotion understanding. Figure 24 (Page 44) illustrates the envisioned structured annotation format.
> > >
> > > (Addresses Weakness 5: Real-World Reflection)
> > >
> > > 2. **Cross-lingual and cross-cultural extensions** (Page 42, Line 2260):
> > >
> > > This subsection builds directly on the revised limitation above and outlines how future versions of EmotionHallucer will incorporate multilingual data, cultural frameworks, and cross-cultural reasoning tasks.
> > >
> > > (Addresses Weakness 1: Language Limitation)
> > >
> > > 3. **Deeper mechanistic analysis** (Page 43, Line 2273):
> > >
> > > We added a new subsection discussing how the benchmark can support future investigations into pretraining bias, modality misalignment, and insufficient emotion-specific supervision.
> > >
> > > (Addresses Weakness 5: Real-World Reflection)
> > >
> > > 4. **Joint emotion understanding and hallucination** (Page 43, Line 2283):
> > >
> > > We expanded this subsection to explain why a staged evaluation is necessary at this early stage, and how our structured open-ended design provides a promising foundation for future unified evaluation.
> > >
> > > (Addresses Weakness 5: Real-World Reflection)
> > >
> > > These revisions connect each identified limitation with a concrete methodological avenue for improvement, showing how EmotionHallucer can evolve into a more comprehensive and practically relevant evaluation suite. We hope this provides the reviewer with clear visibility into how the manuscript has been strengthened.
> > >
> > > We sincerely appreciate the reviewer’s thoughtful feedback and willingness to re-evaluate the submission. Your comments have substantially improved the clarity and impact of the final manuscript.
> > >
> > > Best regards,
> > >
> > > The Authors of Submission 7063

---

### Author Response · Authors · 2025-12-03
**Review and Reviewer-Author Discussion Summary**

Dear PCs, SACs, AC, and Reviewers,

Thank you very much for your valuable contributions to our work. We are writing to provide a concise summary of the discussions to aid your final assessment, particularly as the discussion phase concluded unexpectedly.

---

**Strengths**. We are pleased that the reviewers recognized the novelty, relevance, and methodological rigor of EmotionHallucer, and that they provided broadly positive evaluations. Specifically:

- **This is the first benchmark dedicated to studying emotion hallucinations in MLLMs.** (Fm9U: Strength 1; aDze: Strength 1; abcp: Strength 1; QF5h: Strength 1; jLHy: Strength 1)
Reviewers consistently highlighted that our work fills a previously unexplored yet important gap, establishing the first benchmark specifically tailored to emotional hallucinations in multimodal LLMs.

- **The benchmark is grounded in emotion psychology and multimodal perception.** (Fm9U: Strength 4 & 5; aDze: Strength 1; abcp: Strength 3; QF5h: Strength 2; jLHy: Strength 3)
Reviewers appreciated that the benchmark builds on established findings in emotion psychology and incorporates real-world multimodal cues, making the evaluation scientifically valid and well-defined. They also noted the value of the controlled adversarial QA design and minimal-edit construction.

- **The evaluation is comprehensive and large-scale, covering 41 models across modalities.** (Fm9U: Strength 3; aDze: Strength 3; abcp: Strength 2; QF5h: Strength 3; jLHy: Strength 3)
Reviewers emphasized that the benchmark provides broad and systematic insights into current MLLMs, with clear metrics, structured analysis, and non-trivial coverage across four modalities. They also noted the clarity, well-organized pipeline, and usefulness of the appended documentation.

- **PEP-MEK provides a practical and effective mitigation method.** (Fm9U: Strength 2; aDze: Strength 2; abcp: Strength 3; QF5h: Strength 4; jLHy: Strength 4)
Reviewers agreed that PEP-MEK is an effective and broadly applicable mitigation approach that integrates psychology-informed emotion knowledge with perceptual cues, producing consistent improvements across models and modalities. They also appreciated the ablation studies that clarify component contributions.

---

> ### Author Response · Authors · 2025-12-03
>
> **Concerns and Our Response.** During the discussion period, we actively addressed reviewers’ concerns. Specifically:
>
> - **Annotation quality regarding potential annotation artifacts and noise.** (Fm9U: Weakness 2 and Question 1, aDze: Weakness 1 and Question 2, abcp: Weakness 1)
> In response, we explained in detail that the annotation pipeline (Appendix A) was designed precisely to avoid such issues. Additionally, we reported an 84.6% inter-annotator agreement, demonstrating strong reliability. We believe this addresses the concerns regarding annotation quality and potential artifacts.
>
> - **Cultural and linguistic considerations.** (Fm9U: Weakness 1, abcp: Weakness 1, QF5h: Weakness 1 and Question 1, jLHy: Weakness 2)
> In response, we clarified that English is used only as the interaction language, while the multimodal data itself already spans diverse cultural contexts. We further explained that systematic cross-lingual extensions are natural follow-up directions rather than missing components of the current benchmark. To support this, we also outlined concrete plans for multilingual and cross-cultural extensions in Appendix E, illustrating how the benchmark can naturally evolve without affecting the completeness of the present contribution. (Revision: Page 38, Line 2042; Page 42, Line 2260)
>
> - **Open-ended evaluation and real-world concern** (Fm9U: Weakness 5, aDze: Question 3, abcp: Weakness 3, QF5h: Weakness 3, jLHy: Weakness 1)
> As the first benchmark in this area, we believe that establishing a controlled and reproducible evaluation protocol is more essential. In response, we explained the diagnostic motivation behind binary QA and provided evidence showing consistency between binary and open-ended settings in Appendix. In addition, we outlined an expected structured framework for future open-ended evaluation in Appendix E. (Revision: Page 42, Line 2225; Page 43, Line 2283)
>
> - **Ablations and failure patterns of PEP-MEK.** (aDze: Weakness 2 and Question 1, QF5h: Weakness 2 and Question 2)
> In response, we added latency and token-cost analysis, subcategory-level improvements, and described common failure patterns. These additions strengthened the technical completeness of the method. (Revision: Page 25, Line 1315)
>
> - **Deeper mechanistic interpretation.** (abcp: Weakness 2)
> In response, we provided initial insights into emotion hallucination patterns, and we emphasized that EmotionHallucer already establishes the core foundation needed for deeper causal analysis. The planned structured open-ended and cross-lingual extensions serve as natural future pillars that build on this foundation, enabling progressively richer mechanistic investigations rather than reflecting gaps in the current work. (Revision: Page 43, Line 2273)
>
> - **Future Development of PEP-MEK.** (Fm9U: Question 3, jLHy: Weakness 3)
> In response, we provided ideas for the broader and more efficient development of PEP-MEK in the future.
>
> - **Suboptimal Performance.** (Fm9U: Weakness 3), Result Stability (Fm9U: Weakness 4), Adaptable to different MLLMs (Fm9U: Question 2), Human Baseline (QF5h: Question 3).
> In response, we provided clarifications on these items. First, the suboptimal performance of certain models is not a limitation of the benchmark itself but rather reflects the inherent difficulty of emotion hallucination detection. Second, regarding result stability, we adopted standard control measures commonly used in large-model evaluations to ensure consistent and fair comparisons. Third, on adaptability, our experiments already cover a wide range of different MLLMs, demonstrating the generality of the benchmark across architectures and modalities. Finally, we clarified that a human baseline is not directly applicable because humans naturally possess pair-level priors that make the basic–hallucinated distinction qualitatively different from the model setting.
>
> **Revision Summary.** For ease of review, all revised and newly added content is highlighted in blue. We incorporated all relevant feedback, including the supplementary PEP-MEK ablation experiments (Pages 25–26), the refined description of the language scope (Page 38), and the separation and expansion of the future-work discussion into its own section (Pages 42–44).
>
> **Recognition of Our Responce from Reviewers.** The reviewer Fm9U expressed willingness to raise the score, and we provided all requested modifications with line numbers. Unfortunately, due to the unexpected interruption of the discussion period, the reviewer did not have the opportunity to update their score, but the stated intention was clear.
>
> ---
>
> We hope the above summary supports the AC in making an informed and balanced decision under the current constraints of the review process. We sincerely appreciate the efforts of the reviewers, AC, SAC, and PC. Their insightful feedback has further strengthened our paper, and we are grateful for their work.
>
> Best regards,
>
> The Authors of Submission 7063

---

### Meta-Review · Area_Chair_CLjG · 2026-01-07

**Summary:**

Reviewers unanimously recognize the paper as a strong and timely contribution, introducing the first dedicated benchmark for emotion hallucinations in multimodal large language models. The EmotionHallucer benchmark is widely praised for its psychology-grounded design, adversarial paired QA protocol, multimodal coverage (text, image, audio, video), and large-scale evaluation across 40+ models. The proposed PEP-MEK framework is viewed as a practical and effective mitigation strategy, with consistent performance gains and supportive ablation studies. The paper is also commended for clear organization, methodological rigor, and reproducibility.
The main concerns focus on generalizability and depth of analysis. Multiple reviewers note the English-only setting and lack of cross-linguistic or cross-cultural evaluation, limiting real-world applicability. The adversarial binary QA format, while controlled and low-variance, may not fully capture open-ended or long-form emotion understanding in realistic scenarios. Reviewers also request deeper error and root-cause analyses, including investigation of annotation artifacts, modality-specific failures (especially audio/video), and missing human baselines. In addition, efficiency aspects such as latency and computational overhead of PEP-MEK are insufficiently quantified. Overall, the work is seen as a high-quality benchmark and analysis paper. Therefore, AC's recommendation is to accept as a poster paper.

**Reviewer Concerns:**

Most of the reviewers' questions have been effectively addressed. The author explains the rationale for adopting the binary QA format and the reasons for not applying a human baseline, and conducts experiments on the efficiency aspects of PEP-MEK, including latency and computational overhead. The author also provides a detailed explanation of how the annotation pipeline demonstrates reliability.

**Reviewer Scores:**

Reviewer Fm9U indicated willingness to raise the rating after the authors provide specific details. Considering the authors have submitted a detailed response, I believe reviewer Fm9U will increase the rating. I expect the final rating to be as follows:
- Reviewer Fm9U: 6
- Reviewer aDze: 6
- Reviewer abcp: 4
- Reviewer QF5h: 6
- Reviewer jLHy: 8

---

### Decision · Program_Chairs · 2026-01-26

Accept (Poster)